# Flow-Based Density Ratio Estimation for Intractable Distributions with Applications in Genomics

**Egor Antipov** [* 1 2 3] **Alessandro Palma** [* 1 2 3] **Lorenzo Consoli** [* 1] **Stephan Günnemann** [2 3 4]
**Andrea Dittadi** [1 2 3] **Fabian J. Theis** [1 2 3 4 5]

## Abstract

Estimating density ratios between pairs of intractable data distributions is a core problem in probabilistic modeling, enabling principled comparisons of sample likelihoods under different data-generating processes across conditions. While exact-likelihood models such as normalizing flows offer a promising approach to density ratio estimation, naive evaluations are computationally expensive and prone to discretization errors because they require simulating each distribution's likelihood independently. In this work, we leverage condition-aware flow matching to derive a single dynamical formulation for tracking density ratios along generative trajectories. We demonstrate competitive performance on simulated benchmarks for closed-form ratio estimation, and show that our method supports versatile tasks in single-cell genomics data analysis, where likelihood-based comparisons of cellular states across experimental conditions enable treatment effect estimation and batch correction evaluation.

## 1. Introduction

Estimating the density of high-dimensional data has been a central research objective in both theoretical (Silverman, 2018; Chen et al., 2018a) and applied (Cranmer et al., 2020; Noé et al., 2019) settings. The ability to identify modes and evaluate likelihoods in complex feature spaces enables a deeper understanding of the underlying data-generating pro-

cess, laying the foundation for questions surrounding uncertainty quantification (Kuleshov & Deshpande, 2022; Charpentier et al., 2020) and the principled comparison of probabilistic models (Gutmann & Hyvärinen, 2012; Rhodes et al., 2020). In this context, generative models have emerged as a prominent data-driven approach to density estimation, with methods such as Continuous Normalizing Flows (CNFs) (Chen et al., 2018b) learning expressive approximations of data distributions and enabling the exact evaluation of likelihoods for individual observations.

A notable application of density estimation concerns the study of *likelihood ratios*, in which the likelihoods of a data point under two alternative probabilistic models are compared as a ratio (Kanamori, 2010). While this strategy is both simple and effective, evaluating sample likelihoods separately across different models poses computational challenges, particularly when likelihood approximations involve integral expressions and introduce discretization errors, as in exact likelihood models such as CNFs.

In this work, we introduce a method to efficiently evaluate the likelihood ratios of data points across different conditional models, using CNFs trained with flow matching (Liu et al., 2022b; Albergo & Vanden-Eijnden, 2023; Lipman et al., 2023). Rather than independently estimating likelihoods and forming their ratio, we derive a single dynamical formulation to track the ratio along generative trajectories from data to noise. Our method only requires standard CNF models and estimates likelihood ratios at inference time via the compositional combination of vector fields and density scores. We show improved performance on data-driven synthetic likelihood ratio estimation, while use cases can be extended to other fields requiring density comparisons, such as hypothesis testing and anomaly detection.

Moreover, we explore real-world applications of our likelihood ratio estimation method to high-dimensional cellular data, in which the gene expression of individual cells is measured by single-cell RNA sequencing (scRNA-seq) across heterogeneous conditions (Jovic et al., 2022). Our model enables principled, likelihood-based comparison of biological states arising from distinct experimental designs. Motivated by applications to cellular biology, we call the tool grounded

---

[*]Equal contribution [1]Institute of Computational Biology, Helmholtz Munich [2]School of Computation Information and Technology, Technical University of Munich [3]Munich Center for Machine Learning (MCML) [4]Munich Data Science Institute (MDSI) [5]TUM School of Life Sciences Weihenstephan, Technical University of Munich. Correspondence to: Fabian J. Theis <fabian.theis@helmholtz-munich.de>, Andrea Dittadi <andrea.dittadi@gmail.com>.

*Proceedings of the 43ʳᵈ International Conference on Machine Learning*, Seoul, South Korea. PMLR 306, 2026. Copyright 2026 by the author(s).

in our framework **s**ingle-**c**ell **R**atio (scRatio).

We summarize our main contributions as follows:

- We derive a dynamical formulation to estimate density ratios between CNF models using a single simulation.
- We introduce an inference procedure combining vector fields and score functions learned via flow matching to estimate likelihood ratios at individual data points.
- We demonstrate competitive performance on synthetic closed-form ratio benchmarks across multiple dimensions.
- We showcase the practical utility of our model, scRatio, on genomics data, supporting multiple conditional comparison tasks such as the analysis of perturbation data and the evaluation of batch correction algorithms.

## 2. Related Works

**Likelihood estimation with generative models.** Our approach is related to likelihood approximation frameworks based on CNFs (Chen et al., 2018b; Tong et al., 2024a; Jing et al., 2022) and diffusion models (Song et al., 2021; Karczewski et al., 2025a). We further build on ideas from steering the dynamics of generative models, as commonly done in guidance methods (Ho & Salimans, 2021; Zheng et al., 2023; Bansal et al., 2023) and in compositional generation (Liu et al., 2022a; Skreta et al., 2025a). In this context, recent work on compositional diffusion (Skreta et al., 2025a) introduces strategies for computing the density of a model along trajectories simulated by another one, while a related formulation for density guidance is presented in Karczewski et al. (2025b). We draw inspiration from these approaches to formalize our simulation-based likelihood ratio estimation.

**Density ratio estimation.** Traditional methods for density ratio estimation include moment matching, probabilistic classification, and direct ratio matching, as reviewed in Kanamori (2010), while recent work has revisited this problem. Rhodes et al. (2020) propose a strategy that combines probability paths between distributions with classification-based objectives to estimate density ratios. Other methods exploit Time Score Matching (TSM) (Choi et al., 2022; Chen et al., 2026), which expresses the log density ratio as the integral of the time-derivative of intermediate densities along an interpolation path. In this setting, authors in Yu et al. (2025) introduce a diffusion-based extension of TSM with a closed-form objective on data-conditioned probability paths. In contrast, our model avoids interpolating between numerator and denominator densities and instead leverages a single conditional CNF to track density ratios dynamically, without retraining for each pair of compared distributions.

**Likelihood comparisons for scRNA-seq.** Density estimation with Gaussian processes and mixture models has been used to identify prioritized cellular states during differentiation processes (Otto et al., 2024) and compare sam-

ples of cells as distributions (Wang et al., 2024). Other methods focus on differential abundance analysis, assessing whether a cell state is enriched according to different conditional distributions approximated with k-Nearest-Neighbors (kNN) algorithms (Dann et al., 2022; Burkhardt et al., 2021) or Variational Autoencoders (VAE) (Boyeau et al., 2025). However, none of the mentioned studies relies on flexible, parameterized exact-likelihood estimation as in scRatio.

## 3. Background

### 3.1. Continuous Normalizing Flows

**Unconditional CNFs.** We consider a generative model over $\mathbb{R}^d$ as a time-resolved probability density $p_t : \mathbb{R}^d \to \mathbb{R}_{>0}$, with $t \in [0, 1]$ and $\int p_t(\boldsymbol{x}) \, \mathrm{d}\boldsymbol{x} = 1$, such that $p_0$ is a standard normal distribution and $p_1$ approximates a complex data distribution $q$ from which we wish to sample. We use $\boldsymbol{x}_t \in \mathbb{R}^d$ to denote a sample from the time-marginal distribution with density $p_t$. CNFs learn a time-dependent velocity field $u_t : \mathbb{R}^d \to \mathbb{R}^d$ generating the family $\{p_t\}_{t \in [0,1]}$ with partial time-derivative governed by the continuity equation:

$$\frac{\partial}{\partial t} p_t(\boldsymbol{x}) = -\nabla_{\boldsymbol{x}} \cdot \left( p_t(\boldsymbol{x}) \, u_t(\boldsymbol{x}) \right). \quad (1)$$

Specifically, $u_t$ defines the velocity of the following Ordinary Differential Equation (ODE):

$$\frac{\mathrm{d}}{\mathrm{d}t} \boldsymbol{x}_t = u_t(\boldsymbol{x}_t), \qquad \boldsymbol{x}_0 \sim \mathcal{N}(\mathbf{0}_d, \mathbb{I}_d), \quad (2)$$

where $\mathbf{0}_d$ and $\mathbb{I}_d$ are the $d$-dimensional zero-vector and identity matrix. The solution to Eq. (2) transports samples from the prior $p_0$ to the approximate data distribution $p_1$.

CNFs enable exact likelihood computation on a data sample $\boldsymbol{x}_1$ and can be learned via maximum likelihood optimization. Using Eq. (1) and expanding the total time derivative $\frac{\mathrm{d}}{\mathrm{d}t} \log p_t(\boldsymbol{x}_t)$, Chen et al. (2018b) derived an exact expression for the log-density of the data under the model at $t = 1$:

$$\log p_1(\boldsymbol{x}_1) = \log p_0(\boldsymbol{x}_0) - \int_0^1 \nabla_{\boldsymbol{x}_t} \cdot u_t(\boldsymbol{x}_t) \, \mathrm{d}t, \quad (3)$$

where $\boldsymbol{x}_0$ and $\boldsymbol{x}_t$ are defined as in Eq. (2), and $p_0$ is a tractable standard normal density. As a result, a CNF can be trained by learning a parameterized vector field $u_t^\theta$ that generates the probability path $p_t^\theta$ maximizing the likelihood of the data. During inference, the likelihood of a data point $\boldsymbol{x}_1 \sim q(\boldsymbol{x}_1)$ under the learned model can be computed by integrating Eq. (3) with samples obtained by solving the ODE in Eq. (2) backwards in time.

**Condition-Aware CNFs.** Given a conditioning random variable $Y \in \mathbb{R}^k$, one can extend CNFs to model conditional likelihoods by learning a parameterized family of distributions $p_t^\theta(\boldsymbol{x}_t \mid \boldsymbol{y})$, generated by a conditional vector

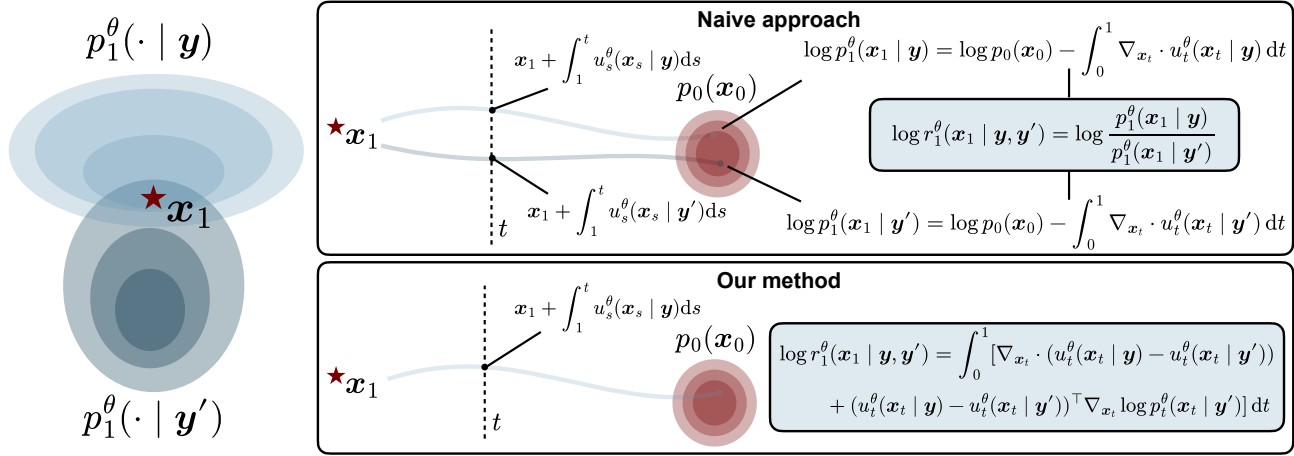

*Figure 1.* Naively approximating the likelihood ratio $q(\boldsymbol{x}_1 \mid \boldsymbol{y})/q(\boldsymbol{x}_1 \mid \boldsymbol{y}')$ requires estimating the likelihood of $\boldsymbol{x}_1$ separately under two conditional generative models $p_t^\theta(\cdot \mid \boldsymbol{y})$ and $p_t^\theta(\cdot \mid \boldsymbol{y}')$, incurring two independent integral evaluations with compounding discretization errors. scRatio instead estimates the log-ratio directly as the solution of a single ODE tracked along trajectories simulated from data to noise under a vector field $b_t$. Here, we illustrate the case where $b_t := u_t^\theta(\cdot \mid \boldsymbol{y})$, i.e., trajectories are simulated under the numerator field.

field $u_t^\theta(\boldsymbol{x}_t \mid \boldsymbol{y})$, for each realization $\boldsymbol{y}$ of $Y$. This enables comparing how likely a data point is under different conditions by evaluating Eq. (3) with distinct conditional fields.

### 3.2. Flow Matching

Flow matching (Lipman et al., 2023; Liu et al., 2022b; Albergo & Vanden-Eijnden, 2023) is a simulation-free approach to optimize CNFs, avoiding computationally expensive operations like Eq. (3). Given real data samples $\boldsymbol{x}_1 \sim q(\boldsymbol{x}_1)$, the model expresses the time-marginal generating field $u_t(\boldsymbol{x}_t)$ presented in Eq. (1) in terms of a data-conditioned field $u_t(\boldsymbol{x}_t \mid \boldsymbol{x}_1)$ as follows:

$$u_t(\boldsymbol{x}_t) := \int u_t(\boldsymbol{x}_t \mid \boldsymbol{x}_1) \frac{p_t(\boldsymbol{x}_t \mid \boldsymbol{x}_1) q(\boldsymbol{x}_1)}{p_t(\boldsymbol{x}_t)} \mathrm{d}\boldsymbol{x}_1 \,.$$

Crucially, regressing $u_t^\theta(\boldsymbol{x}_t)$ against $u_t(\boldsymbol{x}_t \mid \boldsymbol{x}_1)$ is equivalent, up to a constant term, to learning $u_t(\boldsymbol{x}_t)$, which is used for generation from noise according to Eq. (2).

The conditional field $u_t(\boldsymbol{x}_t \mid \boldsymbol{x}_1)$ generates a conditional probability path $p_t(\cdot \mid \boldsymbol{x}_1)$ interpolating between the noise distribution at $t = 0$, $p_0(\cdot \mid \boldsymbol{x}_1) = \mathcal{N}(\boldsymbol{0}_d, \mathbb{I}_d)$, and a distribution at $t = 1$ concentrated around the data point $\boldsymbol{x}_1$, which can be approximated in practice by a tractable distribution $\tilde{q}(\cdot \mid \boldsymbol{x}_1)$ such as a narrow Gaussian or a point mass. Given a closed-form path from noise to data, the target field $u_t(\boldsymbol{x}_t \mid \boldsymbol{x}_1)$ is tractable. Here, we consider Gaussian paths:

$$p_t(\cdot \mid \boldsymbol{x}_1) := \mathcal{N}(\alpha_t \boldsymbol{x}_1, \sigma_t^2 \mathbb{I}_d) \qquad (4)$$

where $\alpha_t$ and $\sigma_t$ define a scheduler for the transition between noise and data. Throughout this work, we fix $\alpha_t = t$ and explore three choices of variance used in the literature: (I) $\sigma_t = 1 - t$ (Tong et al., 2024a), (II)

$\sigma_t = 1 - (1 - \sigma_{\min})t$ (Lipman et al., 2023), and (III) $\sigma_t = \left[(1 - \lambda)t^2 + (\lambda - 2)t + 1\right]^{\frac{1}{2}}$ (Albergo & Vanden-Eijnden, 2023; Tong et al., 2024b), where $\sigma_{\min}$ and $\lambda$ are hyperparameters controlling the magnitude of noise around the data and the shape of the probability path, respectively (see App. B.3 and App. B.4 for more details).

Using Gaussian paths, one can derive a closed-form expression for the conditional field $u_t(\cdot \mid \boldsymbol{x}_1)$ used as a regression target during training (see App. B.1 for a full derivation and Tab. 3 for its explicit form under different schedulers).

### 3.3. Application: scRNA-seq Data

Density estimation via generative models is naturally applicable to single-cell biology, where each observation $\boldsymbol{x}_1 \in \mathbb{R}^d$ is a $d$-dimensional representation of the cellular state, measured under specific experimental conditions. Given a treatment label or biological covariate $\boldsymbol{y}$ associated with each sample, conditional CNFs modeling $\{p_t^\theta(\cdot \mid \boldsymbol{y})\}_{t \in [0,1]}$ can be used to approximate cellular distributions across biological contexts (Klein et al., 2025). Modeling such context-specific densities provides a principled way to assess distributional shifts across experimental conditions, supporting the analysis of perturbation assays.

## 4. Flow-Based Likelihood Ratio Estimation

### 4.1. Problem Statement

Let $\boldsymbol{x}_1 \in \mathbb{R}^d$ be a real data sample and $\boldsymbol{y}, \boldsymbol{y}' \in \mathbb{R}^k$ realizations of a conditioning random variable $Y$ such that $\boldsymbol{y} \neq \boldsymbol{y}'$. Furthermore, consider $\{p_t^\theta(\cdot \mid Y)\}_{t \in [0,1]}$ as a conditional CNF model, such that $p_1^\theta(\cdot \mid Y) \approx q(\cdot \mid Y)$, where $q(\cdot \mid Y)$ is the true conditional data density, and with parameterized

vector fields $\{u_t^\theta(\cdot \mid Y)\}_{t \in [0,1]}$. Our goal is to estimate the ratio between conditional likelihoods, defined as:

$$r^\theta(\boldsymbol{x}_1 \mid \boldsymbol{y}, \boldsymbol{y}') := \frac{p_1^\theta(\boldsymbol{x}_1 \mid \boldsymbol{y})}{p_1^\theta(\boldsymbol{x}_1 \mid \boldsymbol{y}')} \,. \tag{5}$$

In other words, we seek to score each data point based on whether it is more likely under the parameterized distribution conditioned on $\boldsymbol{y}$ or $\boldsymbol{y}'$. While we focus on conditional distributions, the same formulation can be applied to a pair of unconditional generative models on $\mathbb{R}^d$.

A straightforward approach to address this goal is to compute both the numerator and denominator in Eq. (5) by solving the ODE in Eq. (3) using different conditional fields and deriving the ratio between the outcomes. We refer to this practice as the *naive estimation* of $r^\theta(\boldsymbol{x}_1 \mid \boldsymbol{y}, \boldsymbol{y}')$. However, each evaluation of the integral in Eq. (3) is costly, which undermines the efficiency of the naive approach as the number of data points and features increases. Moreover, each separate simulation-based estimation involves non-linear error compounding due to discretization across time.

### 4.2. Simulation-Based Likelihood Ratio Estimation

To address the computational cost and compounding errors of the naive estimation, we propose estimating the likelihood ratio in Eq. (5) via the simulation of a *single ODE*. By eliminating the additional integral evaluations required by the naive method, our approach efficiently scores individual data points while maintaining stable performance. Specifically, we derive a dynamical formulation of the ratio $r_t^\theta$ for $t \in [0,1]$, such that $r_1^\theta = r^\theta$, with $r^\theta$ as defined in Eq. (5).

For CNFs, the vector fields $u_t^\theta(\cdot \mid \boldsymbol{y})$ and $u_t^\theta(\cdot \mid \boldsymbol{y}')$ generate the corresponding conditional densities under the continuity equations in Eq. (1), sharing the same tractable prior:

$$p_0(\cdot \mid \boldsymbol{y}) = p_0(\cdot \mid \boldsymbol{y}') = \mathcal{N}(\boldsymbol{0}_d, \mathbb{I}_d) \,. \tag{6}$$

We then establish the following dynamical formulation of the log-ratio of densities in Eq. (5):

$$\log r_t^\theta(\boldsymbol{x}_t \mid \boldsymbol{y}, \boldsymbol{y}') := \log \frac{p_t^\theta(\boldsymbol{x}_t \mid \boldsymbol{y})}{p_t^\theta(\boldsymbol{x}_t \mid \boldsymbol{y}')} \,, \tag{7}$$

$$\text{with } \log r_0^\theta(\boldsymbol{x}_0 \mid \boldsymbol{y}, \boldsymbol{y}') = 0 \,. \tag{8}$$

We seek to derive an integral equation to estimate $\log r_1^\theta(\boldsymbol{x}_1 \mid \boldsymbol{y}, \boldsymbol{y}')$ akin to the change of variable formula in Eq. (3). However, while each conditional density satisfies a continuity equation, their ratio does not form a proper density function over $\mathbb{R}^d$, and thus cannot be sampled.

Inspired by work on composing diffusion models (Skreta et al., 2025b), we propose tracking the $\log$-ratio between densities in Eq. (7) along interpolation trajectories between the noise and the data under an arbitrary velocity model. We formalize our solution in the following proposition.

**Proposition 4.1.** *Let $\boldsymbol{x}_t$ be the solution of $\mathrm{d}\boldsymbol{x}_t/\mathrm{d}t = b_t(\boldsymbol{x}_t)$, with $t \in [0,1]$. Moreover, let $\{p_t\}_{t \in [0,1]}$ and $\{p_t'\}_{t \in [0,1]}$ be two probability paths generated by vector fields $\{u_t\}_{t \in [0,1]}$ and $\{u_t'\}_{t \in [0,1]}$, each satisfying the continuity equation in Eq. (1), with $p_0 = p_0'$. Assume that $p_t$ and $p_t'$ share the same support for all $t \in [0,1]$ and define the time-dependent density ratio $r_t(\cdot) := p_t(\cdot)/p_t'(\cdot)$. Then the logarithm of the ratio evaluated along the trajectory $\boldsymbol{x}_t$ satisfies:*

$$\frac{\mathrm{d}}{\mathrm{d}t} \log r_t(\boldsymbol{x}_t) = \nabla_{\boldsymbol{x}_t} \cdot \left( u_t'(\boldsymbol{x}_t) - u_t(\boldsymbol{x}_t) \right) \tag{9}$$

$$+ \left( b_t(\boldsymbol{x}_t) - u_t(\boldsymbol{x}_t) \right)^\top \nabla_{\boldsymbol{x}_t} \log p_t(\boldsymbol{x}_t)$$

$$+ \left( u_t'(\boldsymbol{x}_t) - b_t(\boldsymbol{x}_t) \right)^\top \nabla_{\boldsymbol{x}_t} \log p_t'(\boldsymbol{x}_t),$$

*with $\log r_0(\boldsymbol{x}_0) = 0$.*

The derivation is in App. B.5. In summary, the ratio between two likelihoods at a data point $\boldsymbol{x}_1$ can be estimated by integrating Eq. (9) along the trajectory $\boldsymbol{x}_t = \boldsymbol{x}_1 + \int_1^t b_s(\boldsymbol{x}_s)\mathrm{d}s$, where $b_t(\boldsymbol{x}_t)$ is a time-conditional marginal field. Considering our proposed setting with conditional densities approximated by generative models as in Eq. (7), we suggest two natural examples of the simulated field $b_t$:

- **S1**. $b_t$ as the numerator field: With $b_t(\cdot) := u_t(\cdot) := u_t^\theta(\cdot \mid \boldsymbol{y})$ and $u_t'(\cdot) := u_t^\theta(\cdot \mid \boldsymbol{y}')$.
- **S2**. $b_t$ as the unconditional field: With $b_t(\cdot) := u_t^\theta(\cdot)$, $u_t(\cdot) := u_t^\theta(\cdot \mid \boldsymbol{y})$ and $u_t'(\cdot) := u_t^\theta(\cdot \mid \boldsymbol{y}')$.

Note that both cases can be obtained by a single generative model, simultaneously trained conditionally and unconditionally as in Ho & Salimans (2021).

In Fig. 1, we use S1 to visualize how our method compares with its naive counterpart, while we refer to App. B.7 for the explicit integral solution of Eq. (9).

### 4.3. Choice of the Simulation Field $b_t$

When $b_t$ is itself a generative field with associated time-marginal density $p_t^b$, the following bound on the cumulative squared variation in time of the log-ratio holds.

**Proposition 4.2.** *Let $r_t$, $p_t$ and $p_t'$ be defined as in Prop. 4.1, with $p_0 = p_0'$. Moreover, let $b_t$ be a generative field for a distribution $p_t^b$, with $t \in [0,1]$, according to Eq. (1). Define*

$$\Delta\mathrm{KL}(t) := \mathrm{KL}(p_t^b \| p_t') - \mathrm{KL}(p_t^b \| p_t). \tag{10}$$

*Note that $\Delta\mathrm{KL}(0) = 0$, which follows from $p_0 = p_0'$. Under regularity conditions allowing the exchange of derivative and expectation, the cumulative squared rate of change of the log-ratio along trajectories $\boldsymbol{x}_t \sim p_t^b$ satisfies*

$$\int_0^1 \mathbb{E}_{p_t^b} \left[ \left( \frac{\mathrm{d}}{\mathrm{d}t} \log r_t(\boldsymbol{x}_t) \right)^2 \right] \mathrm{d}t \geq \left( \Delta\mathrm{KL}(1) \right)^2, \tag{11}$$

*where $\frac{\mathrm{d}}{\mathrm{d}t} \log r_t(\boldsymbol{x}_t)$ denotes the total time derivative of the log-ratio evaluated along $\boldsymbol{x}_t$.*

The proof is in App. B.9. This result shows that $(\Delta\mathrm{KL}(1))^2$ lower bounds the cumulative squared variation of the log-ratio along trajectories sampled from $p_t^b$. Since $\Delta\mathrm{KL}(1)$ depends only on the terminal distribution $p_1^b$, minimizing this bound suggests choosing $b_t$ such that $p_1^b$ concentrates in regions of high probability mass under both $p_1$ and $p_1'$. This overlap region can be approximated from data samples.

## 4.4. Score Estimation in Eq. (9)

From the Gaussian probability path formulation in Eq. (4), the density score $\nabla_{\boldsymbol{x}_t} \log p_t(\boldsymbol{x}_t)$ in Eq. (9) can be directly reparametrized from the generating vector fields $u_t(\boldsymbol{x}_t)$ according to the following equation (Zheng et al., 2023):

$$\nabla_{\boldsymbol{x}_t} \log p_t(\boldsymbol{x}_t) = \frac{\alpha_t u_t(\boldsymbol{x}_t) - \dot{\alpha}_t \boldsymbol{x}_t}{\sigma_t(\dot{\alpha}_t \sigma_t - \alpha_t \dot{\sigma}_t)} . \tag{12}$$

Thus, learning either the score or the vector field is *equivalent* up to a reparameterization depending on the schedule. In our conditional setting, Eq. (12) allows us to recover $\nabla_{\boldsymbol{x}_t} \log p_t^\theta(\boldsymbol{x}_t \mid \boldsymbol{y}')$ from $u_t^\theta(\boldsymbol{x}_t \mid \boldsymbol{y}')$, enabling direct evaluation of the density-ratio ODE.

However, for many choices of $\alpha_t$ and $\sigma_t$, the denominator in Eq. (12) implies a division by a low number at $t = 1$ (see Tab. 4), causing numerical instability close to the data.

Thus, we choose to parameterize the score with a neural network $s_t^\psi(\boldsymbol{x}_t \mid \boldsymbol{y}') \approx \nabla_{\boldsymbol{x}_t} \log p_t(\boldsymbol{x}_t \mid \boldsymbol{y}')$ trained to regress the conditional score, which is tractably available for Gaussian probability paths (see Tab. 3 for the closed form conditional score objectives and App. B.3 for theoretical details). This provides a smooth estimator of the score close to the data, enabling a stable approximation of the log-density ratio in Prop. 4.1. We provide a first-order error analysis of our neural approximation in App. B.8.

## 4.5. Use Cases in Single-cell Data Analysis

Many tasks in computational biology require comparing the likelihood of cellular profiles under alternative probabilistic models. In what follows, we formalize several examples of scRatio use in cellular data analysis.

**Differential Abundance (DA) analysis.** Consider a dataset $\boldsymbol{X} = \{(\boldsymbol{x}_1^{(i)}, y^{(i)})\}_{i=1}^N$ of $N$ single cells, where $\boldsymbol{x}_1^{(i)} \in \mathbb{R}^d$ denotes a representation of the gene expression profile of cell $i$, and $y^{(i)} \in 0, 1$ is a perturbation label, with 0 and 1 indicating control and treatment conditions, respectively. Moreover, let $p_t^\theta(\cdot \mid Y)_{t \in [0,1]}$ be a generative model approximating the distribution of cells under each realization of conditioning random variable $Y$, and $u_t^\theta(\cdot \mid Y)_{t \in [0,1]}$ its generating velocity field. With scRatio, we tackle a core challenge in single-cell analysis:

*Are cells from a given perturbation more likely under their conditional distribution than under the control distribution?*

Given a perturbed cell $\boldsymbol{x}_1^{(j)}$, with $j \in \mathcal{I}_1 = \{i \mid y^{(i)} = 1\}$, we conduct this analysis by estimating the following ratio:

$$\log r_1^\theta(\boldsymbol{x}_1^{(j)} | 0, 1) = \log \frac{p_1^\theta(\boldsymbol{x}_1^{(j)} | Y = 1)}{p_1^\theta(\boldsymbol{x}_1^{(j)} | Y = 0)} .$$

using Prop. 4.1. Intuitively, when the $\log$ ratio is higher than 0, the cell is more likely under the perturbed distribution than the untreated one. The concept can be naturally extended to settings involving multiple perturbations by training a flow model conditioned on different treatment labels and contrasting each perturbation likelihood with the control likelihood, separately.

**Combining conditions.** Let $\boldsymbol{X} = (\boldsymbol{x}_1^{(i)}, y^{(i)}, z^{(i)})_{i=1}^N$ now be a dataset consisting of the combination of two conditions: $y^{(i)} \in 1, \ldots, k_y$ and $z^{(i)} \in 1, \ldots, k_z, \forall i \in 1, \ldots, N$. In an applied setting, the two conditions can represent combinations of treatments or batch and biological labels. With scRatio, we ask the following:

*Does conditioning on both $y$ and $z$ increase the likelihood of $\boldsymbol{x}_1$ compared to conditioning on $y$ only?*

For cells from a certain combination of conditions $y$ and $z$, we can evaluate the following likelihood ratio:

$$\log r_1^\theta(\boldsymbol{x}_1^{(i)} \mid (y, z), y) = \log \frac{p_1^\theta(\boldsymbol{x}_1^{(i)} \mid y, z)}{p_1^\theta(\boldsymbol{x}_1^{(i)} \mid y)}$$

Importantly, this does not require training separate models for numerator and denominator, as flow matching allows optimizing simultaneously single- and double-conditional models by alternating $z$ with an empty token $\varnothing$ during training (Ho & Salimans, 2021).

A potential use case is treatment combination, where one assesses whether multiple perturbations cause additional state shifts ($\log r_1^\theta > 0$) due to the presence of interaction effects compared to single treatments. Another possible application of this formulation is batch-correction evaluation. If $y$ is a biological annotation and $z$ is a batch label, the ratio indicates the extent to which technical effects drive the variation in the data ($\log r_t^\theta \approx 0$ for mixed data and $\log r_t^\theta \neq 0$ with batch effect). This information can be used to compare different correction procedures depending on their effect on the ratio's magnitude.

# 5. Experiments

## 5.1. Simulated High-Dimensional Gaussians

**Task and Dataset.** First, we follow Yu et al. (2025) and evaluate scRatio on estimating closed-form log-likelihood

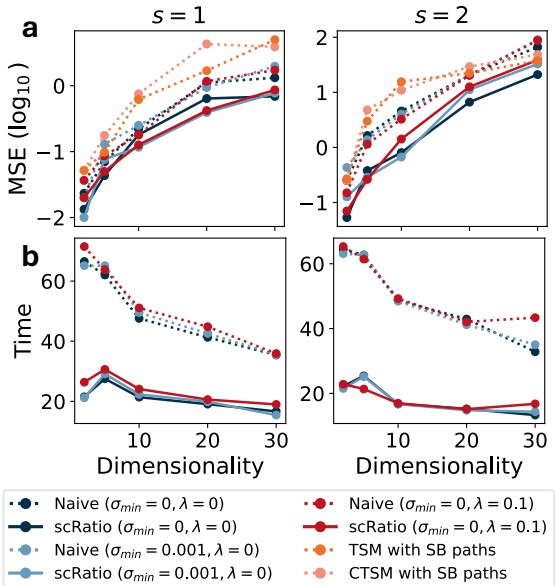

*Figure 2.* **a.** The average MSE across 5 training runs between true and estimated likelihood ratios across models and simulation parameters. SB paths stand for Schrödinger Bridge probability paths. **b.** Runtime in seconds for likelihood ratio estimation comparing scRatio with the naive approach across simulation parameters. Dynamical ratios are simulated with the adaptive `dopri5` solver.

ratios on multivariate Gaussian distributions. Consider the following tractable distributions:

$$q := \mathcal{N}(s\mathbb{1}_d, \mathbb{I}_d) , \quad q' := \mathcal{N}(\mathbf{0}_d, \mathbb{I}_d) ,$$

where $\mathbb{1}_d$ is the one-vector of $d$-dimensions and $s \in \{1, 2\}$ a scalar. The aim is to estimate the true ratio $r := \log \frac{q}{q'}$ using samples from the two distributions. For each value of $s$, we consider the performance across several dimensions $d \in \{2, 5, 10, 20, 30\}$.

**Baselines.** For this experiment, we compare scRatio with three baselines (see App. D.2 for a detailed discussion).

- Naive approach: Instead of approximating $r$ according to Prop. 4.1, we estimate the numerator and denominator likelihoods using Eq. (3) and compute their $\log$-ratio.
- Time Score Matching (TSM): Given an interpolant $p_t$, for $t \in [0, 1]$, between two densities $p_0$ and $p_1$, TSM expresses the ratio between $p_1$ and $p_0$ as the time integral of the score $\frac{\partial}{\partial t} \log p_t(\boldsymbol{x})$. We follow Choi et al. (2022)'s parameterization, which bypasses regressing an intractable time score via integration by parts.
- Conditional TSM (CTSM): Using elements of flow matching to define $p_t$ (see Sec. 3.2), Yu et al. (2025) derive a tractable and more efficient formulation of the time score conditioned on samples from $p_0$ and $p_1$.

To compare with our method in modeling ratios between arbitrary data distributions, we consider variants of TSM and

*Table 1.* MAE ($\downarrow$) between true and predicted MI across data dimensions. Results are averaged over three training runs and reported as mean ± standard error. The best and second-best results per dimension are highlighted in bold and underlined, respectively.

| Model | $d$ | | | | |
| | 20 | 40 | 80 | 160 | 320 |
|---|---|---|---|---|---|
| Classifier baseline | $0.27 \pm 0.04$ | $9.99 \pm 0.00$ | $19.51 \pm 0.18$ | $0.83 \pm 0.11$ | $26.68 \pm 3.70$ |
| TSM with SB paths | $\mathbf{0.03 \pm 0.01}$ | $0.38 \pm 0.02$ | $0.25 \pm 0.10$ | $0.89 \pm 0.11$ | $3.55 \pm 0.12$ |
| CTSM with SB paths | $0.06 \pm 0.01$ | $0.09 \pm 0.04$ | $0.23 \pm 0.03$ | $0.87 \pm 0.10$ | $\underline{2.15 \pm 0.08}$ |
| scRatio $\sigma_{\min} = 0, \lambda = 0$ | $0.05 \pm 0.01$ | $\underline{0.05 \pm 0.02}$ | $\underline{0.05 \pm 0.03}$ | $0.16 \pm 0.05$ | $21.18 \pm 0.14$ |
| scRatio $\sigma_{\min} = 0.1, \lambda = 0$ | $\underline{0.04 \pm 0.02}$ | $\mathbf{0.03 \pm 0.02}$ | $\mathbf{0.02 \pm 0.01}$ | $\underline{0.16 \pm 0.07}$ | $21.00 \pm 0.26$ |
| scRatio $\sigma_{\min} = 0, \lambda = 0.25$ | $\mathbf{0.03 \pm 0.01}$ | $0.09 \pm 0.03$ | $0.07 \pm 0.04$ | $\mathbf{0.11 \pm 0.03}$ | $\mathbf{1.16 \pm 0.27}$ |

CSTM based on Schrödinger Bridge (SB) interpolants between $p_0$ and $p_1$. This formulation relies solely on samples from the two distributions without any distributional assumptions (see App. D.2 for further elaboration). We tuned the baselines in terms of learning rate and SB variance.

**Metrics and Results.** In Fig. 2, we report the Mean Squared Error (MSE) between the true and the estimated $\log$-ratios. For scRatio, we display the three best models for each of the three schedules described in Sec. 3.2 and App. B.4. Notably, scRatio outperforms the baselines for the considered values of $s$ and $d$, while no significant performance difference is apparent across schedules. Moreover, Fig. 2b shows that our approach achieves a significantly lower runtime for estimating the $\log$-ratio compared to the counterpart naive solution. The decrease in runtime with the dimensionality is likely a byproduct of combining adaptive solvers with smoother high-dimensional vector fields. Our results suggest a simultaneous gain in both performance and efficiency. Additional analyses and discussions are available in App. F.2 to F.7.

### 5.2. Mutual Information Estimation

**Task and Dataset.** We replicate the synthetic experiments in Yu et al. (2025) for estimating the Mutual Information (MI) between structured Gaussians. Consider the distributions

$$q = \mathcal{N}(\mathbf{0}_d, \mathbb{I}_d) , \quad q' = \mathcal{N}(\mathbf{0}_d, \boldsymbol{\Sigma}) ,$$

with $d$ being an even number, and where $\boldsymbol{\Sigma}$ is a block-diagonal matrix consisting of $2 \times 2$ blocks with 1 on the diagonal and 0.8 off-diagonal. Let $\boldsymbol{x} \sim q_0$ and split it into $\boldsymbol{x}^{\mathrm{odd}} = (x^1, x^3, ..., x^{d-1})$ and $\boldsymbol{x}^{\mathrm{even}} = (x^2, x^4, ..., x^d)$, one can show that the MI between $\boldsymbol{x}^{\mathrm{odd}}$ and $\boldsymbol{x}^{\mathrm{even}}$ is available in closed form as the following expectation:

$$I(\boldsymbol{x}^{\mathrm{even}}, \boldsymbol{x}^{\mathrm{odd}}) = \mathbb{E}_{q(\boldsymbol{x})} \left[ \log \frac{q(\boldsymbol{x})}{q'(\boldsymbol{x})} \right] \quad (13)$$

(see App. D.3 for more details). Following prior work, we consider how MI estimation scales across dimensions for $d \in \{20, 40, 80, 160, 320\}$.

**Baselines.** We train scRatio under different scheduling strategies to approximate Eq. (13) as a likelihood ratio, using

*Table 2.* Comparison of scRatio with three distinct scheduling settings and baseline methods across increasing levels of DA controlled by the parameter $a$ (Eq. 14). We report (I) the Spearman correlation $\rho(\cdot, a)$ between each metric and the ground-truth DA level $a$, and (II) the average metric value over high-DA regimes ($a \geq 0.3$). Results are averaged over three independent training runs and reported as mean $\pm$ standard error. In bold and underlined are the best and second-best results per metric, respectively.

| Model | $\rho(\text{AUC}, a)$ (↑) | AUC ($a \geq 0.3$) (↑) | $\rho(\text{NAR}, a)$ (↑) | NAR ($a \geq 0.3$) (↑) | $\rho(\text{CSP}, a)$ (↑) | CSP ($a \geq 0.3$) (↑) |
|---|---|---|---|---|---|---|
| MrVI | $0.71_{\pm 0.09}$ | $0.89_{\pm 0.00}$ | $0.83_{\pm 0.01}$ | $7.33_{\pm 0.27}$ | $0.67_{\pm 0.06}$ | $0.97_{\pm 0.00}$ |
| MELD | $0.48_{\pm 0.00}$ | $0.93_{\pm 0.00}$ | $0.17_{\pm 0.00}$ | $6.45_{\pm 0.00}$ | $0.27_{\pm 0.00}$ | $\mathbf{0.99}_{\pm 0.00}$ |
| scRatio ($\sigma_{\min} = 0, \lambda = 0$) | $\mathbf{1.00}_{\pm 0.00}$ | $\underline{0.95}_{\pm 0.00}$ | $\mathbf{1.00}_{\pm 0.00}$ | $\underline{9.64}_{\pm 0.45}$ | $\underline{0.95}_{\pm 0.01}$ | $0.98_{\pm 0.00}$ |
| scRatio ($\sigma_{\min} = 0.1, \lambda = 0$) | $\underline{0.99}_{\pm 0.01}$ | $\mathbf{0.96}_{\pm 0.00}$ | $\mathbf{1.00}_{\pm 0.00}$ | $\mathbf{11.39}_{\pm 0.30}$ | $\underline{0.95}_{\pm 0.03}$ | $0.98_{\pm 0.00}$ |
| scRatio ($\sigma_{\min} = 0, \lambda = 0.01$) | $\underline{0.99}_{\pm 0.01}$ | $\underline{0.95}_{\pm 0.00}$ | $\underline{0.98}_{\pm 0.02}$ | $9.39_{\pm 0.19}$ | $\mathbf{0.98}_{\pm 0.01}$ | $0.98_{\pm 0.00}$ |

100,000 samples from each of the two distributions. The MI is estimated by averaging the learned ratios evaluated on samples from $q$. As baselines, we consider the CTSM and TSM models with SB paths as described in Sec. 5.1 as well as a simple classifier model (App. D.3). Similar to the previous experiment, we adopt CTSM and TSM with SB paths to enable a fair comparison with scRatio in a purely sample-based setting without distributional assumptions.

**Metrics and Results.** As the MI is available for different dimensions (see App. D.3), we report the Mean Absolute Error (MAE) between the predicted and true value. Tab. 1 shows that different variants of scRatio overcome the baselines on four out of five dimensions and tie as best-performing on the remaining one. Crucially, adding noise to the flow matching probability path ($\lambda = 0.25$) yields significantly more precise performance in higher dimensions.

### 5.3. Single-cell Differential Abundance Estimation

**Task.** The most common application of likelihood ratio estimation on cellular data is DA analysis, as described in Sec. 4.5. Here, positive and negative likelihood ratios at individual observations indicate that treatment or control cells, respectively, are overrepresented in the region of gene-expression space corresponding to a cellular state.

**Dataset.** To compare scRatio with existing methods, we design a semi-synthetic dataset with distinct ground truth levels of treatment and control overrepresentation, similar to Boyeau et al. (2025). We use a dataset of 68k Peripheral Blood Mononuclear Cells (PBMC) from Zheng et al. (2017), which we divide into four clusters, with labels $c_i \in \{1, 2, 3, 4\}$ for each cell $i$ (see Fig. 6a). Then, we assign to each cell a control ($y_i = 0$) or treatment ($y_i = 1$) binary label with a cluster-specific probability $\Pi = \{\pi^k\}_{k=1}^4$, such that $y_i \sim \text{Ber}(\pi = \pi^{c_i})$. To test the model at different levels of DA, we create 8 datasets with cluster proportions

$$\Pi^a = \{0.5, 0.5 + a, 0.5 - a, 0.5\}, \qquad (14)$$

for $a \in \{0, 0.05, 0.1, 0.2, 0.3, 0.4, 0.45, 0.5\}$. Thus, in clusters 1 and 4, cases and controls are always equally represented (ratio equals zero), while clusters 2 and 3 have a higher proportion of either treated or untreated cells based on $a$, introducing different levels of DA (see Fig. 6b-c).

**Baselines.** We consider the following baselines:

- MrVI (Boyeau et al., 2025): A disentangled Variational Autoencoder (VAE) model relying on variational posterior approximation for the DA estimation.
- MELD (Burkhardt et al., 2021): A kNN approach based on label smoothing over local neighborhoods.

We do not compare scRatio with Milo (Dann et al., 2022), despite their relatedness in a broader scope, because, unlike methods that score single cells, it predicts DA at the level of cellular neighborhoods.

Similar to scRatio, all models produce an estimate of the true log-likelihood ratio $\log r(\boldsymbol{x} \mid 1, 0) = \log \frac{q(\boldsymbol{x}|1)}{q(\boldsymbol{x}|0)}$ applied to a cell $\boldsymbol{x}$. Positive and negative values indicate higher treatment and control likelihoods, respectively.

**Metrics.** As we do not have a ground-truth ratio per cell, we develop metrics that assess whether the distinct models predict likelihood ratios that match the DA in the dataset.

- AUC: The **A**rea **U**nder the precision-recall **C**urve (AUC) between the absolute values of the estimated ratios and the vector of DA binary labels, which we set to 1 for clusters 2 and 3 and 0 for clusters 1 and 4.
- Normalized Abundance Ratio (NAR): How much larger the absolute ratios are in DA clusters (2 and 3) versus non-DA clusters (1 and 4).
- Correct Sign Proportion (CSP): Proportion of ratios that agree in sign with the real log-odds of DA clusters (positive for cluster 2 and negative for cluster 3).

All of the above metrics are expected to increase with the amount of DA in clusters 2 and 3. Thus, we report their Spearman correlation with $a$ (see Eq. (14)), together with their average values for $a \geq 0.3$ (high DA). For scRatio, we choose the best results for different schedulers optimized over the mean $\text{AUC}(a \geq 0.3)$ metric.

**Results.** In Tab. 2, our model, especially with deterministic paths and $\sigma_{\min} = 0.1$, achieves the best performance across all the metrics, except for $\text{CSP}(a \geq 0.3)$, where MELD marginally overcomes it. We expand on the advantages of using scRatio against competing methods in App. D.4.

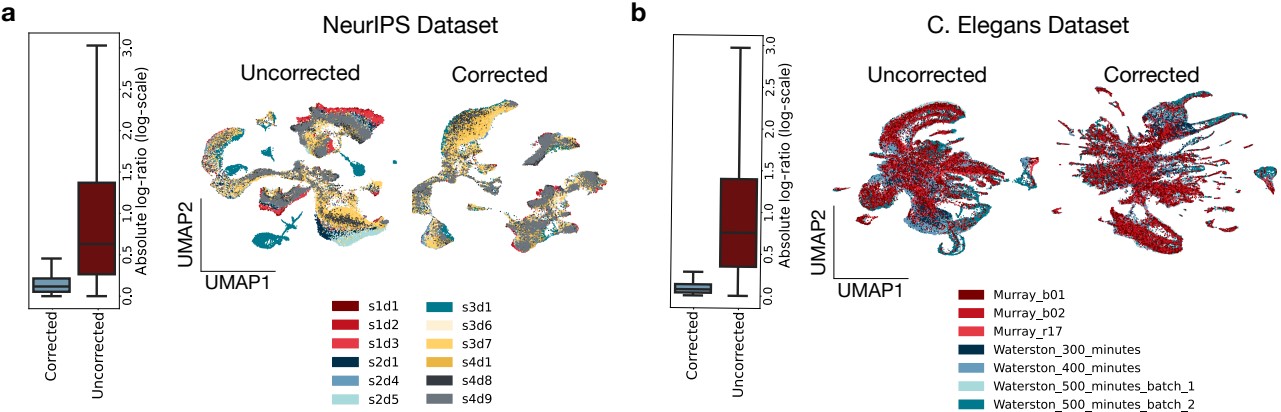

*Figure 3.* Ratio-based batch correction evaluation on the NeurIPS 2021 (panel **a**) and C. Elegans (panel **b**) datasets. Left: Distribution of the absolute estimated log-ratio before and after batch correction for each of the measured batch and cell type combinations in the two datasets, estimated by training scRatio alternatively on the original and batch-corrected representations. Right: UMAP plots calculated before and after batch correction with scVI. To compute the UMAP, we use 50-dimensional representations obtained by PCA and scVI for uncorrected and corrected data, respectively. Batch names are annotated below each plot.

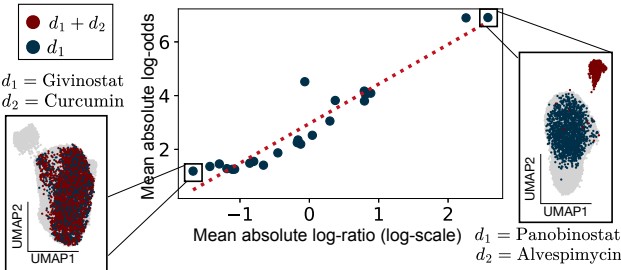

*Figure 4.* On the x-axis, the mean absolute log-likelihood ratio $\log r^\theta(\boldsymbol{x} \mid (d_1, d_2), d_1)$ evaluated on cells perturbed with both $(d_1, d_2)$ and only $d_1$. On the y-axis, the mean log-odds of a classifier trained to discriminate cells treated with $(d_1, d_2)$ from cells treated with $d_1$ only. Each point is a combination. UMAP plots show extreme cases of the combinatorial effect and lack thereof.

## 5.4. Batch Correction Evaluation

**Task.** As a case study, we use scRatio to qualitatively assess the effectiveness of batch correction, a process that removes unwanted sources of variation from the data (technical effects) while preserving meaningful signals (biological effects). Let $\boldsymbol{X} := \{(\boldsymbol{x}^{(i)}, y^{(i)}, b^{(i)})\}_{i=1}^N$ denote the set of $N$ collected measurements, where $y^{(i)}$ and $b^{(i)}$ are, respectively, cell type and batch labels. Moreover, let $\hat{\boldsymbol{x}} := f(\boldsymbol{x}^{(i)}, y^{(i)}, b^{(i)})$ be the batch-corrected representation of sample $\boldsymbol{x}^{(i)}$, where $f(\boldsymbol{x}^{(i)}, y^{(i)}, b^{(i)})$ is a correction function. We train our model to approximate the true ratio $\frac{q(\cdot|y^{(i)}, b^{(i)})}{q(\cdot|y^{(i)})}$ first on samples $\boldsymbol{x}^{(i)}$ and then on their corrected counterparts $\hat{\boldsymbol{x}}^{(i)}$. We use the change in the estimated ratios as a proxy for correction quality. Specifically, we expect a decrease in the absolute value of the log-likelihood ratio upon successful correction, as conditioning on $b^{(i)}$ does not carry additional information about the samples (App. D.5).

**Datasets and Results.** We consider the following datasets: (I) The NeurIPS 2021 dataset (Luecken et al., 2021), consisting of 90,261 bone marrow cells from 12 healthy human donors, and (II) the C. Elegans dataset (Packer et al., 2019), which comprises 89,701 cells across 7 distinct experimental sources. For both datasets, we regressed out the distributional shift induced by the batch label using scVI (Lopez et al., 2018). In Fig. 3a-b, we confirm our hypothesis by observing a decrease in log-ratio magnitude upon batch correction, suggesting that scRatio could complement existing batch correction evaluations (Luecken et al., 2022) with a principled likelihood-based approach (see Fig. 14).

## 5.5. Drug Combination Effect Estimation

**Task and Dataset.** Assessing treatment combinations is a key challenge in computational biology. In this experiment, we use likelihood ratios to detect interaction effects from combinatorial perturbations. For this, we analyze the ComboSciPlex dataset (Srivatsan et al., 2020), which contains 63,378 cells treated with either a single drug $d_1$ or a combination $(d_1, d_2)$. We quantify the difference between cells treated with $(d_1, d_2)$ versus $d_1$ alone by approximating $\log \frac{q(\boldsymbol{x}|d_1, d_2)}{q(\boldsymbol{x}|d_1)}$ using scRatio. Significant interactions correspond to larger absolute values of the log-ratio, suggesting that single and double treatment populations do not mix.

**Metrics.** In the absence of ground truth, we evaluate the plausibility of the estimated likelihood ratios through their correlation with classifier performance. For each pair of single $d_1$ and combined $(d_1, d_2)$ treatments, we train a binary classifier inputing cell states to distinguish between the two conditions, and use the resulting absolute log-odds as a measure of distributional separation. We also estimate absolute log-ratios between the corresponding conditional

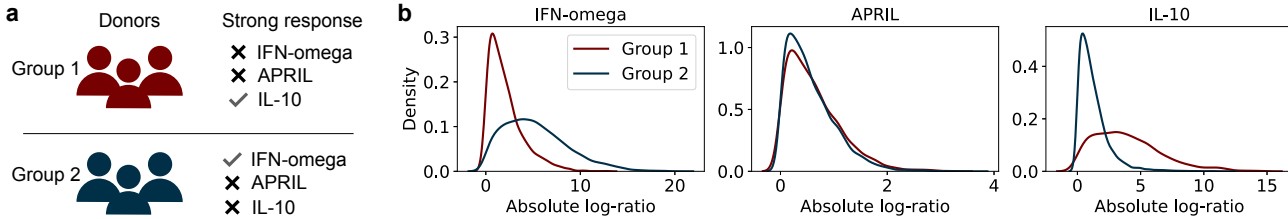

*Figure 5.* Patient-specific differential response to perturbation. (**a**) Depiction of the experimental setup. Following Oesinghaus et al. (2025), we divide donors into two groups responding differently to distinct cytokine treatments. (**b**) Absolute log-likelihood ratios between treated and control distributions evaluated on perturbed cells from different donor groups. A higher ratio indicates evidence of a strong response to a cytokine by a donor group, while log-ratios around 0 refer to a lower shift from controls.

likelihoods evaluated on cells treated with $(d_1, d_2)$ and $d_1$ only (see App. D.6). Good calibration between the quantities indicates that our model pinpoints combinatorial effects.

Note that while a classifier is useful to rank perturbations, it does not directly quantify DA, and in Fig. 8 we show that its likelihood ratio prediction poorly scales with dimensionality, as the performance saturates and fails to quantify likelihood shifts. We use it to validate that our average ratios capture overall responses while preserving a likelihood formulation.

**Results.** In Fig. 4, mean absolute log-odds and log-likelihood ratios show positive correlation. Low values of both correspond to cases where adding $d_2$ does not induce a state shift relative to a single treatment, whereas high values indicate a differential response between single perturbations and their combinations (UMAP plots in Fig. 4). Additional qualitative validation is available in App. F.9.

### 5.6. Patient-Specific Treatment Response Estimation

**Task and Dataset.** Next, we assess whether the log-likelihood ratios estimated by our model can be used to analyze the differential response of patients to treatment. We use a dataset of approximately 10 million PBMC cells from 12 donors (Oesinghaus et al., 2025). For each sample, cells are measured in their control state and perturbed by 90 different cytokines. Let $\boldsymbol{X} := \{(\boldsymbol{x}^{(i)}, s^{(i)}, d^{(i)})\}_{i=1}^N$ be a set of $N$ cells, where $s^{(i)}$ and $d^{(i)}$ are sample and cytokine treatment labels. Here, $d \in \{d^c\} \cup \{d^k\}$, where $d^c$ is a control label and $k \in \{1, ..., n_k\}$ indicates one of the $n_k$ available cytokines. For a cell $\boldsymbol{x}$ from donor $s$ and a treatment $d^k$, we use scRatio to estimate $\log r(\boldsymbol{x} \mid (s, d^k), (s, d^c)) = \log \frac{q(\boldsymbol{x}|s, d^k)}{q(\boldsymbol{x}|s, d^c)}$. Positive average scores correspond to donor-treatment pairs showing a statistical difference between cases and controls.

**Evaluation.** In Oesinghaus et al. (2025), the authors identify cytokine treatments inducing a differential response between two groups of donors. In Fig. 5a, we illustrate our setup. We select two treatments that induce a strong shift from controls in one of the two groups (IL-10 for group 1 and IFN-omega for group 2), and one that exhibits a similar weak response

in both groups (APRIL). For perturbed cells from different donors, we evaluate the log-likelihood ratio between treated cells and controls. In this setting, we qualitatively evaluate our results, inspecting whether the distribution of scores reflects the reported biological differences between donors.

**Results.** Fig. 5b shows that likelihood ratios estimated by our models reflect biological trends reported in the dataset. Higher statistical differences between treated and controls for IFN-omega and IL-10 lead to higher absolute ratios in patient groups 2 and 1, respectively, when compared to the counterpart group. Moreover, the distributions of log-ratios similarly center around zero for both groups for cytokine APRIL, confirming expectations. Thus, scRatio provides a tool for patient stratification and donor-based analysis in large perturbation cohorts.

## 6. Discussion

**Summary.** We introduce scRatio, a method for estimating likelihood ratios between pairs of intractable distributions by explicitly modeling the dynamics of the log-density ratio along sampling trajectories. We derive an ODE that governs the evolution of the log-ratio and approximate it using conditional flow-based generative models. By composing learned velocity fields and scores from multiple conditionings, scRatio tracks likelihood ratios directly along a single simulation path, avoiding redundant ODE solves and reducing the discretization error that accumulates non-linearly during integration, in turn enhancing estimation quality. We demonstrate that this formulation captures complex statistical dependencies across both synthetic benchmarks and real-world single-cell applications.

**Limitations.** When comparing distributions with limited or no overlap in support, the log-likelihood ratios are evaluated along vector fields that are non-generative for at least one of the densities in the ratio. This can lead to numerical instabilities and lower simulation quality, particularly in high-dimensional settings (see C.4 for possible failure detection methods). The method also incurs the overhead of training separate networks for the score and velocity.

## Impact Statement

The presented work addresses a fundamental problem in probabilistic modeling for single-cell data by enabling efficient likelihood-based comparisons between complex conditional data distributions. By providing a principled approach to density ratio estimation, scRatio supports key biological analyses such as treatment effect estimation and evaluation of batch effects. We envision releasing scRatio as a user-friendly, open-source tool to facilitate its adoption in single-cell research. As scRatio operates on biological data, it may be applied in sensitive settings involving clinical information and patient-derived datasets.

## Acknowledgments

A.P. is supported by the Helmholtz Association under the joint research school Munich School for Data Science (MUDS). A.P. and F.J.T. acknowledge support from the European Union (ERC PoC CellCourier, grant number 101248740 and ERC DeepCell, grant number 101054957). Additionally, this work was supported by the Helmholtz Association within the framework of the Helmholtz Foundation Model Initiative (VirtualCell) and Networking Fund (CausalCellDynamics, grant # Interlabs-0029).

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

## A. Code Availability

We made our code available at scRatio repo. All datasets are public and extracted from the cited publications.

## B. Theoretical Supplement

### B.1. Derivation of the Conditional Vector Field

Consider the following conditional Gaussian probability path from Lipman et al. (2023):

$$\boldsymbol{x}_t \sim p_t(\cdot \mid \boldsymbol{x}_1) := \mathcal{N}(\alpha_t \boldsymbol{x}_1, \sigma_t^2 \mathbb{I}_d), \tag{15}$$

where $\boldsymbol{x}_1 \sim q(\boldsymbol{x}_1)$ is a data point from the data distribution $q$, $t \in [0, 1]$ and $(\alpha_t, \sigma_t)$ is a time-dependent scheduler. In the simplest setting, $\alpha_0 = \sigma_1 = 0$ and $\alpha_1 = \sigma_0 = 1$. Nevertheless, we also consider the option of adding noise around the data by setting $\sigma_1 = \sigma_{\min}$, where $\sigma_{\min}$ is a small constant (see App. B.4).

Given the tractable formulation of conditional paths, one can draw a sample from $p_t(\cdot \mid \boldsymbol{x}_1)$ in closed form:

$$\boldsymbol{x}_t = \alpha_t \boldsymbol{x}_1 + \sigma_t \epsilon, \quad \epsilon \sim \mathcal{N}(\mathbf{0}_d, \mathbb{I}_d).$$

As the path from the realization of a noise variable $\epsilon$ and data is tractable, its time derivative can be written as:

$$u_t(\boldsymbol{x}_t \mid \boldsymbol{x}_1) = \dot{\alpha}_t \boldsymbol{x}_1 + \dot{\sigma}_t \epsilon$$
$$= \dot{\alpha}_t \boldsymbol{x}_1 + \frac{\dot{\sigma}_t}{\sigma_t}(\boldsymbol{x}_t - \alpha_t \boldsymbol{x}_1), \tag{16}$$

where we used $\epsilon = \frac{\boldsymbol{x}_t - \alpha_t \boldsymbol{x}_1}{\sigma_t}$.

### B.2. Relationship Between Score and Vector Field

The score of the logarithm of the conditional Gaussian paths in Eq. (15) is:

$$\nabla_{\boldsymbol{x}_t} \log p_t(\boldsymbol{x}_t \mid \boldsymbol{x}_1) = -\frac{\boldsymbol{x}_t - \alpha_t \boldsymbol{x}_1}{\sigma_t^2}. \tag{17}$$

Following Zheng et al. (2023), we isolate $\boldsymbol{x}_1$ and plug the result into Eq. (16). Thus, we get the following relationship:

$$u_t(\boldsymbol{x}_t \mid \boldsymbol{x}_1) = \frac{\dot{\alpha}_t}{\alpha_t} \boldsymbol{x}_t + \frac{\sigma_t}{\alpha_t}(\dot{\alpha}_t \sigma_t - \alpha_t \dot{\sigma}_t) \nabla_{\boldsymbol{x}_t} \log p_t(\boldsymbol{x}_t \mid \boldsymbol{x}_1). \tag{18}$$

The relationship between the conditional score and vector field holds for the marginals as well. Setting $a_t := \frac{\dot{\alpha}_t}{\alpha_t}$ and $b_t := \frac{\sigma_t}{\alpha_t}(\dot{\alpha}_t \sigma_t - \alpha_t \dot{\sigma}_t)$, and using the following results from Zheng et al. (2023)

$$\nabla_{\boldsymbol{x}_t} \log p_t(\boldsymbol{x}_t) = \int \nabla_{\boldsymbol{x}_t} \log p_t(\boldsymbol{x}_t \mid \boldsymbol{x}_1) \frac{p_t(\boldsymbol{x}_t \mid \boldsymbol{x}_1) q(\boldsymbol{x}_1)}{p_t(\boldsymbol{x}_t)} \mathrm{d}\boldsymbol{x}_1$$
$$= \int \nabla_{\boldsymbol{x}_t} \log p_t(\boldsymbol{x}_t \mid \boldsymbol{x}_1) p_t(\boldsymbol{x}_1 \mid \boldsymbol{x}_t) \mathrm{d}\boldsymbol{x}_1, \tag{19}$$

one can show that

$$u_t(\boldsymbol{x}_t) = \int u_t(\boldsymbol{x}_t \mid \boldsymbol{x}_1) p_t(\boldsymbol{x}_1 \mid \boldsymbol{x}_t) \mathrm{d}\boldsymbol{x}_1 \tag{20}$$

$$= \int [a_t \boldsymbol{x}_t p_t(\boldsymbol{x}_1 \mid \boldsymbol{x}_t) + b_t \nabla_{\boldsymbol{x}_t} \log p_t(\boldsymbol{x}_t \mid \boldsymbol{x}_1) p_t(\boldsymbol{x}_1 \mid \boldsymbol{x}_t)] \mathrm{d}\boldsymbol{x}_1 \tag{21}$$

$$= a_t \boldsymbol{x}_t \int p_t(\boldsymbol{x}_1 \mid \boldsymbol{x}_t) \mathrm{d}\boldsymbol{x}_1 + b_t \int \nabla_{\boldsymbol{x}_t} \log p_t(\boldsymbol{x}_t \mid \boldsymbol{x}_1) p_t(\boldsymbol{x}_1 \mid \boldsymbol{x}_t) \mathrm{d}\boldsymbol{x}_1 \tag{22}$$

$$= a_t \boldsymbol{x}_t + b_t \nabla_{\boldsymbol{x}_t} \log p_t(\boldsymbol{x}_t). \tag{23}$$

*Table 3.* Closed-form conditional vector field $u_t(\boldsymbol{x}_t \mid \boldsymbol{x}_1)$ and score $\nabla_{\boldsymbol{x}_t} \log p_t(\boldsymbol{x}_t \mid \boldsymbol{x}_1)$ for different conditional probability path schemes. The tuple $(\alpha_t, \sigma_t)$ represents the schedule of the path between noise and data. A more detailed description of the schedules is in App. B.4.

| $\alpha_t$ | $\sigma_t$ | $u_t(\boldsymbol{x}_t \mid \boldsymbol{x}_1)$ | $\nabla_{\boldsymbol{x}_t} \log p_t(\boldsymbol{x}_t \mid \boldsymbol{x}_1)$ |
|---|---|---|---|
| $t$ | $1-t$ | $(\boldsymbol{x}_1 - \boldsymbol{x}_t)/(1-t)$ | $(t\boldsymbol{x}_1 - \boldsymbol{x}_t)/(1-t)^2$ |
| $t$ | $1-(1-\sigma_{\min})t$ | $(\boldsymbol{x}_1 - (1-\sigma_{\min})\boldsymbol{x}_t)/(1-(1-\sigma_{\min})t)$ | $(t\boldsymbol{x}_1 - \boldsymbol{x}_t)/(1-t+t\sigma_{\min})^2$ |
| $t$ | $C = \left[(1-\lambda)t^2 + (\lambda-2)t+1\right]^{\frac{1}{2}}$ | $(((\lambda-2)t+2)\boldsymbol{x}_1 + (\lambda+2(1-\lambda)t-2)\boldsymbol{x}_t)/(2C^2)$ | $(t\boldsymbol{x}_1 - \boldsymbol{x}_t)/((1-\lambda)t^2 + (\lambda-2)t+1)$ |

*Table 4.* Closed-form relation between score and vector field for different noise schedules. The tuple $(\alpha_t, \sigma_t)$ represents the schedule of the path between noise and data, while $(\dot\alpha_t, \dot\sigma_t)$ is the respective time derivative. The relationship between score and vector field is obtained from Eq. (12).

| $\alpha_t$ | $\sigma_t$ | $\dot\alpha_t$ | $\dot\sigma_t$ | $\nabla_{\boldsymbol{x}_t} \log p_t^\theta(\boldsymbol{x}_t)$ | $u_t^\theta(\boldsymbol{x}_t)$ |
|---|---|---|---|---|---|
| $t$ | $1-t$ | $1$ | $-1$ | $(tu_t^\theta(\boldsymbol{x}_t)-\boldsymbol{x}_t)/(1-t)$ | $(\boldsymbol{x}_t + (1-t)\nabla_{\boldsymbol{x}_t} \log p_t^\theta(\boldsymbol{x}_t))/t$ |
| $t$ | $1-(1-\sigma_{\min})t$ | $1$ | $\sigma_{\min}-1$ | $(tu_t^\theta(\boldsymbol{x}_t)-\boldsymbol{x}_t)/(1-(1-\sigma_{\min})t)$ | $(\boldsymbol{x}_t + (1-(1-\sigma_{\min})t)\nabla_{\boldsymbol{x}_t} \log p_t^\theta(\boldsymbol{x}_t))/t$ |
| $t$ | $C = \left[(1-\lambda)t^2+(\lambda-2)t+1\right]^{\frac{1}{2}}$ | $1$ | $(\lambda-2(\lambda-1)t-2)/2C$ | $(2(tu_t^\theta(\boldsymbol{x}_t)-\boldsymbol{x}_t))/((\lambda-2)t+2)$ | $(2\boldsymbol{x}_t + ((\lambda-2)t+2)\nabla_{\boldsymbol{x}_t} \log p_t^\theta(\boldsymbol{x}_t))/2t$ |

## B.3. Reparameterization Challenges

We provide a list of the closed-form objectives for the conditional score and vector field across scheduling approaches in Tab. 3, as both can be evaluated analytically following Eq. (16) and Eq. (17). Moreover, we expand on the different schedules presented in Tab. 3 in App. B.4.

Based on App. B.2, one can learn the relationship between the score and the generating vector field, retrieving the counterpart through reparameterization in Eq. (12). In Tab. 4 we list the explicit equations tying the score and the generating vector field together in the marginal setting. One of the main challenges of this approach is the stability of the reparameterization method, where most of the tractable results from evaluating Eq. (12) explode at certain values of $t$. We provide qualitative evidence for the reparameterization instability in App. F.4.

**Score explosion.** At $t=1$, the score takes high values as the probability path concentrates most of the mass around the data. This results in the division by a low number in the conditional case, and in most of the reparameterization settings in Tab. 4.

**Vector field explosion.** Given an approximation for the marginal score, deriving the vector field implies a division by 0 at $t=0$ (see Tab. 4). However, if we learned the exact marginal score in Eq. (19), the proportionality with $1/t$ would cancel out. In fact, by the linearity of the conditional formulation, one can write the marginal score in Eq. (19) as $\nabla_{\boldsymbol{x}_t} \log p_t(\boldsymbol{x}_t) = (\mathbb{E}_{p_t(\boldsymbol{x}_1|\boldsymbol{x}_t)}[\boldsymbol{x}_1] - \boldsymbol{x}_t)/\sigma_t^2$ and plug it into Eq. (18), yielding:

$$
\begin{aligned}
u_t(\boldsymbol{x}_t) &= \frac{\dot\alpha_t}{\alpha_t}\boldsymbol{x}_t + \frac{\sigma_t}{\alpha_t}(\dot\alpha_t\sigma_t - \alpha_t\dot\sigma_t)\frac{\alpha_t\mathbb{E}_{p_t(\boldsymbol{x}_1|\boldsymbol{x}_t)}[\boldsymbol{x}_1] - \boldsymbol{x}_t}{\sigma_t^2} \\
&= \frac{\dot\alpha_t}{\alpha_t}\boldsymbol{x}_t + \frac{\dot\alpha_t}{\alpha_t}\left[\alpha_t\mathbb{E}_{p_t(\boldsymbol{x}_1|\boldsymbol{x}_t)}[\boldsymbol{x}_1] - \boldsymbol{x}_t\right] - \frac{\dot\sigma_t}{\sigma_t}\left[\alpha_t\mathbb{E}_{p_t(\boldsymbol{x}_1|\boldsymbol{x}_t)}[\boldsymbol{x}_1] - \boldsymbol{x}_t\right] \\
&= \frac{\dot\sigma_t}{\sigma_t}\boldsymbol{x}_t + \dot\alpha_t\mathbb{E}_{p_t(\boldsymbol{x}_1|\boldsymbol{x}_t)}[\boldsymbol{x}_1] - \frac{\dot\sigma_t}{\sigma_t}\alpha_t\mathbb{E}_{p_t(\boldsymbol{x}_1|\boldsymbol{x}_t)}[\boldsymbol{x}_1] \\
&= \frac{\dot\sigma_t\boldsymbol{x}_t + (\dot\alpha_t\sigma_t - \alpha_t\dot\sigma_t)\mathbb{E}_{p_t(\boldsymbol{x}_1|\boldsymbol{x}_t)}[\boldsymbol{x}_1]}{\sigma_t}.
\end{aligned}
\tag{24}
$$

Note that Eq. (24) does not have $\alpha_t$ in the denominator. However, given a parameterization $s^\psi$ of the score, we have $s^\psi \approx \nabla_{\boldsymbol{x}_t} \log p_t(\boldsymbol{x}_t)$. Thus, computing the vector field as a function of a parameterized score still involves a division by a low number, causing numerical instability.

## B.4. Explanation of the Schedules

In the above Tab. 3 and Tab. 4, we introduce three different scheduling options. We differentiate between schedules using the following taxonomy:

- Deterministic path between noise samples and data. Point mass around the data.
- Deterministic path between noise samples and data. Gaussian distribution around the data.
- Gaussian distribution around the path between the noise samples and the data. Point mass around the data.

Note that, while the probability paths are always stochastic in flow matching, the standard formulation form Lipman et al. (2023) involves deterministic interpolations between the data and the prior distribution, once the prior samples are fixed. In the following, we use Eq. (4) as a reference for the probability path.

**Deterministic path between noise samples and data. Point mass around the data.** The schedule consisting of $\alpha_t = t$ and $\sigma_t = 1 - t$ defines a deterministic interpolation between a noise sample $\epsilon \sim \mathcal{N}(\mathbf{0}_d, \mathbb{I}_d)$ and a data point $\boldsymbol{x}_1$. This can be easily seen after rewriting the formula for $\boldsymbol{x}_t$:

$$\boldsymbol{x}_t = t\boldsymbol{x}_1 + (1-t)\epsilon\,. \tag{25}$$

In other words, the interpolation exactly retrieves $\boldsymbol{x}_1$ and $\epsilon$ at times $t = 1$ and $t = 0$, respectively.

**Deterministic path between noise samples and data. Gaussian distribution around the data.** A common approach is to set a small variance $\sigma_{\min}$ at $t = 1$ (Lipman et al., 2023), preventing probability paths from degenerating at the data. This leads to $\alpha_t = t$ and $\sigma_t = 1 - (1 - \sigma_{\min})t$.

**Gaussian distribution around the path between the noise samples and the data. Point mass around the data.** The last schedule we explore sets $\alpha_t = t$ and $\sigma_t = \left[(1 - \lambda)t^2 + (\lambda - 2)t + 1\right]^{\frac{1}{2}}$. This is equivalent to drawing noise samples from $\epsilon \sim \mathcal{N}(\mathbf{0}_d, \mathbb{I}_d)$ and adding Gaussian noise to the path in Eq. (25) with variance $\lambda\,t(1 - t)$, as in Tong et al. (2024b). Specifically, we can rewrite this setting as follows:

$$\begin{aligned}
\boldsymbol{x}_t &\sim \mathcal{N}(\boldsymbol{\mu}_t, \lambda\,t(1 - t)\mathbb{I}_d)\,, \\
\boldsymbol{\mu}_t &\sim \mathcal{N}(\alpha_t\boldsymbol{x}_1, (1 - t)^2\mathbb{I}_d)\,.
\end{aligned} \tag{26}$$

The above hierarchical model can be collapsed into a single Gaussian path with, as variance, the sum of the variances of the two nested models. One can draw a sample thereof as:

$$\begin{aligned}
\boldsymbol{x}_t &= \boldsymbol{\mu}_t + \sqrt{\lambda\,t(1 - t)}\epsilon \\
&= (\alpha_t\boldsymbol{x}_1 + (1 - t)\epsilon') + \sqrt{\lambda\,t(1 - t)}\epsilon \\
&= \alpha_t\boldsymbol{x}_1 + ((1 - t)\epsilon' + \sqrt{\lambda\,t(1 - t)}\epsilon)\,,
\end{aligned} \tag{27}$$

with $\epsilon, \epsilon' \sim \mathcal{N}(\mathbf{0}_d, \mathbb{I}_d)$. By the property of the sum of Gaussian distributions, Eq. (27) implies:

$$\begin{aligned}
\boldsymbol{x}_t &\sim \mathcal{N}(\alpha_t\boldsymbol{x}_1, [\lambda\,t(1 - t) + (1 - t)^2]\mathbb{I}_d) \\
&= \mathcal{N}(\alpha_t\boldsymbol{x}_1, [(1 - \lambda)t^2 + (\lambda - 2)t + 1]\mathbb{I}_d)\,,
\end{aligned} \tag{28}$$

which is the schedule reported in Tab. 3 and Tab. 4. In the simplest case where $\lambda = 1$, $\sigma_t = \sqrt{1 - t}$.

### B.5. Derivation of Prop. 4.1

Let $\boldsymbol{x}_t \in \mathbb{R}^d$ be an integral trajectory as the solution of the following ODE:

$$\frac{\mathrm{d}}{\mathrm{d}t}\boldsymbol{x}_t = b_t(\boldsymbol{x}_t)\,, \tag{29}$$

with $t \in [0, 1]$. Moreover, let $\{p_t(\boldsymbol{x}_t)\}_{t \in [0,1]}$ and $\{p'_t(\boldsymbol{x}_t)\}_{t \in [0,1]}$ be two probability paths generated, respectively, by the vector fields $\{u_t(\boldsymbol{x}_t)\}_{t \in [0,1]}$ and $\{u'_t(\boldsymbol{x}_t)\}_{t \in [0,1]}$ according to the continuity equation:

$$\frac{\partial}{\partial t}p_t(\boldsymbol{x}) = -\nabla \cdot \big(p_t(\boldsymbol{x})\,u_t(\boldsymbol{x})\big), \tag{30}$$

$$\frac{\partial}{\partial t}p'_t(\boldsymbol{x}) = -\nabla \cdot \big(p'_t(\boldsymbol{x})\,u'_t(\boldsymbol{x})\big)\,, \tag{31}$$

such that $p_0 = p'_0 = \mathcal{N}(\mathbf{0}_d, \mathbb{I}_d)$. Note that these paths correspond to two distinct generative models sampling some data distribution starting from standard Gaussian noise.

We define the logarithm of the time-marginal density ratio as the solution of the following ODE:

$$\frac{\mathrm{d}}{\mathrm{d}t} \log r_t(\boldsymbol{x}_t) := \frac{\mathrm{d}}{\mathrm{d}t} \log \frac{p_t(\boldsymbol{x}_t)}{p'_t(\boldsymbol{x}_t)} \tag{32}$$

$$:= \frac{\mathrm{d}}{\mathrm{d}t} \log p_t(\boldsymbol{x}_t) - \frac{\mathrm{d}}{\mathrm{d}t} \log p'_t(\boldsymbol{x}_t), \tag{33}$$

with $\log r_0(\boldsymbol{x}_0) = 0$.

We express the total derivative of the individual densities as follows:

$$\frac{\mathrm{d}}{\mathrm{d}t} \log p_t(\boldsymbol{x}_t) = \frac{1}{p_t(\boldsymbol{x}_t)} \frac{\mathrm{d}}{\mathrm{d}t} p_t(\boldsymbol{x}_t) \tag{34}$$

$$= \frac{1}{p_t(\boldsymbol{x}_t)} \left[ \frac{\partial}{\partial t} p_t(\boldsymbol{x}_t) + (\nabla_{\boldsymbol{x}_t} p_t(\boldsymbol{x}_t))^\top \frac{\mathrm{d}}{\mathrm{d}t} \boldsymbol{x}_t \right] \tag{35}$$

$$= \frac{1}{p_t(\boldsymbol{x}_t)} \left[ -\nabla_{\boldsymbol{x}_t} \cdot (p_t(\boldsymbol{x}_t) u_t(\boldsymbol{x}_t)) + (\nabla_{\boldsymbol{x}_t} p_t(\boldsymbol{x}_t))^\top b_t(\boldsymbol{x}_t) \right] \tag{36}$$

$$= \frac{1}{p_t(\boldsymbol{x}_t)} \left[ -(\nabla_{\boldsymbol{x}_t} p_t(\boldsymbol{x}_t))^\top u_t(\boldsymbol{x}_t) - p_t(\boldsymbol{x}_t) \nabla_{\boldsymbol{x}_t} \cdot u_t(\boldsymbol{x}_t) + (\nabla_{\boldsymbol{x}_t} p_t(\boldsymbol{x}_t))^\top b_t(\boldsymbol{x}_t) \right] \tag{37}$$

$$= -\nabla_{\boldsymbol{x}_t} \cdot u_t(\boldsymbol{x}_t) + (b_t(\boldsymbol{x}_t) - u_t(\boldsymbol{x}_t))^\top \frac{\nabla_{\boldsymbol{x}_t} p_t(\boldsymbol{x}_t)}{p_t(\boldsymbol{x}_t)} \tag{38}$$

$$= -\nabla_{\boldsymbol{x}_t} \cdot u_t(\boldsymbol{x}_t) + (b_t(\boldsymbol{x}_t) - u_t(\boldsymbol{x}_t))^\top \nabla_{\boldsymbol{x}_t} \log p_t(\boldsymbol{x}_t). \tag{39}$$

and

$$\frac{\mathrm{d}}{\mathrm{d}t} \log p'_t(\boldsymbol{x}_t) = \frac{1}{p'_t(\boldsymbol{x}_t)} \frac{\mathrm{d}}{\mathrm{d}t} p'_t(\boldsymbol{x}_t) \tag{40}$$

$$= \frac{1}{p'_t(\boldsymbol{x}_t)} \left[ \frac{\partial}{\partial t} p'_t(\boldsymbol{x}_t) + (\nabla_{\boldsymbol{x}_t} p'_t(\boldsymbol{x}_t))^\top \frac{\mathrm{d}}{\mathrm{d}t} \boldsymbol{x}_t \right] \tag{41}$$

$$= \frac{1}{p'_t(\boldsymbol{x}_t)} \left[ -\nabla_{\boldsymbol{x}_t} \cdot (p'_t(\boldsymbol{x}_t) u'_t(\boldsymbol{x}_t)) + (\nabla_{\boldsymbol{x}_t} p'_t(\boldsymbol{x}_t))^\top b_t(\boldsymbol{x}_t) \right] \tag{42}$$

$$= \frac{1}{p'_t(\boldsymbol{x}_t)} \left[ -(\nabla_{\boldsymbol{x}_t} p'_t(\boldsymbol{x}_t))^\top u'_t(\boldsymbol{x}_t) - p'_t(\boldsymbol{x}_t) \nabla_{\boldsymbol{x}_t} \cdot u'_t(\boldsymbol{x}_t) + (\nabla_{\boldsymbol{x}_t} p'_t(\boldsymbol{x}_t))^\top b_t(\boldsymbol{x}_t) \right] \tag{43}$$

$$= -\nabla_{\boldsymbol{x}_t} \cdot u'_t(\boldsymbol{x}_t) - (u'_t(\boldsymbol{x}_t) - b_t(\boldsymbol{x}_t))^\top \frac{\nabla_{\boldsymbol{x}_t} p'_t(\boldsymbol{x}_t)}{p'_t(\boldsymbol{x}_t)} \tag{44}$$

$$= -\nabla_{\boldsymbol{x}_t} \cdot u'_t(\boldsymbol{x}_t) - (u'_t(\boldsymbol{x}_t) - b_t(\boldsymbol{x}_t))^\top \nabla_{\boldsymbol{x}_t} \log p'_t(\boldsymbol{x}_t). \tag{45}$$

Then we plug the results from Eq. (39) and Eq. (45) into Eq. (33) to get:

$$\begin{aligned}
\frac{\mathrm{d}}{\mathrm{d}t} \log r_t(\boldsymbol{x}_t) = {}& \nabla_{\boldsymbol{x}_t} \cdot (u'_t(\boldsymbol{x}_t) - u_t(\boldsymbol{x}_t)) \\
& + (b_t(\boldsymbol{x}_t) - u_t(\boldsymbol{x}_t))^\top \nabla_{\boldsymbol{x}_t} \log p_t(\boldsymbol{x}_t) + (u'_t(\boldsymbol{x}_t) - b_t(\boldsymbol{x}_t))^\top \nabla_{\boldsymbol{x}_t} \log p'_t(\boldsymbol{x}_t).
\end{aligned} \tag{46}$$

### B.6. Remarks on Prop. 4.1

We provide an interpretation of the effect of the individual terms in Eq. (46) on the density ratio as follows:

(I) $r_t(\boldsymbol{x}_t)$ increases when the field $u_t$ concentrates probability mass more strongly around $\boldsymbol{x}_t$ than $u'_t$, as quantified by $\nabla_{\boldsymbol{x}_t} \cdot (u'_t(\boldsymbol{x}_t) - u_t(\boldsymbol{x}_t))$. This suggests that the mass aggregates more at the numerator than at the denominator of the ratio.

(II) $(b_t(\boldsymbol{x}_t) - u_t(\boldsymbol{x}_t))^\top \nabla_{\boldsymbol{x}_t} \log p_t(\boldsymbol{x}_t)$ and $(u'_t(\boldsymbol{x}_t) - b_t(\boldsymbol{x}_t))^\top \nabla_{\boldsymbol{x}_t} \log p'_t(\boldsymbol{x}_t)$ suggest that:

- $r_t(\boldsymbol{x}_t)$ decreases when $p'_t$ grows faster along its generated field $u'_t$ than the simulating field $b_t$.
- $r_t(\boldsymbol{x}_t)$ increases when $p_t$ grows faster along its generated field $u_t$ than the simulating field $b_t$.

In other words, these terms can be interpreted as *correction terms* that account for the fact that the densities in the ratio are tracked along trajectories that do not follow their own generating vector fields. Either term vanishes when the simulation field $b_t$ coincides with $u_t$ or $u'_t$.

### B.7. Integral Solution of the Ratio

The explicit formulation of the log-likelihood ratios between conditional densities when the simulating field is also the generating field of the numerator (**S1** setting) is:

$$
\log r_1^\theta(\boldsymbol{x}_1 \mid \boldsymbol{y}, \boldsymbol{y}') = - \int_1^0 [\nabla_{\boldsymbol{x}_t} \cdot (u_t^\theta(\boldsymbol{x}_t \mid \boldsymbol{y}') - u_t^\theta(\boldsymbol{x}_t \mid \boldsymbol{y}))
$$
$$
+ (u_t^\theta(\boldsymbol{x}_t \mid \boldsymbol{y}') - u_t^\theta(\boldsymbol{x}_t \mid \boldsymbol{y}))^\top \nabla_{\boldsymbol{x}_t} \log p_t^\theta(\boldsymbol{x}_t \mid \boldsymbol{y}')] \mathrm{d}t \,,
$$
$$
\text{with} \quad \boldsymbol{x}_t = \boldsymbol{x}_1 + \int_1^t u_s^\theta(\boldsymbol{x}_s \mid \boldsymbol{y}) \mathrm{d}s \,. \tag{47}
$$

Under **S2**, we provide the same formulation for S2 when the simulating vector field $b_t(\boldsymbol{x}_t)$ is the velocity $u_t^\theta(\boldsymbol{x}_t)$ generating the unconditional density $p_t^\theta(\boldsymbol{x}_t)$:

$$
\log r_1^\theta(\boldsymbol{x}_1 \mid \boldsymbol{y}, \boldsymbol{y}') = - \int_1^0 [\nabla_{\boldsymbol{x}_t} \cdot (u_t^\theta(\boldsymbol{x}_t \mid \boldsymbol{y}') - u_t^\theta(\boldsymbol{x}_t \mid \boldsymbol{y}))
$$
$$
+ (u_t^\theta(\boldsymbol{x}_t \mid \boldsymbol{y}') - u_t^\theta(\boldsymbol{x}_t))^\top \nabla_{\boldsymbol{x}_t} \log p_t^\theta(\boldsymbol{x}_t \mid \boldsymbol{y}')
$$
$$
+ (u_t^\theta(\boldsymbol{x}_t) - u_t^\theta(\boldsymbol{x}_t \mid \boldsymbol{y}))^\top \nabla_{\boldsymbol{x}_t} \log p_t^\theta(\boldsymbol{x}_t \mid \boldsymbol{y})] \mathrm{d}t \tag{48}
$$
$$
\text{with} \quad \boldsymbol{x}_t = \boldsymbol{x}_1 + \int_1^t u_s^\theta(\boldsymbol{x}_s) \mathrm{d}s \,.
$$

Note that, different from Eq. (47), none of the terms disappear, similar to the general setting described in Prop. 4.1.

As explained in Sec. 4.5, $u_t^\theta(\boldsymbol{x}_t)$ follows the same parameterization as the conditional counterpart and can be achieved during training by replacing the conditioner with an empty token $\varnothing$ with a certain probability $\beta$.

**S1 vs. S2 choice (Sec. 4.2).** We use S1 across the whole paper, except for the MI experiment in Sec. 5.2, where S2 worked significantly better in a high-dimensional setting (especially for $d = 320$). We recommend working with S2 in the presence of a lower overlap between the conditional densities in the ratio and a higher dimensionality, while prioritizing S1 in all other cases for efficiency (see Prop. 4.2 and App. B.9).

### B.8. Parameterization error analysis

We present a first-order error analysis derived by approximating both the vector field and the score as neural networks. For simplicity, we consider the setting (**S1**) from Sec. 4.2.

**The general case.** Here, we do not assume a distributional form of the data distribution.

**Proposition B.1.** *Let $u_t, u'_t$ be the true vector fields for the numerator and denominator flows, and let $\hat{u}_t, \hat{u}'_t$ be their neural approximations. Moreover, let $\nabla \log p'_t$ denote the true score of the denominator density and $\hat{s}_t$ its approximation. Define the approximation errors*

$$
\delta_u = \hat{u}_t - u_t, \tag{49}
$$
$$
\delta_{u'} = \hat{u}'_t - u'_t, \tag{50}
$$
$$
\delta_s = \hat{s}_t - \nabla \log p'_t. \tag{51}
$$

*Given $\epsilon_t = \log \hat{r}_t - \log r_t$ as the error between the true and approximated log-likelihood ratios evaluated along the same trajectory, with $\epsilon_0 = 0$, and assume $\|u_t - u'_t\| \leq L_u$. Then, the first-order approximation of the final error satisfies*

$$
|\epsilon_1| \lesssim \int_0^1 (|\nabla \cdot (\delta_{u'} - \delta_u)| + \|\delta_{u'} - \delta_u\| \cdot \|\nabla \log p'_t\| + L_u \|\delta_s\|) \, \mathrm{d}t. \tag{52}
$$

*Proof.* Under **S1**, the time derivatives of the true and approximate log-ratios are given by

$$\frac{\mathrm{d}}{\mathrm{d}t} \log r_t = \nabla \cdot (u'_t - u_t) + (u'_t - u_t)^\top \nabla \log p'_t, \tag{53}$$

$$\frac{\mathrm{d}}{\mathrm{d}t} \log \hat{r}_t = \nabla \cdot (\hat{u}'_t - \hat{u}_t) + (\hat{u}'_t - \hat{u}_t)^\top \hat{s}_t. \tag{54}$$

Substituting $\hat{u}_t = u_t + \delta_u$, $\hat{u}'_t = u'_t + \delta_{u'}$, and $\hat{s}_t = \nabla \log p'_t + \delta_s$, we obtain

$$\frac{\mathrm{d}}{\mathrm{d}t} \log \hat{r}_t = \nabla \cdot \big((u'_t + \delta_{u'}) - (u_t + \delta_u)\big) \tag{55}$$

$$+ \big((u'_t + \delta_{u'}) - (u_t + \delta_u)\big)^\top (\nabla \log p'_t + \delta_s). \tag{56}$$

Define the error derivative

$$\frac{\mathrm{d}}{\mathrm{d}t} \epsilon_t = \frac{\mathrm{d}}{\mathrm{d}t} \log \hat{r}_t - \frac{\mathrm{d}}{\mathrm{d}t} \log r_t. \tag{57}$$

After the cancellation of shared terms, this yields

$$\frac{\mathrm{d}}{\mathrm{d}t} \epsilon_t = \nabla \cdot (\delta_{u'} - \delta_u) + (\delta_{u'} - \delta_u)^\top \nabla \log p'_t \tag{58}$$

$$+ (u'_t - u_t)^\top \delta_s + (\delta_{u'} - \delta_u)^\top \delta_s. \tag{59}$$

Using a first-order error approximation, we drop the second-order term $(\delta_{u'} - \delta_u)^\top \delta_s$. Taking absolute values and applying the triangle inequality,

$$|\epsilon_1| = \left| \int_0^1 \frac{\mathrm{d}}{\mathrm{d}t} \epsilon_t \, dt \right| \le \int_0^1 \left| \frac{\mathrm{d}}{\mathrm{d}t} \epsilon_t \right| \mathrm{d}t. \tag{60}$$

Applying Cauchy–Schwarz on the first-order error approximation,

$$|\epsilon_1| \lesssim \int_0^1 \Big( |\nabla \cdot (\delta_{u'} - \delta_u)| + \|\delta_{u'} - \delta_u\| \cdot \|\nabla \log p'_t\| + \|u'_t - u_t\| \cdot \|\delta_s\| \Big) \mathrm{d}t. \tag{61}$$

Finally, using $\|u'_t - u_t\| \le L_u$ yields the result. $\qquad\square$

**The Gaussian Case.** Here, we show how the above derivation translates to the Gaussian simulation setting.

**Proposition B.2.** *Consider the ratio* $\log r_t = \log \frac{p_1}{p'_1}$ *where*

$$p_0 = p'_0 = \mathcal{N}(\mathbf{0}_d, \mathbb{I}_d), \quad p_1 = \mathcal{N}(c \cdot \mathbf{1}_d, \mathbb{I}_d), \quad p'_1 = \mathcal{N}(\mathbf{0}_d, \mathbb{I}_d). \tag{62}$$

*Assume a conditional probability path with* $\sigma_t = 1 - t$ *and* $\alpha_t = t$. *Then the expected log-ratio error satisfies*

$$\mathbb{E}[|\epsilon_1|] \lesssim \int_0^1 \Big( \mathbb{E}[|\nabla \cdot (\delta_{u'} - \delta_u)|] + \frac{\sqrt{d}}{\tilde{\sigma}_t} \mathbb{E}[\|\delta_{u'} - \delta_u\|] + c \frac{\sqrt{d}(1 - t)}{\tilde{\sigma}_t^2} \mathbb{E}[\|\delta_s\|] \Big) dt, \tag{63}$$

*where* $\tilde{\sigma}_t$ *is the marginal variance induced by the flow and* $d$ *the data dimensionality. In particular, the velocity and score error terms scale as* $\mathcal{O}(\sqrt{d})$.

*Proof.* Under Gaussian probability paths, the marginal distributions are also Gaussian:

$$p_t = \mathcal{N}(t\, c \cdot \mathbf{1}_d, \tilde{\sigma}_t^2 \mathbb{I}_d), \tag{64}$$

$$p'_t = \mathcal{N}(\mathbf{0}_d, \tilde{\sigma}_t^2 \mathbb{I}_d), \tag{65}$$

where the marginal variance $\tilde{\sigma}_t$ is a function of $\alpha_t$ and $\sigma_t$.

The score of $p'_t$ is given by

$$\nabla \log p'_t(\boldsymbol{x}_t) = -\frac{\boldsymbol{x}_t}{\tilde{\sigma}_t^2}, \tag{66}$$

hence

$$\|\nabla \log p'_t(\boldsymbol{x}_t)\| = \frac{1}{\tilde{\sigma}_t^2}\|\boldsymbol{x}_t\|. \tag{67}$$

Moreover,

$$\mathbb{E}[\|\nabla \log p'_t(\boldsymbol{x}_t)\|] \approx \frac{\sqrt{d}}{\tilde{\sigma}_t}. \tag{68}$$

Using the relationship between score and vector field in Gaussian flow matching and using Eq. (64) and Eq. (65), we obtain

$$\|u'_t(\boldsymbol{x}_t) - u_t(\boldsymbol{x}_t)\| = \frac{1-t}{\tilde{\sigma}_t^2}\|c \cdot \mathbf{1}_d\| = c\frac{\sqrt{d}(1-t)}{\tilde{\sigma}_t^2}. \tag{69}$$

Plugging these expressions into the general error bound in Eq. (61) and taking expectations yields

$$\mathbb{E}[|\epsilon_1|] \lesssim \int_0^1 \left( \mathbb{E}[|\nabla \cdot (\delta_{u'} - \delta_u)|] + \frac{\sqrt{d}}{\tilde{\sigma}_t}\mathbb{E}[\|\delta_{u'} - \delta_u\|] + c\frac{\sqrt{d}(1-t)}{\tilde{\sigma}_t^2}\mathbb{E}[\|\delta_s\|] \right) \mathrm{d}t. \tag{70}$$

$$\square$$

**Discussion.** The bound decomposes the total error into three contributions: (I) divergence error, (II) velocity approximation error, and (III) score approximation error. The latter two terms scale explicitly with $\sqrt{d}$, showing that approximation errors in the vector field and score are amplified with dimensionality, while the divergence term depends on the structure of the approximation errors.

### B.9. Proof of Prop. 4.2

*Proof.* First, note that the expected log-ratio under $p_t^b$ admits the following identity:

$$\begin{aligned}
\mathbb{E}_{p_t^b}[\log r_t(\boldsymbol{x}_t)] &= \mathbb{E}_{p_t^b}\left[\log \frac{p_t(\boldsymbol{x}_t)}{p'_t(\boldsymbol{x}_t)}\right] \\
&= \mathbb{E}_{p_t^b}\left[\log \frac{p_t^b(\boldsymbol{x}_t)}{p'_t(\boldsymbol{x}_t)}\right] - \mathbb{E}_{p_t^b}\left[\log \frac{p_t^b(\boldsymbol{x}_t)}{p_t(\boldsymbol{x}_t)}\right] \\
&= \mathrm{KL}(p_t^b\|p'_t) - \mathrm{KL}(p_t^b\|p_t) = \Delta\mathrm{KL}(t).
\end{aligned} \tag{71}$$

Since $p_0 = p'_0$, we have $\Delta\mathrm{KL}(0) = 0$. Applying Jensen's inequality $(\mathbb{E}[X^2] \geq (\mathbb{E}[X])^2)$ pointwise in $t$:

$$\mathbb{E}_{p_t^b}\left[\left(\frac{\mathrm{d}}{\mathrm{d}t}\log r_t(\boldsymbol{x}_t)\right)^2\right] \geq \left(\mathbb{E}_{p_t^b}\left[\frac{\mathrm{d}}{\mathrm{d}t}\log r_t(\boldsymbol{x}_t)\right]\right)^2. \tag{72}$$

Integrating both sides over $[0, 1]$:

$$\int_0^1 \mathbb{E}_{p_t^b}\left[\left(\frac{\mathrm{d}}{\mathrm{d}t}\log r_t(\boldsymbol{x}_t)\right)^2\right]\mathrm{d}t \geq \int_0^1 \left(\mathbb{E}_{p_t^b}\left[\frac{\mathrm{d}}{\mathrm{d}t}\log r_t(\boldsymbol{x}_t)\right]\right)^2\mathrm{d}t. \tag{73}$$

Under the regularity conditions allowing the exchange of derivatives and expectations:

$$\int_0^1 \left( \mathbb{E}_{p_t^b} \left[ \frac{d}{dt} \log r_t(\boldsymbol{x}_t) \right] \right)^2 dt = \int_0^1 \left( \frac{d}{dt} \mathbb{E}_{p_t^b} [\log r_t(\boldsymbol{x}_t)] \right)^2 dt = \int_0^1 \left( \frac{d}{dt} \Delta\mathrm{KL}(t) \right)^2 dt, \qquad (74)$$

where the last equality uses Eq. (71). Applying the Cauchy-Schwarz inequality:

$$\int_0^1 \left( \frac{d}{dt} \Delta\mathrm{KL}(t) \right)^2 dt \geq \left[ \int_0^1 \frac{d}{dt} \Delta\mathrm{KL}(t)\, dt \right]^2 = [\Delta\mathrm{KL}(1) - \Delta\mathrm{KL}(0)]^2 = (\Delta\mathrm{KL}(1))^2, \qquad (75)$$

where the final equality uses $\Delta\mathrm{KL}(0) = 0$. Chaining the inequalities completes the proof. $\qquad\square$

## C. Additional details

### C.1. Batch Correction Task

Evaluation of batch correction in single-cell RNA sequencing typically relies on neighborhood-based analyses. A common protocol consists of constructing local neighborhoods using k-nearest neighbors (kNN) before and after correction, followed by measuring changes in batch label purity within these neighborhoods (Luecken et al., 2022).

Our approach follows the same underlying principle of assessing batch mixing, but replaces heuristic purity measures with a likelihood-based criterion. Specifically, we evaluate whether corrected cell embeddings are more likely under a distribution conditioned jointly on batch and cell type labels than under a distribution conditioned on cell type alone. This formulation provides a principled proxy for batch mixing that is directly comparable across correction methods, while leveraging the expressiveness of exact-likelihood generative models.

### C.2. Drug Combination Analysis as Differential Abundance

We frame the drug combination experiment as a differential abundance (DA) problem, where the reference distribution is conditioned on a single-drug treatment rather than a control condition. In single-cell analysis, perturbation effects are commonly studied through likelihood-based formulations, including approaches based on kNN graphs (Dann et al., 2022), manifold learning (Burkhardt et al., 2021), and variational autoencoders (Boyeau et al., 2025). These methods have also been extended to combinatorial perturbation settings (Otto et al., 2025).

Perturbations primarily induce shifts in the distribution of cellular states, altering their relative frequencies across the data manifold. Consequently, the objective is to quantify changes in probability mass rather than to perform classification. While classifiers can provide rankings of perturbations, they do not directly estimate differential abundance.

### C.3. Literature Positioning of Flow-Based Likelihood Ratio Estimation

**Classification-based Ratio Estimation.** A common strategy for estimating density ratios relies on classification-based approaches. These methods reduce ratio estimation to a binary classification problem. While simple and widely applicable, their performance degrades in high-dimensional settings when the supports of the underlying distributions differ. In such cases, the classifier tends to saturate, resulting in poor ratio estimates.

In contrast, our approach models likelihoods directly using normalizing flows, enabling explicit ratio estimation. This generative formulation improves robustness to increasing dimensionality. However, as discussed in the limitations, the lack of shared support between distributions can still affect performance.

**Moment Matching and Kernel-based Methods.** Moment matching and kernel-based approaches, such as KLIEP (Savkin & Tombari, 2020), estimate density ratios without explicitly parameterizing the underlying densities. Instead, they rely on importance reweighting to match moments or minimize divergence measures. Although these methods are theoretically well-founded, they typically scale poorly to high-dimensional data and are sensitive to the choice of kernel.

Our method instead learns densities via a flow-based model while tracking likelihood ratios during simulation, yielding a more flexible neural estimator that avoids explicit kernel design.

**Score-based Methods.** Score-based approaches, including TSM (Choi et al., 2022) and CTSM (Yu et al., 2025), estimate density ratios by defining interpolation paths between distributions and integrating the corresponding time-dependent score function. In particular, CTSM derives a closed-form expression for the ratio under a flow matching formulation.

In contrast, our approach uses ordinary differential equations to transform prior noise into target densities without assuming a predefined interpolation path between distributions. Once a conditional continuous normalizing flow is trained, density ratios for arbitrary conditioning pairs can be computed without retraining. This differs from TSM and CTSM, which typically require retraining for each pair of distributions.

## C.4. Failure Detection

**Failure Scenarios.** As observed in the discussion section, scRatio may struggle to effectively estimate the likelihood ratio between densities with non-overlapping supports. This is due to the fact that one of the densities is tracked using a non-simulating field, which, in turn, increases the estimation error. We propose two approaches to identify such failure scenarios with little to no additional computational overhead.

**Histogram Inspection.** When comparing distributions $p(\cdot \mid y)$ and $p(\cdot \mid y')$ with low shared support, the ratio $\log \frac{p(\cdot \mid y)}{p(\cdot \mid y)}$ will explode in either direction. The arguably simplest way to diagnose this eventuality is to visualize the histograms of the log ratios of points under $p(\cdot \mid y)$ and under $p(\cdot \mid y')$. The overlap of such histograms will indicate similar weighting of some manifold regions. Non-overlapping histograms would signal a low shared support between the considered densities.

**Density of Pulled-Back Noise under Prior.** We note that, when estimating density ratios between densities $p(\cdot \mid y)$ and $p(\cdot \mid y')$, one simulates trajectories from data to noise backward in time. An alternative method to uncover the lack of shared support between $p(\cdot \mid y)$ and $p(\cdot \mid y')$ is based on the observation that, for overlapping distributions, the pulled-back noise of points under $p(\cdot \mid y')$, obtained by estimating the likelihood using $u_t(\cdot \mid y)$ as the simulating field, should distribute as the (tractable) noise distribution $p_0(\cdot)$. This can be verified by evaluating any divergence or likelihood measure against such a tractable noise distribution.

# D. Experimental Setup

## D.1. Parameterizations

As explained in App. B.3, we parameterize the score and the vector field with neural networks. Our model consists of the following units:

1. **Encoder:** A fully connected neural network that encodes observations into a latent space. The number of latent dimensions is a hyperparameter.
2. **Condition representation network:** A neural network that takes one-hot-encoded conditions as input and returns a dense representation.
3. **Time encoder:** A standard sinusoidal time encoder (Vaswani et al., 2017) followed by a fully-connected representation module.
4. **Separate score and vector field head:** Two fully connected neural networks inputting a concatenation between latent state, time, and condition encodings and outputting predictions for the marginal score and vector fields (see Tab. 4 for the conditional objectives used for the optimization).

For all modules, we use the SELU activation function. Moreover, we fix the architecture of both the score and vector field heads to 1024 neurons for three layers.

## D.2. Gaussian Simulation Experiment

**Task.** In the first synthetic experiment, we follow Yu et al. (2025) and evaluate the model on estimating closed-form log-likelihood ratios on multivariate Gaussian distributions. We consider the following tractable ratio:

$$\log r(\boldsymbol{x}) = \log \frac{q(\boldsymbol{x})}{q'(\boldsymbol{x})},$$
$$q = \mathcal{N}(s\mathbb{1}_d, \mathbb{I}_d) \quad q' = \mathcal{N}(\boldsymbol{0}_d, \mathbb{I}_d),$$

where $\mathbb{1}_d$ is the $d$-dimensional all-ones vector, $s \in \{1, 2\}$ a scalar defining different datasets, and $\boldsymbol{x} \in \mathbb{R}^d$. For each value of $s$, we consider the model's performance across $d \in \{2, 5, 10, 20, 30\}$.

**Choices for scRatio.** We use a condition-aware flow matching model $\{p_t^\theta(\boldsymbol{x}_t \mid y)\}_{t \in [0,1]}$, with parameterized vector field $\{u_t^\theta(\boldsymbol{x}_t \mid y)\}_{t \in [0,1]}$, to implement scRatio. Here, $p_t^\theta(\cdot \mid y = 0)$ and $p_t^\theta(\cdot \mid y = 1)$ approximate, respectively, $q'$ and $q$. After training the flow model, we simulate the ratio using Prop. 4.1 as a simulation field while tracking the ratio. As a dataset, we sample 100,000 points from the two distributions and split them into $90\%$ training and $10\%$ test sets. Models are trained with a learning rate of 1e-4 across 100,000 steps. The results are averaged across 5 repetitions.

**scRatio Parameter Sweep.** As presented in Sec. 5.1, we evaluate our approach across different schedulers. The results shown in Fig. 2 are based on, for each scheduler, the configuration that yielded the best test-set performance. Alongside the scheduling hyperparameters, we also consider model size and encoding mechanisms, and provide the grid of tested hyperparameters in Tab. 5.

*Table 5.* The hyperparameter grid explored for the Gaussian simulation dataset. Each model with a different configuration of hyperparameters is compared based on the test-set ratio estimation performance measured as MSE.

| Hyperparameters | Values |
| --- | --- |
| Latent space size | 32, 64, 128, 256, 512 |
| $\lambda$ | 0, 0.1, 0.25, 0.5, 0.75, 1 |
| $\sigma_{\min}$ | 0, 1e-4, 1e-3, 0.01, 0.1 |
| Use sinusoidal time embedding | True, False |
| Sinusoidal time embedding size | 32, 64, 128 |
| Condition embedding size | 32, 64, 128 |

**Baselines.** For this experiment, we compare scRatio with three baselines.

- **Naive approach:** In the naive approach, instead of estimating the ratio according to Prop. 4.1, we separately evaluate the numerator and denominator likelihoods. Notably, the difference between scRatio and the naive approach is only at

inference time. Therefore, in our results, we use the same trained conditional model to evaluate both scRatio and the naive solution to exclude any performance differences grounded in neural network optimization.

- **Time Score Matching (TSM)** (Choi et al., 2022): TSM relies on probability paths between pairs of densities. For convenience, given a point $\boldsymbol{x} \in \mathbb{R}^d$, we restate the task as the estimation of the following ratio between pairs of probability distributions:

$$\log r(\boldsymbol{x}) = \log \frac{p_1(\boldsymbol{x})}{p_0(\boldsymbol{x})},$$

where $p_0(\boldsymbol{x})$ and $p_1(\boldsymbol{x})$ are two densities over $\mathbb{R}^d$. Let $\{p_t(\boldsymbol{x}_t)\}_{t\in[0,1]}$ be a time-resolved interpolant between $p_0(\boldsymbol{x})$ and $p_1(\boldsymbol{x})$. One can express the ratio between densities as follows:

$$\log r(\boldsymbol{x}) = \log \frac{p_1(\boldsymbol{x})}{p_0(\boldsymbol{x})} = \int_0^1 \frac{\partial}{\partial t} \log p_t(\boldsymbol{x}) \mathrm{d}t \, .$$

Hence, the goal of TSM is to parameterize the time-score $\frac{\partial}{\partial t} \log p_t(\boldsymbol{x}) \mathrm{d}t$ with a neural network $s_t^{\psi}(\boldsymbol{x})$ and integrate it through time. More specifically, Choi et al. (2022) use the following simulation-free objective relying on integration by parts:

$$\mathcal{L}_{\mathrm{TSM}}(\psi) = 2 \, \mathbb{E}_{p_0(\boldsymbol{x})} \left[ s_0^{\psi}(\boldsymbol{x}) \right] - 2 \, \mathbb{E}_{p_1(\boldsymbol{x})} \left[ s_1^{\psi}(\boldsymbol{x}) \right] + \mathbb{E}_{p_t(\boldsymbol{x})} \left[ 2 \frac{\partial}{\partial t} s_t^{\psi}(\boldsymbol{x}) + 2 \, \dot{\lambda}(t) \, s_t^{\psi}(\boldsymbol{x}) + \lambda(t) \, s_t^{\psi}(\boldsymbol{x})^2 \right] \, ,$$

with $\lambda(t)$ representing a time-dependent weighting function.

- **Conditional Time Score Matching (CTSM)** (Yu et al., 2025): CTSM borrows concepts from flow matching (Albergo & Vanden-Eijnden, 2023; Lipman et al., 2023; Tong et al., 2024a) to define a TSM objective conditioned on the data points. By defining a tractable interpolation between samples from $p_0$ and $p_1$, the time-score of the conditional $p_t$ is available in closed form and can be regressed against. In analogy with flow matching, regressing the conditional version of the CTSM objective corresponds to learning the marginal TSM in expectation. Moreover, the authors offer a vectorized variant (CTSMv), where they split the optimization problem across the $d$ dimensions.

**Choice of Sample-based Ratio Estimation Variants for TSM and CTSM.** In the simulation experiments described in Sec. 5.1 and Sec. 5.2, one of the two densities in the ratio is a closed-form standard normal. This opens the door to assuming tractability of $p_0$ and using closed-form probabilistic interpolants relying exclusively on samples from $p_1$.

For a fair comparison with scRatio, which does not assume any distributional form for either the numerator or denominator of the density ratio, we focus on the CTSM and TSM model variants with Schrödinger Bridge (SB) probability paths. In essence, instead of relying on schedules interpolating between the data and a standard normal distribution, we define paths between samples from both distributions as stochastic interpolants. Let $\boldsymbol{x}_0 \sim p_0(\boldsymbol{x}_0)$ and $\boldsymbol{x}_1 \sim p_1(\boldsymbol{x}_1)$ be samples from the two distributions whose ratio we seek to estimate. The SB path is:

$$\boldsymbol{x}_t = t\boldsymbol{x}_1 + (1-t)\boldsymbol{x}_0 + \sigma\sqrt{t(1-t)}\epsilon \, ,$$

where $\epsilon \sim \mathcal{N}(\mathbf{0}_d, \mathbb{I}_d)$. Note that, with this formulation, $p_0$ can be any distribution for which samples are available, allowing for data-driven likelihood ratio estimation similar to scRatio.

**TSM and CTSM parameter sweep.** To ensure a fair comparison against these baselines, we conducted a grid search over the optimization learning rate and the path variance of the Schrödinger bridge version of TSM and CTSM. In particular, we set the learning rate to $\{5 \cdot 10^{-4}, 10^{-3}, 5 \cdot 10^{-3}\}$ and the path variance to $\{0.1, 0.5, 1.0\}$. We repeated each experiment 5 times, selected the best configuration from the average MSE from the validation set, and eventually reported the same quantity evaluated on a test set, shared across all compared methods.

### D.3. Mutual Information Estimation Experiment

**Task.** In this experiment, we use density ratios as a proxy for Mutual Information (MI) estimation. More specifically, we consider the following two Gaussian distributions:

$$q = \mathcal{N}(\mathbf{0}_d, \boldsymbol{\Sigma}) \, , q' = \mathcal{N}(\mathbf{0}_d, \mathbb{I}_d) \, ,$$

where $d$ is an even number of dimensions and $\mathbf{\Sigma}$ is a block-diagonal matrix of $2 \times 2$ blocks with 1's on the diagonals and 0.8 in the off-diagonal entries. In other words, each odd dimension $k$ is correlated with the even dimension $k+1$ and uncorrelated with the others.

Given this structured setup and an observation $\boldsymbol{x} \sim q(\boldsymbol{x})$, one can show that the expectation of the $\log$ density ratio between $q$ and $q'$ corresponds to the MI between the vectors $\boldsymbol{x}^{\mathrm{odd}} = (x^1, x^3, ..., x^{d-1})$ and $\boldsymbol{x}^{\mathrm{even}} = (x^2, x^4, ..., x^d)$.

**Relationship between MI and density ratio.** Consider the two multivariate random variables $X^{\mathrm{even}}$ and $X^{\mathrm{odd}}$ representing the even and odd dimensions of $X$, respectively. Individually, $X^{\mathrm{even}}$ and $X^{\mathrm{odd}}$ are multivariate standard normal as:

- Odd dimensions are correlated only with even dimensions (and vice versa) and uncorrelated with each other.

- The individual dimensions of a multivariate Gaussian random variable are also normally distributed.

In other words, the following is true:

$$q(X^{\mathrm{even}}) = q(X^{\mathrm{odd}}) = \mathcal{N}(\mathbf{0}_{\frac{d}{2}}, \mathbb{I}_{\frac{d}{2}}), \quad q(X^{\mathrm{even}}, X^{\mathrm{odd}}) = \mathcal{N}(\mathbf{0}_d, \mathbf{\Sigma}). \tag{76}$$

Now, we write the MI as a Kullback-Leibler (KL) divergence:

$$\begin{aligned} I(X^{\mathrm{even}}, X^{\mathrm{odd}}) &= \mathbb{E}_{q(X^{\mathrm{even}}, X^{\mathrm{odd}})} \left[ \log \frac{q(X^{\mathrm{even}}, X^{\mathrm{odd}})}{q(X^{\mathrm{even}})q(X^{\mathrm{odd}})} \right] \\ &= \mathbb{E}_{q(X^{\mathrm{even}}, X^{\mathrm{odd}})} \left[ \log \frac{q(X)}{q'(X)} \right], \end{aligned} \tag{77}$$

where the last step follows from the fact that $X^{\mathrm{even}}$ and $X^{\mathrm{odd}}$ are independent multivariate standard normal random variables and their concatenation $X = (X^{\mathrm{even}}, X^{\mathrm{odd}})$ follows a standard multivariate normal distribution, i.e., $q'(X) = \mathcal{N}(\mathbf{0}_d, \mathbb{I}_d)$. In other words, considering a Monte Carlo approximation of the MI in Eq. (77), we can estimate the MI by averaging an estimate of the $\log$-likelihood ratio between distributions over samples from $q$.

**Ground truth MI.** By the properties of the multivariate normal distribution, the ground truth MI of Eq. (77) is available in closed form as follows:

$$I(X^{\mathrm{even}}, X^{\mathrm{odd}}) = \frac{1}{2} \log \left( \frac{1}{|\mathbf{\Sigma}|} \right),$$

where we indicate with $|\mathbf{\Sigma}|$ the determinant of the covariance matrix of $q$. Since $\mathbf{\Sigma}$ is a block diagonal, we have:

$$|\mathbf{\Sigma}| = |\mathbf{\Sigma}^{(b)}|^{\frac{d}{2}},$$

where $\mathbf{\Sigma}^{(b)}$ is a single block. Hence, the actual MI depends on the dimensionality of the data (see Tab. 6 for the ground truth values). Here, we use a rounded version of the real MI as a ground truth for our experiments, similar to prior work.

*Table 6.* The real MI values for different data dimensions.

| $d$ | MI |
|---|---|
| 20 | 5 |
| 40 | 10 |
| 80 | 20 |
| 160 | 40 |
| 320 | 80 |

**Dataset and Evaluation.** We consider a dataset of 100,000 data points drawn independently from both distributions. Out of these, we leave out 10,000 samples from $q$ as a test set. The models are evaluated based on the Mean Absolute Error (MAE) between the average $\log$-likelihood ratio estimated on the samples from $q$ and the ground truth MI.

**Choices for scRatio.** We train scRatio as in App. D.2, considering a smaller grid for hyperparameter tuning that focuses on scheduling (see Tab. 7). Additionally, in this experiment, we leverage S2 from Sec. 4.2 and set $b_t(\boldsymbol{x})$ from Prop. 4.1 to $b_t(\boldsymbol{x}) := u_t^\theta(\boldsymbol{x}_t)$, namely the field generating the unconditional distribution $p_t^\theta(\boldsymbol{x}_t)$, as we empirically find that it improves the results.

*Table 7.* The hyperparameter grid explored for the MI estimation experiment.

| Hyperparameters | Values |
|---|---|
| $\lambda$ | 0, 0.1, 0.25, 0.5 |
| $\sigma_{\min}$ | 0, 0.01, 0.1 |

**Baselines.** We choose TSM and CTSM with SB paths and optimize them as in App. D.2. Additionally, we add a simple Multi-Layer Perceptron (MLP) classifier baseline, where the likelihood ratio is approximated by the $\log$-odds of a classifier. While the $\log$-odds do not normally correspond to exact likelihood ratios, we argue that they represent a reasonable approximation in this setting, as the numerator and denominator distributions are equally sampled.

### D.4. Differential Abundance Experiment

**Task.** In this experiment, we show how to use scRatio for the Differential Abundance (DA) estimation task. DA is a standard analysis in single-cell perturbation studies that estimates whether a cellular state is significantly enriched in a condition (e.g., a donor or a perturbation) compared to another.

**Dataset.** We provide a description of the data in Sec. 5.3. In essence, we create a synthetic version of the popular PBMC68k dataset (Zheng et al., 2023) by assigning different proportions of synthetic treatment labels to four different clusters, derived with the Leiden algorithm (Traag et al., 2019) with a resolution parameter of 0.4. By varying the parameter $a$ in Eq. (14), we create more or less differentially abundant clusters for treatments 2 and 3, while we introduce no DA for clusters 1 and 4 (see Fig. 6).

**Metrics.** Since we do not have a ground truth, our metrics assess the ability of detecting differential abundance and the absence thereof at different levels of the parameter $a$. All the metrics quantify the extent of DA continuously, meaning that their value should vary for different choices of $a$. We describe the metrics in Sec. 5.3. For each of such metrics, we present:

- Mean value for $a \geq 0.3$: This suggests whether the metrics are promising in the presence of high DA.

- The Spearman correlation between the metric and the value of $a$. Metrics are expected to calibrate with the level of DA introduced. Hence, we expect a high correlation between their values and $a$.

In these experiments, we evaluate the ratio predictions directly on the dataset used for model training, as a baseline like MELD does not enable generalization.

**Experimental setup for scRatio.** We train scRatio for 100,000 steps with a learning rate of 1e-4. As a representation for gene expression, we use the top-$k$ Principal Components (PCs) for different values of $k$. We explore the hyperparameters on a grid presented in Tab. 8 and evaluate model optimality based on the Mean AUC metric (see Tab. 2).

*Table 8.* The hyperparameter grid explored for the DA estimation experiment on the PBMC68k dataset.

| Hyperparameters | Grid |
|---|---|
| $\lambda$ | 0, 0.01, 0.1, 0.25, 0.5 |
| $\sigma_{\min}$ | 0, 0.01, 0.1 |
| Batch size | 256, 512,1024 |
| Top-k PCs | 10, 20, 30, 40, 50 |

**Baselines.** We consider the following baseline models developed specifically for the task. Although following different approaches, all the methods produce a score that can be interpreted as a $\log$-likelihood ratio between the densities conditioned on different treatment labels.

- **MrVI** (Boyeau et al., 2025): MrVI is a disentangled Variational Autoencoder (VAE) compatible with DA analysis in single-cell data. It learns a low-dimensional latent representation in which biological and technical sources of variation are separated, while conditioning on experimental covariates. DA is assessed through variational posterior approximations of cell-type or neighborhood-specific abundances across conditions. We train the model across 200 epochs with default parameters.

- **MELD** (Burkhardt et al., 2021): MELD estimates differential abundance by smoothing condition labels over a kNN

graph built from single-cell embeddings. It computes a continuous, cell-level density estimate for each condition via local averaging and derives DA scores by comparing these smoothed densities. We ran the model with default parameters.

**Notes on advantages of scRatio over competing models.** We present some advantages of modeling likelihood ratios with scRatio compared to baseline models.

- **Exact likelihood estimation:** Using generative CNFs, our model enables numerically tractable likelihood-ratio estimation via ODE integration, while all other methods rely on heuristic approaches (MrVI) or neighborhood-based approximations (MELD).

- **Flexibility:** Because it relies on conditional vector fields, scRatio can model flexible likelihood ratios based on the conditioning variables used during training. This allows tailoring the estimated likelihood ratio to a flexible research question, such as combining different hierarchies of variables or estimating batch-specific effects.

- **Generalization.** Our deep generative model is a proxy for future generalization studies, while models like MELD produce graph-smoothed density ratio scores evaluated on the observed dataset, rather than likelihood-based generative estimates.

### D.5. Batch Correction Experiment

**Task.** In Sec. 5.4 we show that likelihood ratios reveal the presence of batch effect. Let $y$ and $b$ be cell type and batch labels and $\boldsymbol{X} = \{\boldsymbol{x}^{(i)}, y^{(i)}, b^{(i)}\}_{i=1}^N$ a single-cell dataset of $N$ observations. Furthermore, let $\{p_t^\theta(\cdot \mid y, b)\}_{t \in [0,1]}$ and $\{p_t^\theta(\cdot \mid y)\}_{t \in [0,1]}$ be doubly- and singly-conditional time-resolved densities generated by the neural vector fields $\{u_t^\theta(\cdot \mid y, b)\}_{t \in [0,1]}$ and $\{u_t^\theta(\cdot \mid y, \varnothing)\}_{t \in [0,1]}$ (Ho & Salimans, 2021). We use Prop. 4.1 to estimate the following ratio:

$$\log r_1^\theta\Big(\boldsymbol{x}^{(i)} \mid (y^{(i)}, b^{(i)}), y^{(i)}\Big) = \log \frac{p_1^\theta(\boldsymbol{x}^{(i)} \mid y^{(i)}, b^{(i)})}{p_1^\theta(\boldsymbol{x}^{(i)} \mid y^{(i)})} \; . \tag{78}$$

We distinguish two settings:

- **Presence of batch effect:** The ratio in Eq. (78) is nonzero, as adding the batch variable affects the conditional density around the data points.

- **Absence of batch effect:** The ratio in Eq. (78) is approximately 0, as adding information on the batch does not increase the likelihood of the data. In other words, the batch covariate does not add information beyond the annotated cell type.

In Sec. 5.4 we propose using this notion to evaluate the effectiveness of batch correction. Given an arbitrary batch correction function $f(\boldsymbol{x}, y, b)$, we evaluate the following ratios:

$$\log r_1^\theta\Big(\boldsymbol{x}^{(i)} \mid (y^{(i)}, b^{(i)}), y^{(i)}\Big), \quad \log r_1^\theta\Big(f(\boldsymbol{x}^{(i)}, y^{(i)}, b^{(i)}) \mid (y^{(i)}, b^{(i)}), y^{(i)}\Big) \; .$$

If $\log r_1^\theta\big(f(\boldsymbol{x}^{(i)}, y^{(i)}, b^{(i)}) \mid (y^{(i)}, b^{(i)}), y^{(i)}\big)$ drops considerably towards 0 compared to $\log r_1^\theta\big(\boldsymbol{x}^{(i)} \mid (y^{(i)}, b^{(i)}), y^{(i)}\big)$, a substantial extent of batch information has been removed.

**Datasets.** For this analysis, we choose two datasets holding both technical and biological annotations:

- NeurIPS (Luecken et al., 2021): 90,261 bone marrow cells from 12 healthy donors.
- *C. Elegans* (Packer et al., 2019): 89,701 cells across 7 sources.

As a preprocessing step, we normalize the counts to sum to $10,000$, $\log$-transform them, and perform dimensionality reduction by taking the first 50 PCs of the data (Heumos et al., 2023).

**Batch correction tool.** As a batch correction tool we use scVI (Lopez et al., 2018) with default hyperparameters.

**Evaluation.** Since this is a qualitative experiment, we limit ourselves to the empirical evaluation of how the likelihood ratios change when training scRatio on cellular representations before and after batch correction. In the former case, we use the top-50 PCs as a representation. For the latter, we consider a batch-corrected 50-dimensional latent space predicted by scVI.

**Hyperparameters.** We focus on the flow-matching formulation with deterministic paths between noise and data samples and $\sigma_{\min} = 0.01$. Moreover, we train a doubly-conditional model, a singly-conditional model, and an unconditional

counterpart by randomly replacing conditioning variables $y$ and $b$ with a null token $\varnothing$ with probability $0.2$. We train the model for $100{,}000$ steps and take the absolute value of the predicted ratios, as their distance from zero semantically reflects the presence of a batch effect (see Fig. 3 and Fig. 14).

### D.6. Drug Combination Experiment

In this experiment, we demonstrate how scRatio can be used to score combinations of drugs. More specifically, we assess whether treating cells with two perturbations induces a state shift compared to treatment with a single perturbation.

**Task.** We propose a formulation analogous to the batch correction task (see Sec. 5.4 and App. D.5), but instead of considering batch and cell type labels, we consider combinations of drug labels. In analogy with Eq. (78), we define two drug conditions, denoted by $d_1$ and $d_2$, and estimate the following ratio:

$$\log r_1^\theta\Big(\boldsymbol{x}^{(i)} \mid (d_1^{(i)}, d_2^{(i)}), d_1^{(i)}\Big) = \log \frac{p_t^\theta\Big(\boldsymbol{x}^{(i)} \mid d_1^{(i)}, d_2^{(i)}\Big)}{p_t^\theta\Big(\boldsymbol{x}^{(i)} \mid d_1^{(i)}\Big)} \; . \tag{79}$$

In other words, we assess whether a cell is more likely under a distribution conditioned on both drugs than under a distribution conditioned only on the first. From an application perspective, this corresponds to evaluating the distributional shift induced by a drug combination and enables hypothesis generation regarding synergistic effects and chemical interactions. A strong combinatorial effect corresponds to a large absolute $\log$-likelihood ratio, whereas its absence corresponds to ratios concentrated around zero.

**Dataset.** For this experiment, we use the ComboSciPlex dataset (Srivatsan et al., 2020), which contains $63{,}378$ cells treated with combinations of drugs. We investigate whether combining Alvespimycin, Dacinostat, Dasatinib, Givinostat, Panobinostat, or SRT2104 with $> 20$ additional drugs induces a significant state shift compared to treatment with each drug alone.

**Experimental setup.** We estimate Eq. (79) using scRatio. To approximate the denominator, we use a doubly conditional model combining a drug label with a control label, which conceptually corresponds to the absence of a second perturbation. We train scRatio for $200{,}000$ steps in its deterministic variant, using a Gaussian distribution around the data ($\lambda = 0$ and $\sigma_{\min} = 0.001$), with a batch size of $512$. As a data representation, we use $5$-dimensional autoencoder embeddings.

**Evaluation.** As is common for real-world datasets, no ground-truth likelihood ratios are available. Since we use this experiment as a qualitative case study, we provide two complementary analyses:

- **Qualitative analysis:** For reported active perturbation combinations, we show that the estimated ratios attain large absolute values, whereas for inactive double perturbations, the ratios remain close to zero. This qualitative validation is available in App. F.9.
- **Correlation with classification probability:** We show that the estimated absolute $\log$ ratios are well calibrated with the binary classification absolute $\log$-odds of a deep classifier trained to discriminate between singly and doubly perturbed cells.

### D.7. Patient-specific Treatment Response Experiment

**Task.** In this experiment, we demonstrate how scRatio can be used to quantify patient-specific responses to perturbations. More specifically, we assess whether cytokine stimulation induces a donor-dependent state shift compared to the corresponding control condition.

In analogy to the drug combination experiment (see App. D.6), we formulate a conditional likelihood ratio comparing treated and control cells within the same donor. Let $s$ denote a donor (sample) label and $d$ a treatment, where $d^k$ indexes cytokines and $d^c$ represents the control condition. For a perturbed cell $\boldsymbol{x}^{(i)}$ from donor $s^{(i)}$ treated with cytokine $d^{(i)}$, we estimate

$$\log r_1^\theta\Big(\boldsymbol{x}^{(i)} \mid (s^{(i)}, d^{(i)}), (s^{(i)}, d^c)\Big) = \log \frac{p_1^\theta\big(\boldsymbol{x}^{(i)} \mid s^{(i)}, d^{(i)}\big)}{p_1^\theta\big(\boldsymbol{x}^{(i)} \mid s^{(i)}, d^c\big)} \; . \tag{80}$$

In other words, we evaluate whether a treated cell is more likely under the donor-specific treatment distribution than under the donor-specific control distribution. A strong donor-specific treatment effect corresponds to large absolute $\log$-likelihood ratios, whereas the absence of a differential response results in ratios concentrated around zero.

**Dataset.** We use a large-scale PBMC perturbation dataset comprising approximately 10 million cells from 12 donors (Oesinghaus et al., 2025). For each donor, cells are profiled in a control condition and under stimulation with 90 different cytokines. The dataset includes previously reported donor groupings with cytokine-specific differential responses, enabling qualitative validation of patient-specific effects.

**Experimental setup.** We train scRatio to model conditional distributions over cell states given donor and treatment labels. As in previous experiments, we use the deterministic variant of scRatio with a Gaussian distribution around the data ($\lambda = 0$ and $\sigma_{\min} = 0.001$). Training is performed for 100,000 steps with a batch size of 512. Cells are represented using low-dimensional autoencoder embeddings to capture their transcriptional state.

To estimate Eq. (80), we condition the numerator on both donor and cytokine labels, and the denominator on the same donor paired with the control label. This design ensures that estimated likelihood ratios capture treatment-induced shifts while controlling for inter-donor variability.

**Evaluation.** As no ground-truth likelihood ratios are available, we conduct a qualitative evaluation guided by biological findings reported in Oesinghaus et al. (2025). Specifically, we analyze cytokines that were shown to induce group-specific responses (e.g., IL-10 and IFN-omega), as well as cytokines with weak or similar effects across donor groups (e.g., APRIL). For each donor-treatment pair, we examine the distribution of estimated log-likelihood ratios for treated cells relative to controls. Consistency between reported biological differences and the magnitude and separation of likelihood ratio distributions serves as validation that scRatio captures meaningful patient-specific treatment effects.

# E. Algorithms

We provide an algorithmic description of the scRatio inference procedure here. For simplicity, we consider trajectories simulated under the numerator field. Note that, when evaluating functions on collections of objects, we override the notation to emphasize the vectorizability of the operation and assume the returned object is the same collection mapped with the used function.

---

**Algorithm 1** Inference with scRatio on population with label $\boldsymbol{y}$.

---

1: **Input:** Trained Neural Velocity Field $u_t^\theta(\cdot \mid Y)$ and Score $s_t^\psi(\cdot \mid Y)$. Two realizations $\boldsymbol{y}$ and $\boldsymbol{y}'$ with $\boldsymbol{y} \neq \boldsymbol{y}'$ of conditioning random variable $Y$. Set of data-points $\boldsymbol{X}_1 := \{(\boldsymbol{x}_1^{(i)}, \boldsymbol{y})\}_{i=1}^N$, with $\boldsymbol{x}_1^{(i)} \sim q(\cdot \mid Y = \boldsymbol{y})$.

2: **Output:** Estimated log-likelihood ratio $\log \frac{p^\theta(\cdot|Y=\boldsymbol{y})}{p^\theta(\cdot|Y=\boldsymbol{y}')}$

3: $\frac{\mathrm{d}}{\mathrm{d}t} r_t^\theta(\boldsymbol{x}_t^{(i)} \mid \boldsymbol{y}, \boldsymbol{y}') \leftarrow \nabla_{\boldsymbol{x}_t^{(i)}} \cdot \left( u_t^\theta(\boldsymbol{x}_t^{(i)} \mid Y = \boldsymbol{y}') - u_t^\theta(\boldsymbol{x}_t^{(i)} \mid Y = \boldsymbol{y}) \right) +$

     $+ \left( u_t^\theta(\boldsymbol{x}_t^{(i)} \mid Y = \boldsymbol{y}') - u_t^\theta(\boldsymbol{x}_t^{(i)} \mid Y = \boldsymbol{y}) \right)^\top s_t^\psi(\boldsymbol{x}_t^{(i)} \mid Y = \boldsymbol{y}')$      // Define ratio dynamics to estimate Eq. (9).

4: $\hat{b}_t(\boldsymbol{x}_t^{(i)} \mid \boldsymbol{y}, \boldsymbol{y}') \leftarrow [u_t^\theta(\boldsymbol{x}_t^{(i)} \mid Y = \boldsymbol{y}), \frac{\mathrm{d}}{\mathrm{d}t} r_t^\theta(\boldsymbol{x}_t^{(i)} \mid \boldsymbol{y}, \boldsymbol{y}')]$      // Define augmented dynamics.

5: $[\hat{\boldsymbol{X}}_0, \boldsymbol{r}_0] \leftarrow \mathrm{OdeSolve}([\boldsymbol{X}_1, \boldsymbol{0}], \hat{b}_t(\cdot \mid \boldsymbol{y}, \boldsymbol{y}'), t_0 = 1, t_1 = 0)$      // Solve ODE backwards in time.

6: **return** $-\boldsymbol{r}_0$      // Return log ratio.

---

# F. Additional Results

## F.1. Differential Abundance Experiments

A general depiction of the semi-synthetic PBMC68k datasets can be found in Fig. 6.

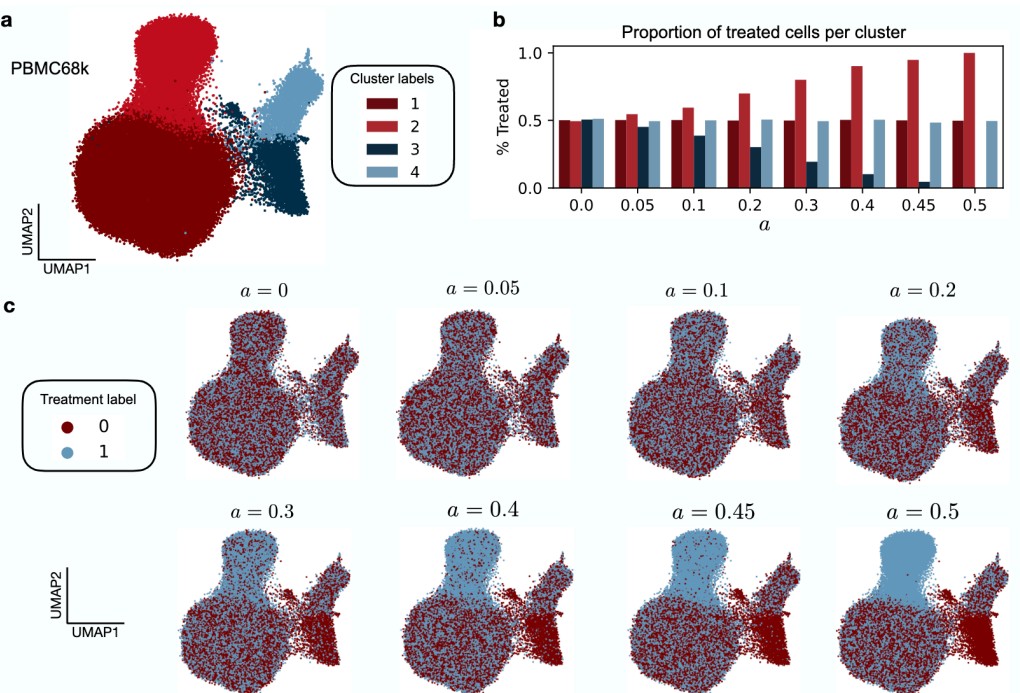

*Figure 6.* **a.** UMAP plot of the PBMC68K dataset colored by cluster labels. The clusters are assigned to each cell through the Leiden algorithm (Traag et al., 2019). **b.** The proportion of treated cells per cluster for different values of $a$ (see Eq. (14) for a description of $a$). **c.** UMAP plot colored by the treatment label sampled for each cluster using different values of $a$.

### F.2. Additional Results on Gaussian Simulation Experiments

We report the Mean Squared Error (MSE) between the true and estimated $\log$-ratios across different dimensions, averaged over 5 runs. Results in Fig. 7 show that scRatio outperforms all baselines for both locations 1 and 2 across all dimensions, except for location 1 at 2 dimensions, where TSM and the naive approach achieve comparable performance. Different scRatio schedules yield similar performance across dimensions.

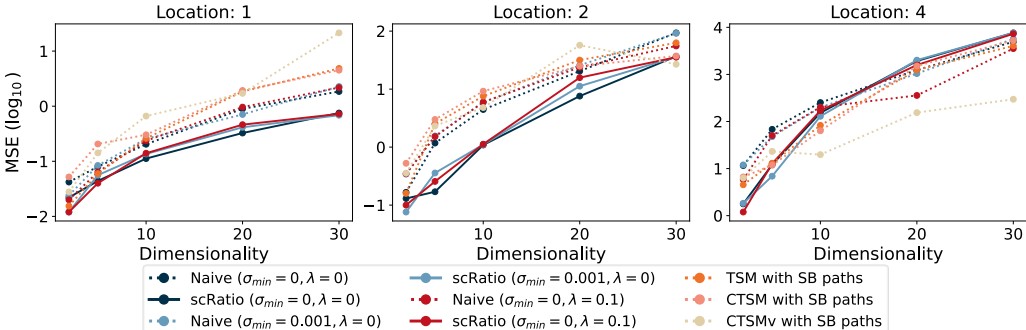

*Figure 7.* Mean Squared Error (MSE) of different methods across varying data dimensionalities and values of $s$ (mean shifts from the origin). TSM and CTSM denote baseline models, and SB paths indicate Schrödinger Bridge probability paths.

For location 4, however, we observe a different behavior, with some baselines outperforming scRatio. This can be attributed to the limited overlap between the distributions for which the ratio is estimated. Specifically, baselines such as TSM, CTSM, and CTSMv learn probability paths directly between these distributions, whereas scRatio learns probability paths separately from the source to each target distribution. The effect of distribution overlap as a function of the distance between Gaussian means is illustrated in Fig. 8, where we also add a classifier baseline similar to Sec. 5.2.

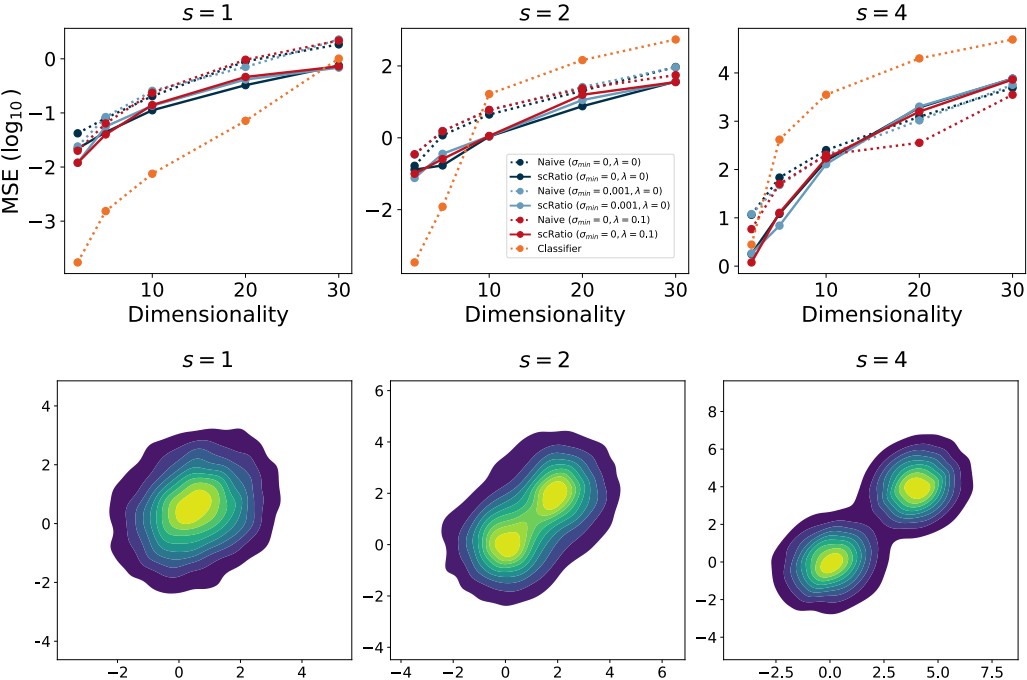

*Figure 8.* **a.** Comparison between scRatio and a standard classifier baseline on the task of predicting likelihood ratios between Gaussians (see Sec. 5.1). Performance is evaluated for different numbers of features (x-axis) as the MSE between ground truth and predicted ratios. **b.** Overlap between 2D Gaussian distributions as a function of the distance between their means. In all plots, one of the distributions is fixed as a standard Gaussian.

## F.3. Performance as a Function of Sample Complexity on Gaussian Simulations

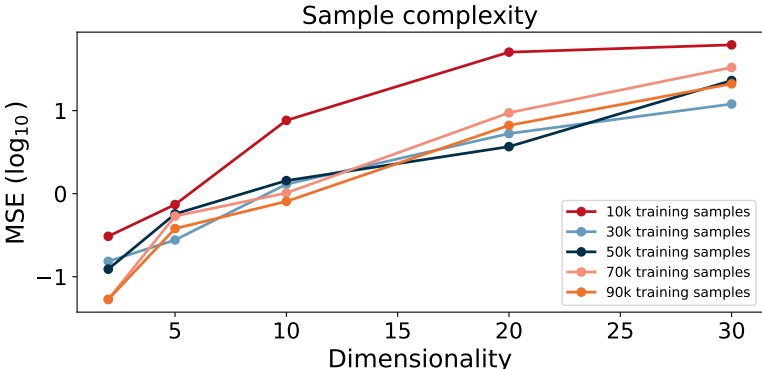

*Figure 9.* scRatio's MSE results for different levels of sample complexity on the Gaussian log-likelihood ratio estimation task. From Sec. 5.1 we use the harder setting with $s = 2$ and $d \in \{2, 5, 10, 20, 30\}$ (Yu et al., 2025). For the best hyperparameter configuration, we retrain the model with different numbers of training points and report the performance on the test set.

## F.4. Effect of Reparameterization on Gaussian Simulations

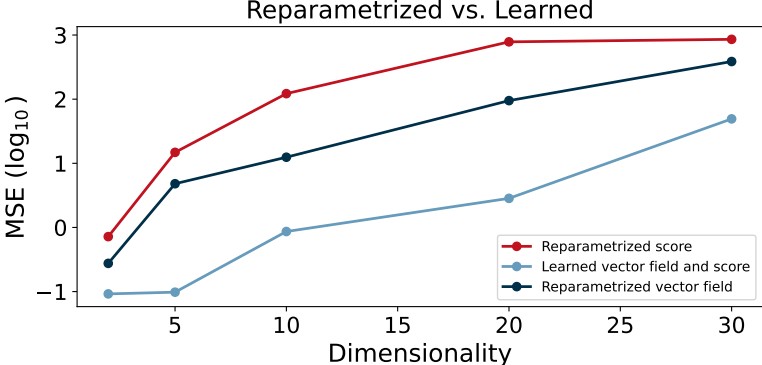

*Figure 10.* scRatio's MSE results for neural parameterization and closed-form reparameterization of the scores on the Gaussian log-likelihood ratio estimation task. From Sec. 5.1 we use the harder setting with $s = 2$ and $d \in \{2, 5, 10, 20, 30\}$ (Yu et al., 2025). For the best hyperparameter configuration in Sec. 5.1, we compare the performance of the model across dimensions between closed-form reparameterization and neural parameterization of the score and vector field in Prop. 4.1.

## F.5. MSE vs Ratios' Absolute Value on Gaussian simulations

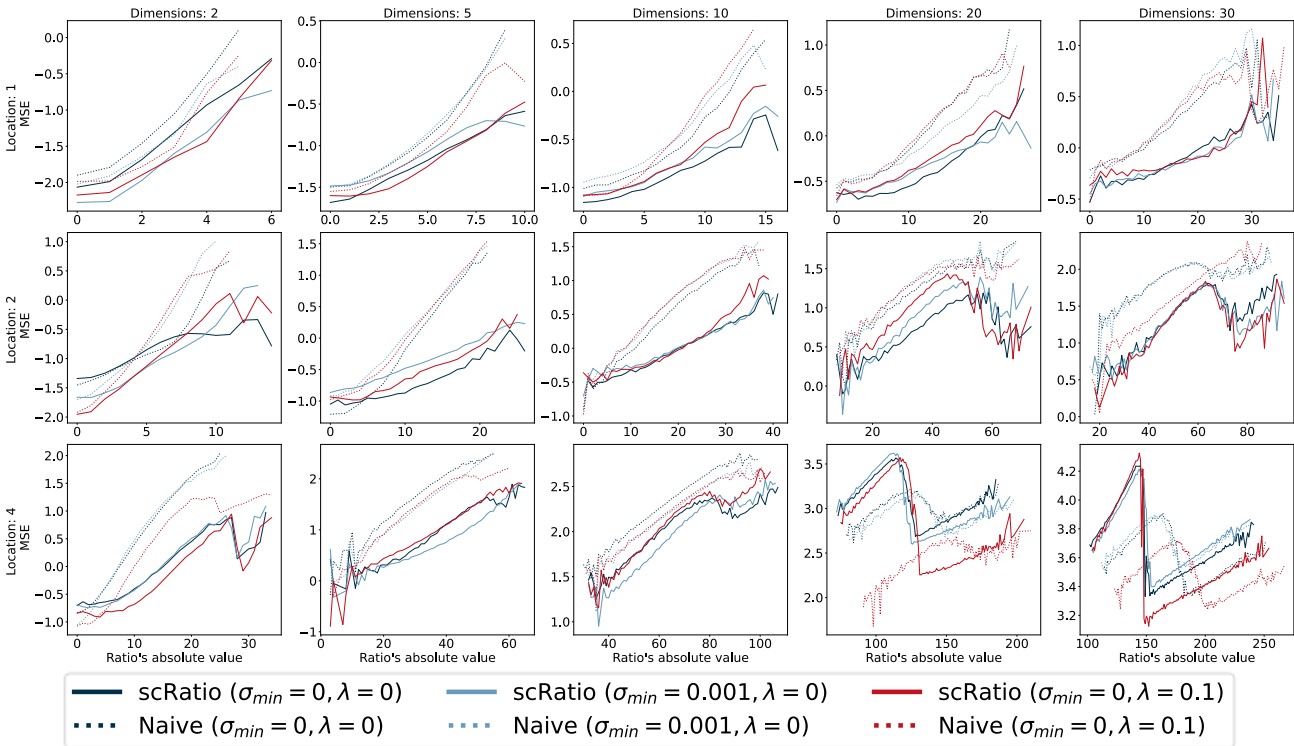

*Figure 11.* Mean Squared Error (MSE) as a function of the absolute value of the log-ratio, shown for different locations and dimensionalities.

## F.6. The Relationship Between Performance and Stochasticity on Gaussian Simulations

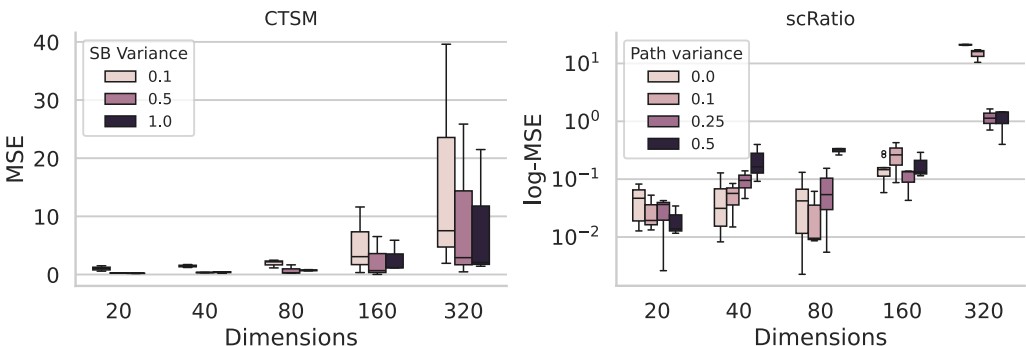

*Figure 12.* Test MSE as a function of dimensions and path variance for CTSM and scRatio across hyperparameter configurations. Boxplots represent the distribution of MSE (linear scale for CTSM, log-scale for scRatio) as a function of input dimensions. Boxes for both models are colored by the level of path stochasticity in the interpolants (Schrödinger Bridge and flow matching path variance, respectively).

## F.7. Effect of Score and Velocity Approximation Quality on Gaussian Simulations

**The impact of suboptimal neural approximation.** To assess the impact of the neural approximation quality on the log-likelihood ratio estimation, we conducted an ablation experiment on the velocity field $v_t^\theta$ and score $s_t^\psi$ networks. For this experiment, while evaluating over all the previously tested dimensions, we fixed the location to $s = 2$. To ensure compatible intermediate representations, we set both the state and the condition encoders to the identity function. Let $\mathcal{J} := \{1000, 5000, 10000, 50000, 100000\}$ denote the set of training step checkpoints. Then, for each number of training

steps $J \in \mathcal{J}$, we trained a pair of score and velocity fields, denoted by $(v_t^{\theta,J}, s_t^{\psi,J})$. We then estimated the ratio for $(J, J') \in \mathcal{J} \times \mathcal{J}$, by combining the neural networks $(v_t^{\theta,J}, s_t^{\psi,J'})$ and reported the Mean Squared Error (MSE) against the ground-truth ratio. This cross-evaluation setup allows us to pinpoint the impact of the degraded approximation of $v_t^{\theta}$ and $s_t^{\psi}$ both jointly and individually.

Our results show that poorer neural approximations reduce ratio accuracy. Performance is generally more sensitive to the velocity field than the score, suggesting the latter could be trained for fewer steps. In higher dimensions, the score accuracy becomes more relevant.

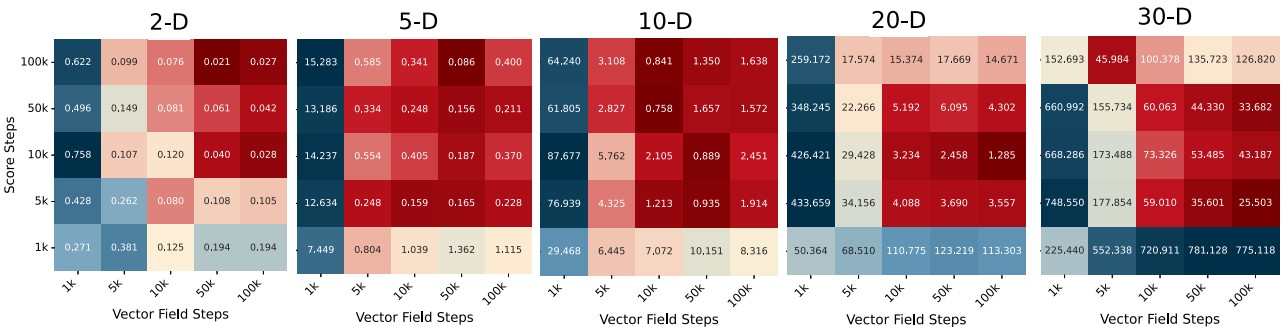

*Figure 13.* Heatmap of Mean Squared Error per number of training steps for velocity $u_t^{\theta}$ and score $s_t^{\psi}$ fields. On the $x$-axis the number of steps used to train the velocity field $u_t^{\theta}$ is presented, the $y$-axes shows the number of steps for the score field $s_t^{\psi}$, while the hue value encodes the MSE attained by the combination of the two.

## F.8. Additional Results on Batch Correction Evaluation

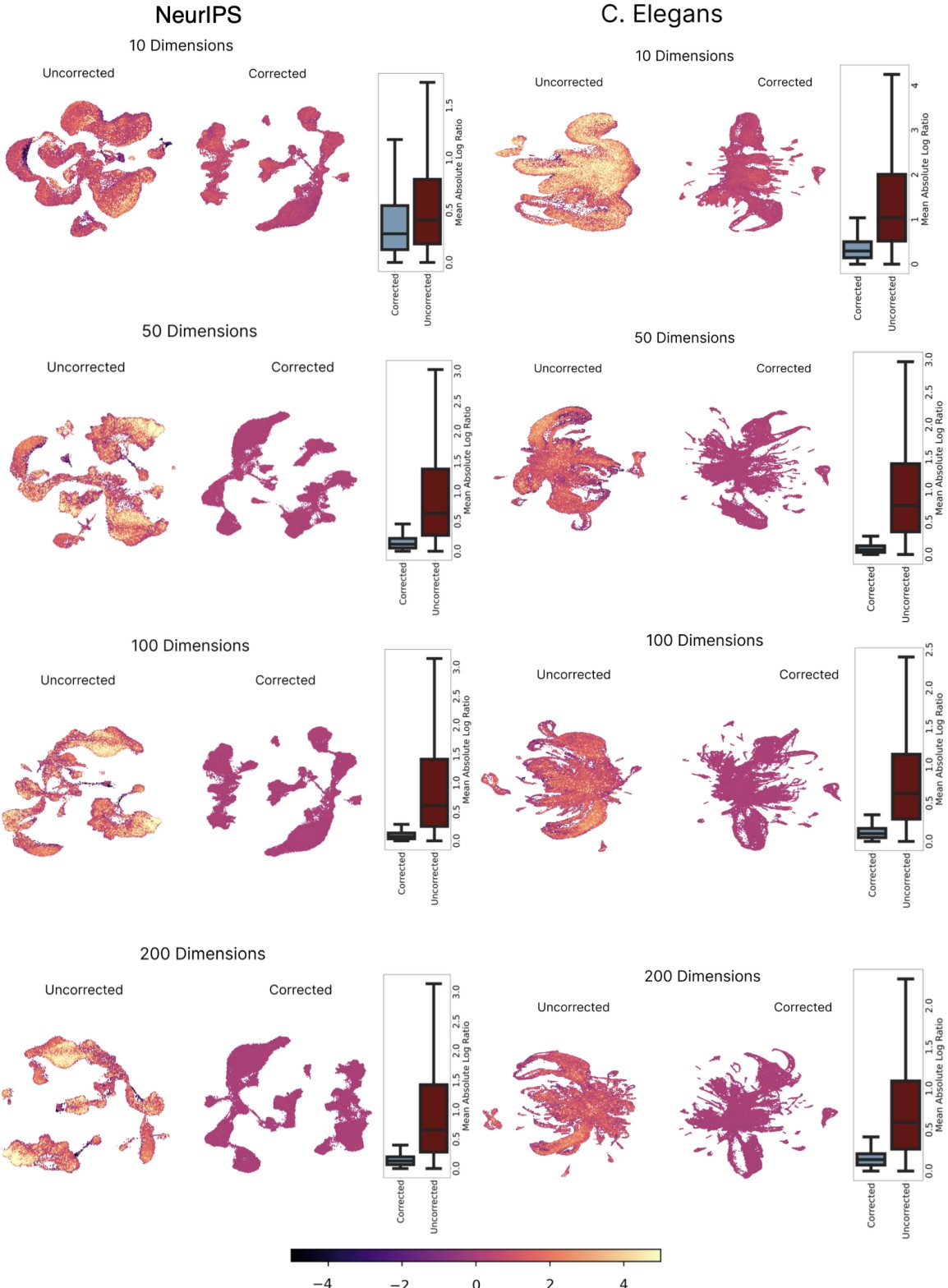

*Figure 14.* UMAP projections for the two datasets across different dimensionalities (10, 50, 100, and 200). Each cell is colored by the estimated log-likelihood ratio, computed as described in App. D.5. Beside each UMAP, the corresponding box plot shows the distribution of the mean absolute log-ratio per condition combination.

## F.9. Additional Results on Combinatorial Drug Estimation

We present additional plots in Fig. 15 to illustrate how the estimated log-ratio changes as the combinatorial effect increases. Starting with the first UMAP, which has no combinatorial effect, and progressing to those with increasingly pronounced effects, the box plots clearly show the corresponding rise in the estimated log-ratio.

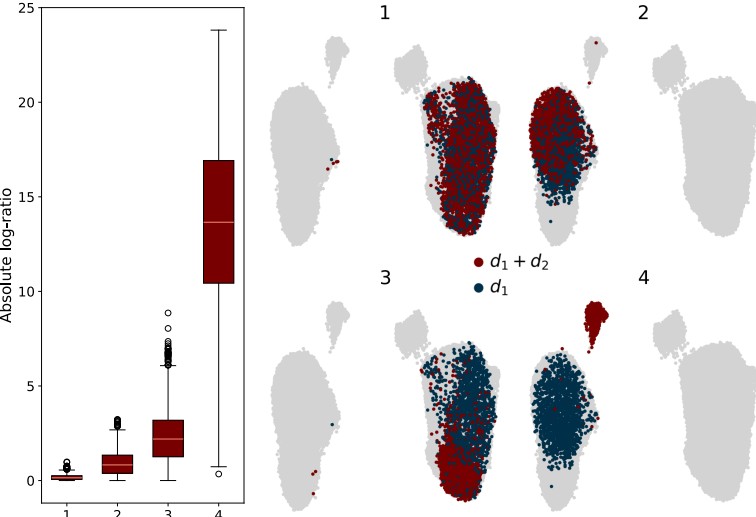

*Figure 15.* Right: Four UMAP plots showing the gradual increase of combinatorial effects across several single perturbations and combinations thereof. Left: Corresponding box plots for each UMAP, displaying the absolute values of the estimated log-likelihood ratios.

## F.10. Ablation Results on the Gaussian Simulation Experiments

*Table 9.* Sweep results on Gaussian simulation experiments for different noise schedules. $\lambda$ controls the stochasticity along conditional probability paths from noise to data, while $\sigma_{\min}$ is the variance of Gaussian noise around data points. The $\alpha_t$ parameter is kept fixed to $\alpha_t = t$, with $t \in [0, 1]$. MSE results for different shifts from the origin ($s$) are averaged across three training replicates. Best results are highlighted per dimensionality.

| $d$ | $\lambda$ | $\sigma_{\min}$ | MSE ($s = 1.0$) | MSE ($s = 2.0$) |
|---|---|---|---|---|
| 2 | 0.0 | 0.0 | **0.01** | **0.05** |
|   |     | 0.0001 | **0.01** | **0.05** |
|   |     | 0.001 | **0.01** | 0.13 |
|   |     | 0.01 | **0.01** | **0.05** |
|   |     | 0.1 | 0.02 | 0.17 |
|   | 0.10 | 0.0 | 0.02 | 0.07 |
|   | 0.25 | 0.0 | 0.02 | 0.16 |
|   | 0.50 | 0.0 | 0.02 | 0.08 |
|   | 0.75 | 0.0 | 0.02 | 0.10 |
|   | 1.00 | 0.0 | **0.01** | 0.19 |
| 5 | 0.0 | 0.0 | **0.04** | 0.38 |
|   |     | 0.0001 | 0.06 | 0.19 |
|   |     | 0.001 | 0.08 | 0.29 |
|   |     | 0.01 | 0.05 | 0.39 |
|   |     | 0.1 | 0.05 | **0.17** |
|   | 0.10 | 0.0 | 0.05 | 0.26 |
|   | 0.25 | 0.0 | 0.06 | 0.41 |
|   | 0.50 | 0.0 | 0.06 | 0.59 |
|   | 0.75 | 0.0 | 0.10 | 0.97 |
|   | 1.00 | 0.0 | 0.20 | 0.94 |
| 10 | 0.0 | 0.0 | 0.18 | 0.81 |
|   |     | 0.0001 | 0.17 | 0.88 |
|   |     | 0.001 | 0.12 | **0.67** |
|   |     | 0.01 | 0.15 | 0.87 |
|   |     | 0.1 | **0.11** | 1.32 |
|   | 0.10 | 0.0 | 0.13 | 1.42 |
|   | 0.25 | 0.0 | 0.17 | 1.33 |
|   | 0.50 | 0.0 | 0.18 | 2.73 |
|   | 0.75 | 0.0 | 0.12 | 6.93 |
|   | 1.00 | 0.0 | 0.21 | 6.79 |
| 20 | 0.0 | 0.0 | 0.64 | **6.64** |
|   |     | 0.0001 | **0.36** | 10.32 |
|   |     | 0.001 | 0.39 | 11.17 |
|   |     | 0.01 | 0.57 | 10.89 |
|   |     | 0.1 | 0.74 | 14.66 |
|   | 0.10 | 0.0 | 0.42 | 12.61 |
|   | 0.25 | 0.0 | 0.49 | 17.75 |
|   | 0.50 | 0.0 | 1.17 | 39.93 |
|   | 0.75 | 0.0 | 0.72 | 43.47 |
|   | 1.00 | 0.0 | 0.63 | 48.24 |
| 30 | 0.0 | 0.0 | **0.69** | **21.03** |
|   |     | 0.0001 | 0.86 | 44.19 |
|   |     | 0.001 | 0.77 | 32.69 |
|   |     | 0.01 | 0.79 | 41.02 |
|   |     | 0.1 | 0.86 | 55.97 |
|   | 0.10 | 0.0 | 0.86 | 38.67 |
|   | 0.25 | 0.0 | 0.82 | 81.42 |
|   | 0.50 | 0.0 | 1.59 | 121.23 |
|   | 0.75 | 0.0 | 2.09 | 82.06 |
|   | 1.00 | 0.0 | 1.41 | 82.14 |

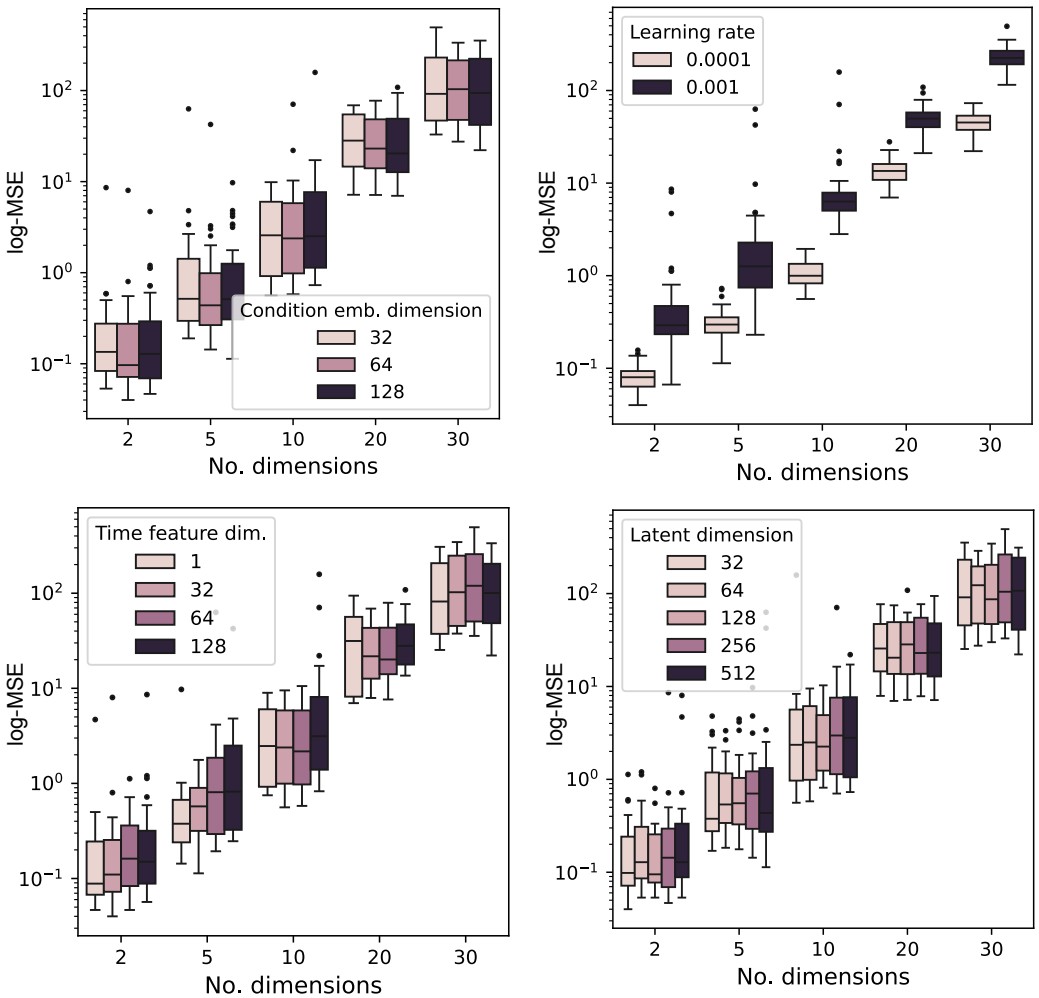

*Figure 16.* Performance as a function of number of features for different model hyperparameters (number of condition embedding dimensions, learning rate, time feature size, number of latent dimensions). Evaluations are conducted on Gaussian data from Sec. 5.1 with $s = 2$. The performance quality is measured through the MSE between ground-truth and estimated likelihood ratios.

