# OpenReview forum: "Flow-Based Density Ratio Estimation for Intractable Distributions with Applications in Genomics"
_ICML.cc/2026/Conference — ICML 2026 regular_

### Official Review · Reviewer_h3MX · 2026-03-03

**Soundness:** 2
**Presentation:** 3
**Significance:** 3
**Originality:** 2
**Overall Recommendation:** 4
**Confidence:** 4

**Summary:**

This paper studies nonparametric density ratio estimation using conditional flow-based models, and applications of density ratio estimation in genomics, such as differential abundance analysis and batch correction. In particular, as is standard for conditional flow-based models, the authors consider time-varying conditional distributions $p_t(x | y)$, and seek to compute $p_1(x | y)/p_1(x | y')$ for distinct conditions $y$ and $y'$. As input, they assume one has already estimated conditional vector fields $u_t(x | y)$ and $u_t(x | y')$ and conditional score functions $s_t(x | y)$ and $s_t(x | y')$ that generate $p_t(x | y)$. Their main insight (Proposition 4.1) shows that the log of the likelihood ratio $r_t(x) = p_t(x | y) / p_t(x | y')$ has a closed-form time derivative in terms of these estimated functions and an arbitrary velocity field (with the numerator's velocity field or the unconditional velocity field suggested as canonical choices). Their proposed algorithm integrates this derivative to return the log ratio.

In their experiments, they compare to recent density ratio estimation methods such as time score matching (TSM) and conditional time score matching (CTSM), with a particular choice of stochastic interpolation between the numerator and denominator distributions. In toy experiments, they follow previous papers by evaluating on high-dimensional multivariate Gaussian distributions, for which the ground truth log likelihood ratio and other quantities such as mutual information can be computed in closed form. Across different noise schedules, their method (with optimized hyperparameters) outperforms the baselines, with reduced runtime compared to the "naive" baseline of first computing the likelihoods, then dividing. In semi-synthetic experiments, they demonstrate their method on differential abundance estimation, batch correction, and drug combination effect estimation, against task-appropriate baselines (MrVI and MELD) and metrics, again demonstrating improved performance.

**Compliance With Llm Reviewing Policy:**

Affirmed.

**Final Justification:**

The author's initial rebuttal addressed my concerns about the soundness of the experimental results (by improving baseline optimization) and methodological motivations beyond computational improvement (e.g. numerical stability).

Their second response slightly alleviated my concerns about technical innovation, making a good case based on the application-motivated nature of the work. I appreciated the insights into the choice of the simulating field $b_t$, but would need to see these ideas further developed and better related to similar work on DRE using the "minimum path variance principle" [1], which appears closely related.

Moreover, I have some concerns about the statement "the ability to customize ratio formulations by training a single conditional model is a significant advantage in DRE, and, to our knowledge, marks a novel direction beyond re-training and interpolating between all pairs of densities", which points to a potential flaw in the overall motivation behind the method. In particular, the idea to solve DRE by training a single conditional flow matching model is essentially a proposal to bring us back to the "naive" DRE approach of separately estimating likelihoods (see [2] for why this approach is naive), modulo some benefits such as parameter sharing and some kind of "implicit bridging" that may be induced by the choice of $b_t$. However, properly making these connections and addressing these concerns would require substantial work.

Since these concerns cropped up only towards the end of the reviewing period and the authors did not have a chance to address them, I will still raise my score to 4. Overall, the paper does make a sufficiently novel contribution (when also accounting for the new insights on $b_t$) and the method performs well empirically. Thus, I would now consider it beyond the "accept" threshold, but I believe the technical insight/novelty would need to be more fully fleshed out for a higher score.


[1] Chen et al. (2026), "A Minimum Variance Path Principle for Accurate and Stable Score-Based Density Ratio Estimation"
\
[2] Sugiyama et al. (2010), "Density Ratio Estimation: A Comprehensive Review"

**Key Questions For Authors:**

1. **Computational improvement:** Is my assessment of the computational improvement mentioned in Weakness #1 correct? If so, is there a better motivation for the method besides computational improvement? A better understanding of the methodological advance could raise my score.
2. **Results for fair hyperparameter selection:** Can you report results for fair hyperparameter selection (see Weakness #2)? To evaluate the paper, it is critical to see these results and determine whether the better performance is solely due to hyperparameter selection, which could raise my score.

**Limitations:**

Yes

**Strengths And Weaknesses:**

### Strengths
1. **Strong motivation:** The problem of density ratio estimation is very fundamental, with implications for many downstream tasks. The authors provide a good review of these connections, along with a good review of compelling applications to genomics.
2. **Clear setup and presentation:** The mathematical notation and setup is quite clear and easy to follow, and the paper has a very logical progression.
3. **Extensive experiments:** I appreciate the complementary roles of the toy and semisynthetic experiments, and the thoughtful design of the semisynthetic experiments to reflect realistic scenarios while still allowing for insightful quantitative evaluation.
### Weaknesses
1. **Weak computational/methodological improvement:** To the best of my understanding, the computational improvement is only a constant factor at inference time (solving one ODE backward in time, instead of solving two). Moreover, this improvement is only present at inference time, whereas the main computational cost would typically be actually training the neural velocity fields and score functions. This improvement is based on a pretty straightfoward computation of the derivative of $\log r_t(x \mid y)$, so the overall methodological contribution seems fairly weak.
2. **Unfair hyperparameter selection:** The hyperparameters of the proposed method (scRatio) are selected for the best performance on the test set, whereas TSM and CTSM are forced to use fixed hyperparameters. For the experimental results to be fair, hyperparameter selection should be performed for all methods, and the hyperparameters should be selected on a validation set rather than on the test set.
### Minor Weaknesses
1. **Clarity of experimental setup:** I found the noise schedules confusing. To my understanding, there are two options, each with one hyperparameter: option (II) from Lipman et al., with hyperparameter $\sigma_{min}$, and option (III) from Tong et al., with hyperparameter $\lambda$. However, the experiments seem to be framed in terms of a single parameterization with two parameters, but always at least one hyperparameter is zero.

---

> ### Author Rebuttal · Authors · 2026-03-30
>
> We are thankful for the reviewer's feedback and hope this reply clarifies our contributions. New results are available [here](https://figshare.com/s/46bb00e542dcce432831) and referenced as R.x (x = fig. number) in the rebuttal.
>
> > A1. Contextualization.
>
> **Computational improvement**
>
> Indeed, the computational improvement corresponds to halving the inference complexity. Crucially, this speed-up also benefits *approximation quality*.
>
> ODE simulation requires numerical discretization, which introduces errors that accumulate non-linearly over integration. By halving the number of required ODE solves, we reduce this source of discretization error, improving approximation accuracy. This effect is empirically confirmed by comparisons with the Naive method in the synthetic experiments.
>
> **Inference focus**
>
> This is an important point. Flow matching and diffusion models are simulation-free generative methods that cast training as a *cheap regression task*. The flow matching algorithm samples from two distributions, computes a simple linear interpolation, performs a forward pass through a neural velocity field, and updates the model. Since expensive optimal transport is not required to generate from a prior, training is efficient by design. Training scRatio is no more expensive than optimizing a conditional flow matching model.
>
> Conversely, inference is a major bottleneck, as it requires ODE integrations on high-dimensional data, where more discretization steps yield more accurate solutions. Current research, therefore, focuses on improving inference efficiency and accuracy. A partial list:
>
> 1. **Flow maps/consistency models.** Methods that learn one-step approximations of the generative integral [1].
> 2. **Trajectory rectification.** Approaches like Rectified Flow [2] iteratively straighten ODE trajectories, so fewer discretization steps are needed.
> 3. **ODE solvers.** Models like DPM-Solver design high-order solvers tailored to diffusion and flow ODEs [3].
>
> Note that:
>
> * scRatio extends these research directions to likelihood ratio estimation, which has not been explicitly addressed in the generative flow literature.
> * While a 2x speed-up is constant, it is practically important for likelihood computation on large datasets (e.g., emerging 100M-cell perturbation datasets), where inference can take days.
>
> **Technical innovation.**
>
> Estimating the ratio between conserved densities along a generative trajectory has not been explored before. Prior work has formalized tracking a time-resolved density while sampling from another [4], but none uses this to define a dynamical comparison between likelihoods with the same generative prior.
>
> Viewed through the lens of generative likelihood ratio estimation, our approach provides an alternative to recent diffusion-based TSM/CTSM methods (see L894-923 in the paper). Beyond exact likelihood estimation via flows, a key advantage is that scRatio requires training a *single model* once: different conditioning variables allow estimation of any pairwise ratio of intractable distributions. In contrast, TSM, CTSM, and related methods require retraining a model for each pair of likelihoods.
>
> > A2. Baseline optimization
>
> Apologies. We used baseline learning rates (lr) from the official repositories but overlooked tuning the Schrödinger Bridge (SB) variance. Note that authors in [6] only tuned the lr hyperparameter.
>
> To address this, we ran experiments with broader optimization for both scRatio and the baselines, tuning SB variance and lr, and performing systematic sweeps on the synthetic benchmarks.
>
> *Procedure*
>
> For all models, we sample 90k training, 10k validation (model selection), and 10k test points.
>
> *Baseline hparams*
>
> * lr = [5e−4, 1e−3, 5e−3] (tuned as in [5])
> * SB var = [0.1, 0.5, 1.0]
>
> *scRatio hparams*
>
> In addition to scheduling, we also sweep lr over the same baseline range.
>
> **Results**
>
> * **R.3**, Gaussian simulation results. Trends match the original figure.
> * **R.2**, MI estimation table. Baselines improve after tuning, but scRatio variants still outperform TSM and CTSM in most settings.
>
> > A3. Noise schedules
>
> These two parameterizations are not mutually exclusive. As detailed in the App. A.4, $\sigma_{\min}$ sets the noise at the data ($t=1$), while $\lambda$ controls noise along the probability paths, defining stochastic interpolants.
>
> One could combine both to have stochastic paths and noise in the data. However, we focus on three settings as the main flow matching scheduling choices:
>
> 1) $\lambda=0$, $\sigma_{\min}=0$ (ref Tong et al, 2024a in paper).
> 2) $\lambda>0$, $\sigma_{\min}=0$ (ref Albergo et al, 2023 in paper).
> 3) $\lambda=0$, $\sigma_{\min}>0$ (ref Lipman et al., 2022 in paper).
>
> This choice reflects that combining the two parameters is uncommon in the literature. See **R.8** for the explored configurations and their performance.
>
> [1] arXiv.2511.22688
>
> [2] arXiv.2209.03003
>
> [3] arXiv.2206.00927
>
> [4] arXiv.2412.17762
>
> [5] arXiv.2502.02300

---

> > ### Author Rebuttal · Reviewer_h3MX · 2026-04-03
> >
> > Thank you for the thoughtful response.
> >
> > **Computational improvement:** Thank you for verifying my understanding about the change in inferential complexity and for giving more insight. The argument that this strategy also reduces *discretization* error is one that I had not thought about, and I think it will be useful to include in the paper. Also, the example of tasks where likelihood computation takes days (e.g. the 100M-cell perturbation datasets) is good motivation for even such a constant-factor improvement. This point was satisfactorily resolved.
> >
> > **Technical innovation:** I agree that using such models for density ratio estimation (DRE) has not directly been explored before (to the best of my knowledge). My point was that Proposition 4.1 itself is not a major contribution --- indeed, as the authors themselves point out in the paper and their response, this kind of insight is used in prior work on composing diffusion models. Thus, the choice to use this strategy for DRE seems more like a "first step", and reaching the level of "significant contribution" would require further insights, e.g. more principled choices of how to optimally choose the simulating field $b_t$.
> >
> > Hence, I feel my point about methodological novelty is only partially resolved. Beyond the use of Proposition 4.1, could the authors describe any other aspects of the paper which can be regarded as novel technical innovations?
> >
> > **Hyperparameter selection:** The new procedure for hyperparameter selection of the baselines is more fair, and I appreciate the authors addressing this point. It is reassuring to hear that the trends remain the same. Hence, this point is adequately resolved. I understand that there was limited time for the rebuttal, so the 3 x 3 grid is enough for this stage, but a larger grid would be preferable for the revision.
> >
> > **Noise schedules:** I agree that, hypothetically, the two parameterizations are not mutually exclusive, but that combining the two parameters (i.e., having both nonzero) is uncommon in the literature. My main point was about the presentation, which I found more confusing than necessary. Instead of defining three parameterizations in lines 153-156, could you just define a single joint parameterization with these three as special cases? The figures are anyways reflecting this joint parameterization (by using expression like $(\sigma_{min} = 0, \lambda = 0)$, so I'm just saying it would be cleaner/less confusing. The response did not indicate any proposed changes to the presentation, so this point is still unresolved but fairly minor.
> >
> > Overall, the response made a good case for the benefit of halving the number of ODE steps and stability of the results for different hyperparameter choices, addressing two out of three of my most important concerns (the issue about technical innovation remains). I already lean towards raise my score from 3 to 4 (at the end of the Author-Reviewer discussion period, so that I can write an accurate "Final Justification"), and addressing this final point would help.

---

> > > ### Author Response · Authors · 2026-04-06
> > >
> > > Thank you for your iterative feedback and for the opportunity to elaborate on our contribution beyond the rebuttal space. We are sincerely grateful for the reviewer's engagement.
> > >
> > > **The nature of our contribution.**
> > >
> > > Prop. 4.1 is indeed our main mathematical contribution. As a first argument, we agree that this formulation builds upon existing concepts, but, to us, this does not discount the value of the submission.
> > >
> > > Indeed, while ensuring mathematical rigor, our primary goal is to provide a more flexible approach to estimating likelihood ratios. Hence, we believe the novelty should be evaluated holistically through the lens of both the task and the generative modeling contribution (similar to other conference publications, such as flows for simulation-based inference [1]).
> > >
> > > In our opinion, the ability to customize ratio formulations by training a single conditional model is a significant advantage in DRE, and, to our knowledge, marks a novel direction beyond re-training and interpolating between all pairs of densities. Besides the ODE-based density ratio tracking, technical insights also involve heuristics surrounding likelihood estimation using flow matching schedulers (such as addressing singularities with parameterizations, see first-order error analysis for reviewer **U3Wf**).
> > >
> > > **Track compliance**
> > >
> > > The second point we would like to bring to attention is that our work is fundamentally application-motivated, and our primary area of submission is "Applications->Health / Medicine" (see above). The Call for Papers reads:
> > >
> > > `Innovative techniques, problems, and datasets that are of interest to the machine learning community and driven by the needs of end-users in applications...`
> > >
> > > Here, we think we are presenting an innovative technique, as flow-based generative likelihood ratio tracking is novel, and the design of our study is community-oriented. This, to us, is also a technical novelty, and the main reason behind Sec. 4.4 in the paper.
> > >
> > > Shortly, single-cell methods are still based on kNN purity or approximate VAEs estimation, and lack:
> > > * The advantage of exact likelihood estimation.
> > > * A parameterized conditional model enabling flexible design of likelihood ratios across settings.
> > >
> > > scRatio is a tool for all of the above, bridging multiple problems comparing cells under conditional distributions.
> > >
> > > **Choice of $b_t$**
> > >
> > > We have decided to investigate the choice of the $b_t$ a bit more, as kindly suggested by the reviewer.
> > >
> > > Recall the ratio ODE.
> > >
> > > $\frac{d}{dt}\log r_t=\nabla \cdot (u_t'-u_t) + (u_t'-b_t)\nabla\log p_t' + (b_t-u_t)\nabla\log p_t$.
> > >
> > > Assume that $b_t$, $u_t$ and $u_t'$ generate $q_t$, $p_t$ and $p_t'$, respectively, according to the continuity equation.
> > >
> > > A natural criterion for choosing $b_t$ is to minimize the expected squared rate of change of $\log r_t$ over $q_t$’s support, promoting smoother trajectories that remain in regions where both $p_t$ and $p_t'$ assign substantial probability, improving stability and reducing abrupt changes. First, note that:
> > >
> > > $$\mathbb{E}_{q_t}[\log r_t] = \mathbb{E}\_{q_t}\left[\log\frac{q_t}{p_t'}-\log\frac{q_t}{p_t}\right] = \mathrm{KL}[q_t||p_t']-\mathrm{KL}[q_t||p_t] = \Delta\mathrm{KL}(t)$$
> > >
> > > Using Jensen's inequality ($\mathbb{E}[X^2]\geq(\mathbb{E}[X])^2$):
> > >
> > > $$\int_0^1 \mathbb{E}\_{q_t}\left[\left(\frac{d}{dt}\log r_t\right)^2\right] dt
> > > \geq \int_0^1\left(\mathbb{E}_{q_t}\left[\frac{d}{dt}\log r_t\right]\right)^2 dt$$
> > >
> > > Under mild regularity assumptions allowing the exchange of derivative and expectation, it holds that.
> > >
> > > $$\int_0^1\left(\mathbb{E}\_{q_t}\left[\frac{d}{dt}\log r_t\right]\right)^2 dt = \int_0^1\left(\frac{d}{dt}\mathbb{E}_{q_t}\left[\log r_t\right]\right)^2 dt
> > > = \int_0^1 \left(\frac{d}{dt}\Delta \mathrm{KL}(t)\right)^2dt.$$
> > >
> > > Applying Cauchy-Schwarz:
> > >
> > > $$\int_0^1 \left[\frac{d}{dt}\Delta \mathrm{KL}(t)\right]^2dt
> > > \geq \left[\int_0^1 \frac{d}{dt}\Delta \mathrm{KL}(t)dt\right]^2.$$
> > >
> > > Ending up with
> > >
> > > $$\int_0^1 \mathbb{E}\_{q_t}\left[\left(\frac{d}{dt}\log r_t\right)^2\right] dt
> > > \geq (\Delta \mathrm{KL}(1))^2.$$
> > >
> > > Though this is *only a first-step insight*, it shows that $(\Delta \mathrm{KL}(1))^2$ lower bounds the cumulative squared change in $\frac{d}{dt}\log r_t$. Heuristically, one could choose the available $b_t$ whose terminal distribution $q_1$ lies in regions where both $p_1$ and $p_1'$ place large probability mass, using data as an approximation.
> > >
> > > We will add this to the paper.
> > >
> > > **Noise schedules**
> > >
> > > Thank you for the remark. Our schedule definition can be indeed confusing, therefore we will move to a general definition of $\sigma_t = \left[ \lambda^2 \, t \, (1 - t) + (1 - (1 - \sigma_{min}) \, t)^2 \right]^{1/2}$. If one sets $\lambda$, $\sigma_{min}$, or both of them to $0$, then one will get exactly the three schedules defined in the paper. Since combining these parameters is uncommon in the literature, we will add this definition to Appendix A.4.
> > >
> > >
> > > [1] Wildberger et al. "Flow matching for scalable simulation-based inference."

---

### Official Review · Reviewer_EPKE · 2026-03-11

**Soundness:** 2
**Presentation:** 2
**Significance:** 2
**Originality:** 2
**Overall Recommendation:** 3
**Confidence:** 3

**Summary:**

This paper proposes a likelihood-ratio estimation framework based on conditional continuous normalizing flows trained with flow matching. Rather than estimating likelihoods under two models separately and then taking their ratio, the method derives a single dynamical formulation to track density ratios along generative trajectories. The approach is evaluated on synthetic benchmarks with closed-form ratios and further demonstrated on single-cell genomics applications, including conditional comparison of cellular states, perturbation analysis, and batch correction evaluation. Overall, the paper presents an interesting probabilistic formulation for density-ratio estimation and explores its relevance to high-dimensional biological data analysis.

**Compliance With Llm Reviewing Policy:**

Affirmed.

**Key Questions For Authors:**

I do not have additional key questions for the authors at this stage.

**Limitations:**

The main limitation is that the biological application setting is relatively limited and not itself novel, as the single-cell tasks considered are already well studied. In addition, the current experiments do not yet provide sufficient evidence that the proposed approach offers clear advantages over existing methods. The discussion of limitations would be stronger if it more explicitly acknowledged these points.

**Strengths And Weaknesses:**

**Strength**： The main strength of the paper is methodological neatness rather than broad empirical impact: it gives a reasonably clean CNF-based formulation for likelihood-ratio estimation, and the synthetic results suggest that the approach is technically workable in controlled settings.

**Weakness:**
1. Although the method is evaluated on synthetic density-ratio benchmarks and several single-cell use cases, the empirical validation in the single-cell setting is not yet fully convincing. In particular, the manuscript does not include sufficiently strong comparisons against highly relevant generative and perturbation-modeling baselines commonly used in single-cell analysis, such as scVI, scGen, CPA, CellOT, or established batch-correction evaluation frameworks. As a result, it remains unclear whether the proposed likelihood-ratio formulation provides practical advantages over existing latent-variable or conditional modeling approaches on the downstream tasks considered.

2. The biological applications are potentially interesting, but at present they mainly illustrate possible uses of the method rather than establishing a clear practical benefit. The tasks considered—condition comparison, perturbation effect estimation, and batch correction assessment—are already well studied in the single-cell literature, and the manuscript does not yet demonstrate that explicit density-ratio estimation yields more informative, robust, or actionable results than existing approaches. Consequently, the practical value of the method in real biological analysis remains somewhat under-supported.

3. While the method appears technically well motivated, its broader significance within the single-cell analysis ecosystem is not yet fully established. The paper presents an interesting probabilistic tool, but the single-cell component currently reads more as an application domain than as a setting in which the method enables a qualitatively new capability. Without stronger evidence of necessity or clear performance gains over existing models, the manuscript does not yet fully justify why this approach should become a preferred framework for single-cell conditional comparison tasks.

---

> ### Author Rebuttal · Authors · 2026-03-30
>
> We thank the reviewer for their feedback and hope this reply clarifies the raised points and supports a more positive evaluation of our work.
>
> > A1. Missing Comparisons
>
> We highlight that the baselines proposed by the reviewer solve different tasks from those of the paper, and are therefore not directly relevant to our evaluation. We will clarify the differences in what follows.
>
> **Our model and task**
>
> scRatio estimates *exact log-likelihood ratios* under alternative conditional distributions. In other words, it scores cells by asking:
>
> "Are cell states more likely under condition $y$ than $y'$?"
>
> * In perturbation settings, we score regions of the cellular manifold that are more likely under perturbation than control.
> * In batch-correction evaluation, we assess whether batch labels carry meaningful information in cell embeddings before and after correction.
>
> **Unrelated baselines**
>
> The suggested baselines are, instead, *predictive models* and solve different mathematical and applied problems:
>
> * scGen, CPA, CellOT: perturbation prediction methods that approximately parameterize cellular response to treatment. They *generate new cells* under perturbations but do not score likelihood ratios across distributions as scRatio does.
>
> * scVI: a batch-correction method that removes technical effects from embeddings. It is not designed to score perturbation effects or quantify batch correction and therefore does not overlap with our task.
>
> > A2. Included comparisons.
>
> We also note that Sec. 5.3 includes domain-specific baselines. In particular, MrVI belongs to the latent-variable, conditioned VAE class that the reviewer claims is missing from our benchmarks.
>
> > A3. Additional limits
>
> Thanks for the comment. We address all the points raised below.
>
> **Possible uses vs. clear benefits**
>
> The selected tasks (differential abundance, batch-correction evaluation) are standard pipelines for single-cell data. Demonstrating scRatio's compatibility with *community-accepted* tasks, relative to existing methods, already illustrates its practical benefits.
>
> **Well-studied task**
>
> Our work introduces a theoretically principled *methodological update to existing approaches*, so it is natural to frame scRatio within established tasks. This aligns with the principle of methodological papers at technical conferences: to demonstrate the reliability of a model in well-understood application settings.
>
> **Actionable results**
>
> Beyond showing closed-form ratio estimation, we *do provide evidence* that scRatio consistently outperforms existing models on the perturbation effect prediction task (see Tab. 2 and Sec. 5.3). MrVI and MELD are established differential abundance estimation approaches for single cells, and our model consistently overcomes them in a controlled setting.
>
> We also note that, while batch-correction quantification already exists, scRatio is the first to frame it as a likelihood ratio using generative models. Our results show that the ratios capture expected trends before and after correction.
>
> > A4. Novel capabilities
>
> scRatio also introduces novel capabilities as the first method implementing versatile flow-based likelihood ratio estimation with conditional generative models. Its flexibly parameterized conditional flows allow approximation of *any* likelihood ratio $\frac{p(\mathbf{x}|y)}{p(\mathbf{x}|y')}$ for different conditioning variables $y$ and $y'$.
>
> This design enables all presented tasks within a **single model** by simply changing the conditioning. In contrast, all the baselines targeting perturbation predictions and batch effect evaluation are task-specific. To our knowledge, no existing single-cell model enables as flexible a likelihood ratio design and distribution comparison as scRatio.
>
> > A5. New experiment on patient stratification
>
> To broaden the biological scope of scRatio, we add an experiment focusing on patient-specific treatment response estimation. You find the results in Fig. **R.11** at [link](https://figshare.com/s/46bb00e542dcce432831).
>
> Using a large PBMC dataset [1] with 12 donors and 90 cytokine treatments across 10M cells, we compute $\log$-likelihood ratios comparing treated versus control cells for *each donor-cytokine combination*. Results show that scRatio captures donor-specific responses, with higher scores for treatments known to induce differential effects, and near-zero scores for weak or uniform responses, supporting its utility for patient stratification in large perturbation studies.
>
> > A6. Limitation section
>
> We updated the section with a short paragraph.
>
> `From an application perspective, the biological case studies we consider focus on established single-cell analysis tasks. Our goal in this work is primarily to introduce a methodological advance for versatile likelihood ratio estimation across relevant scenarios. Further research will be dedicated to exploring the approach across a broader range of biological applications.`
>
> [1] doi.org/10.64898/2025.12.12.693897

---

> > ### Author Rebuttal · Reviewer_EPKE · 2026-04-05
> >
> > all complete

---

> > > ### Author Response · Authors · 2026-04-06
> > >
> > > We sincerely thank the reviewer for confirming that all concerns have been fully addressed. Given that our clarifications and additional results resolved the issues, we hope the updated score will reflect this resolution. We truly appreciate their time invested in the review process.

---

### Official Review · Reviewer_U3Wf · 2026-03-12

**Soundness:** 3
**Presentation:** 3
**Significance:** 3
**Originality:** 3
**Overall Recommendation:** 5
**Confidence:** 4

**Summary:**

The paper introduces scRatio, a flow‑based method to estimate likelihood ratios
between (conditional) continuous normalizing flow models by tracking a single
dynamical quantity along generative trajectories. It leverages flow‑matching learned
vector fields and score estimators to compute log‑ratios efficiently with one simulation
instead of separately evaluating two costly likelihood integrals.

**Compliance With Llm Reviewing Policy:**

Affirmed.

**Final Justification:**

I have read the paper carefully and considered the authors’ rebuttal. The manuscript presents an interesting and novel generative approach to learning density ratios, a challenging problem where existing methods often struggle to scale to high dimensions or to handle unbounded ratios, even in simple Gaussian settings. The proposed method appears to scale more effectively in high‑dimensional regimes, which is a meaningful advance. My original score was 4; after the rebuttal and subsequent discussion, I raise it to 5.

**Key Questions For Authors:**

How sensitive are results to the choice of $\alpha_t$ and $\sigma_t$ schedules?
Do you have recommended default values?

There is a large literature on density‑ratio estimation; could you discuss the advantages
and disadvantages of your method relative to representative existing approaches?

Please also include a comparison with non‑diffusion, non‑flow methods (for example,
classifier‑based estimators), evaluating when each class of method is preferable.

**Limitations:**

Yes.

**Strengths And Weaknesses:**

Strengths
The paper addresses a practical bottleneck: computational cost of evaluating likelihood ratios
for exact‑likelihood generative models, and motivates applications in genomics (single‑cell comparisons).

Deriving an ODE for the log‑ratio that can be tracked along a single simulated trajectory is
conceptually neat and reduces inference cost compared with the naive two‑integral approach.

 Experiments include synthetic closed‑form ratio benchmarks, mutual information estimation, runtime
comparisons, and a genomics use case, showing both accuracy and efficiency gains in several settings.

Weaknesses

The approach requires accurate conditional vector fields and score estimators; the paper shows
promising results but gives limited analysis of how estimation error in those components affect density ratio
estimation.

The score reparameterization (Eq. 11) can involve small denominators for some schedules;
while the authors adopt score regression to mitigate this, more diagnostics analysis would help.

The paper compares to TSM/CTSM and a naive baseline, but additional ablations (e.g., sensitivity
to scheduler choice, score vs. field parameterization, and sample complexity) would clarify robustness.

---

> ### Author Rebuttal · Authors · 2026-03-29
>
> We thank the reviewer for the helpful feedback. We address the concerns point by point, where experiments are available [here](https://figshare.com/s/46bb00e542dcce432831), referenced by file name (R.x, x = fig. number).
>
> > A1. Estimation Error
>
> We add the following experiment:
>
> *Data.* We use the simulation from Sec. 5.1 with 100k samples, estimating the likelihood ratio between a Gaussian with mean $\mu=2 \cdot \mathbb 1_d$ ($\mathbb 1_d$ --> d-dimensional vector of ones) and a standard normal.
>
> *Experiment.* We train score and velocity networks for 100k iterations, saving checkpoints at various steps. These represent different approximation qualities. Ratios are then computed from all score/velocity checkpoint pairs.
>
> *Results.* Fig. **R.5** shows that poorer neural approximations reduce ratio accuracy. Performance is generally more sensitive to the velocity field than the score, suggesting the latter could be trained for fewer steps. In higher dimensions ($d=30$), score accuracy becomes more important.
>
> > A2. Reparametrizations.
>
> We add **R.7**, where we empirically show that reparameterization instabilities significantly degrade predictions compared to our method.
>
> > A3. Ablations.
>
> **$\alpha_t / \sigma_t$**
>
> We fix $\alpha_t=t$, as linear interpolants are standard in flow matching. We instead vary the path variance $\sigma_t$ as a function of the following parameters (Sec. 3.1, L152–159):
> - $\lambda$: stochasticity along the path.
> - $\sigma_{\min}$: Gaussian noise at $t=1$ around the data.
>
> *Gaussians*. Ablations in **R.8** show that deterministic variants ($\lambda=0$) with small $\sigma_{\min}$ achieve the best performance on this simple task. In low dimensions, stochastic and deterministic paths perform similarly.
>
> *MI estimation*. **R.2** shows that increasing $\lambda$ does not help for small $d$, but improves performance in high dimensions, suggesting stochasticity helps bridge numerical gaps and mitigate the curse of dimensionality (more evidence in **R.4**).
>
> **Neural parameterization**
>
> We ablate model and training components, including state and time-feature dimensions, learning rate, and condition embedding size. **R.9** shows that most variation is driven by the learning rate (lower performs better), with no clear trend for other components.
>
> **Sample complexity**
>
> Using the best models, we analyze sample complexity in the Gaussian setting of Sec. 5.1. Results (**R.6**) across training sizes show strong degradation at small $N$, as limited data reduces likelihood generalization. For $N>30k$, performance stabilizes, but this threshold is likely application-dependent.
>
> > A4. Related work
>
> **Classification.** Common approaches estimate ratios via classification (e.g., Noise Contrastive Estimation or Telescoping Ratio Estimation). While simple, performance degrades in high dimensions when distribution supports differ, since the classifier saturates and produces poor ratio estimates.
>
> scRatio models likelihoods directly via normalizing flows, providing explicit ratio estimation and improved robustness to dimensionality through its generative formulation. As noted in the Limitations section, the lack of shared support can still affect our method.
>
> **Moment matching / kernel-based (KLIEP).** These approaches do not need direct density parameterization and use matching moments or divergences through importance reweighting. While theoretically supported, they scale poorly and depend heavily on kernel choices.
>
> In contrast, scRatio trains a flow matching model to estimate densities while tracking ratios during simulation, yielding a more flexible neural estimator.
>
> **Score-based (TSM/CTSM).** These methods define interpolation paths $p_t$ between distributions and compute ratios via the integral of the time-score $\partial_t \log p_t(\mathbf{x})$. CTSM, specifically, derives a closed form for $p_t$ via flow matching.
>
> scRatio instead uses ODEs to transform prior noise into the densities in the ratio, without assuming a specific interpolation path. Once a conditional CNF is trained, ratios for any conditioning pair can be computed without retraining, unlike TSM/CTSM, which must retrain for each pair.
>
> > A5. Classifier baseline.
>
> We add classifier-based ratio estimation to the synthetic experiments using $\log \frac{p(y|\mathbf{x})}{p(y'|\mathbf{x})}$ as a proxy. While unsuitable for class imbalance, it fits our setup with equally sampled distributions. The classifier is a neural network optimized across various depths and widths.
>
> *MI estimation*: (**R.2**): classifier performance is worse than scRatio across all settings.
>
> *Gaussians*: In **R.10**, it remains competitive for the low-dimensional scenario with strong support overlap, but performance drops sharply as dimensionality increases or overlap decreases.
>
> Overall, classifiers work in simple synthetic contexts but struggle in high-dimensional embeddings or complex distribution shifts typical of biomedical datasets.

---

> > ### Author Rebuttal · Reviewer_U3Wf · 2026-04-02
> >
> > Thank you to the authors for their rebuttal and for the additional experiments involving the density ratio of Gaussians. However, the results would be more convincing if a basic error analysis, even assuming Gaussian densities, were provided in addition to the numerical experiments.

---

> > > ### Author Response · Authors · 2026-04-04
> > >
> > > Thank you very much for the follow-up. Apologies, we interpreted the comment as a request for an empirical study. Below, we present a first-order error analysis confirming our results.
> > >
> > > **General case**
> > >
> > > For simplicity, we let $\hat{u}_t$ and $\hat{u_t}'$ be the neural approximations of the numerator and denominator vector fields $u_t$ and $u_t'$, and $\hat{s}_t'$ that for the score $\nabla \log p_t'$. We use $u_t$ as the simulation field in Prop 4.1.
> > >
> > > The derivatives of the true and approximate time-marginal ratios are:
> > >
> > > $$\frac{d}{dt}\log r_t = \nabla \cdot (u_t'-u_t) + (u_t'-u_t)^{\top}\nabla \log p'_t$$
> > >
> > > $$\frac{d}{dt}\hat{\log r_t} = \nabla \cdot (\hat{u}_t'-\hat{u}_t) + (\hat{u}_t'-\hat{u}_t)^{\top}\hat{s}_t'$$
> > >
> > > We define the approximation deltas as follows:
> > >
> > > $$\delta_u = \hat{u}_t - u_t$$
> > >
> > > $$\delta_{u'} = \hat{u}_t' - u_t'$$
> > >
> > > $$\delta_{s} = \hat{s}_t' - \nabla \log p_t'$$
> > >
> > > We want to find a bound for the compounding errors along the neural ODE. We define
> > >
> > > $$\frac{d}{dt}\epsilon_t = \frac{d}{dt}\hat{\log r_t} - \frac{d}{dt}\log r_t$$
> > >
> > > and seek to bound:
> > >
> > > $$|\epsilon_1| = \left| \int_0^1 \frac{d}{dt}\epsilon_t \, dt\right| ,$$
> > >
> > > where, at the prior, we assume $\epsilon_0=0$. Given the equations for $\delta_u$, $\delta_{u'}$, and $\delta_s$ above, we can rewrite the approximated ratio as follows.
> > >
> > > \begin{equation}
> > > \frac{d}{dt}\hat{\log r_t} = \nabla \cdot ((u_t'+\delta_{u'})  - (u_t + \delta_{u})) + ((u_t'+\delta_{u'})  - (u_t + \delta_{u}))^{\top}(\nabla \log p_t' + \delta_s)
> > > \end{equation}
> > >
> > > Plugging this into the equation for $\frac{d}{dt}\epsilon_t$ we get:
> > >
> > > \begin{equation}
> > >     \frac{d}{dt}\epsilon_t = \nabla \cdot (\delta_{u'}-\delta_{u}) + (\delta_{u'}-\delta_{u})^{\top}\nabla \log p_t' + (u_t'- u_t)^{\top}\delta_s + (\delta_{u'}-\delta_u)^{\top}\delta_s
> > > \end{equation}
> > >
> > > Since we are doing a first-order error analysis, we drop the $(\delta_{u'}-\delta_u)^{\top}\delta_s$ term.
> > >
> > > As we need absolute errors, we use the triangle inequality:
> > >
> > > $$|\epsilon_1| = \left| \int_0^1 \frac{d}{dt}\epsilon_t \, dt \right| \leq   \int_0^1 \left|\frac{d}{dt}\epsilon_t \right|dt$$
> > >
> > > The Cauchy-Schwarz theorem yields:
> > >
> > > $$|\epsilon_1| \leq \int_0^1 \left( |\nabla \cdot (\delta_{u'}-\delta_{u})| + ||\delta_{u'}-\delta_{u}|| \cdot ||\nabla \log p_t'|| + ||u_t'-u_t|| \cdot ||\delta_s|| \right) dt$$
> > >
> > > Assuming $||u_t-u_t'||\leq L_u$ (reasonable in flow matching), we have
> > >
> > > $$|\epsilon_1| \leq \int_0^1 \left( |\nabla \cdot (\delta_{u'}-\delta_{u})| + ||\delta_{u'}-\delta_{u}||\cdot ||\nabla \log p_t'||  + L_u ||\delta_s|| \right) dt \quad\quad(1)$$
> > >
> > > **Gaussian case**
> > >
> > > In the Gaussian example, we estimate $\log r_t=\log p_1/p_1'$, where:
> > >
> > > $$p_1=\mathcal{N}(c \cdot \mathbb{1}_d, I_d)$$
> > >
> > > $$p_0=p_0'=p_1'=\mathcal{N}(0_d, I_d)$$
> > >
> > > and use flow matching with $\sigma_t=1-t$ and $\alpha_t=t$. Here, the bound becomes clearer. First, note that if a target distribution is Gaussian, the marginal distribution:
> > >
> > > $$p_t(\mathbf{x}_t)= \int p_t(\mathbf{x}_t \mid \mathbf{x}_1)q(\mathbf{x}_1) \, d\mathbf{x}_1$$
> > >
> > > is also normal. From the marginalization of Gaussians, we get:
> > >
> > > * $p_t = \mathcal{N}(t \, c \, \mathbb{1}_d,\tilde{\sigma}_t^2 I_d)$
> > > * $p_t' = \mathcal{N}(0, \tilde{\sigma}_t^2 I_d)$
> > >
> > > where $\tilde{\sigma}_t$ is a function of $\alpha_t$ and $\sigma_t$. Now, we analyze the bounding constants of the error in (1):
> > >
> > > 1. *Gaussian score*
> > >
> > > $$||\nabla \log p_t'(\mathbf{x}_t)||= \left\|-\frac{\mathbf{x}_t}{\tilde{\sigma}_t^2}\right\|=\frac{1}{\tilde{\sigma}_t^2}||\mathbf{x}_t||$$
> > >
> > > Where we highlight that $\mathbb{E}[||\nabla \log p_t'(\mathbf{x}_t)||]\approx \frac{\sqrt{d}}{\tilde{\sigma}_t}$.
> > >
> > > 1. *Velocity difference*. We use the relationship between score and vector field in Tab. 4 in the appendix, since the score is tractable for Gaussian paths.
> > >
> > > $$||u_t'(\mathbf{x}_t) - u_t(\mathbf{x}_t)|| = \frac{1-t}{t}\left\|-\frac{\mathbf{x}_t}{\tilde{\sigma}_t^2}+\frac{\mathbf{x}_t - t c \mathbb{1}_d}{\tilde{\sigma}_t^2}\right\|=\frac{1-t}{\tilde{\sigma}_t^2}||c\cdot \mathbb{1}_d||=c \frac{\sqrt{d}\,(1-t)}{\tilde{\sigma}_t^2}$$
> > >
> > > * Finally, taking the expectation of (1) and plugging our results in, we get:
> > >
> > > $$
> > > \mathbb{E}[|\epsilon_1|]
> > > \lesssim
> > > \int_0^1
> > > \Big(
> > > \mathbb{E}[|\nabla \cdot (\delta_{u'}-\delta_u)|]
> > > +
> > > \frac{\sqrt{d}}{\sigma_t}  \mathbb{E}[||\delta_{u'}-\delta_u||]
> > > +
> > > c \frac{\sqrt{d}\,(1-t)}{\sigma_t^2}  \mathbb{E}[||\delta_s||]
> > > \Big) dt
> > > $$
> > >
> > >
> > > **Connection to the empirical results**
> > >
> > > The bound decomposes the error into three contributions:
> > > 1. The divergence error.
> > > 2. The velocity approximation error.
> > > 3. The score approximation error.
> > >
> > > Here, the terms 2. and 3. grow explicitly with $\sqrt{d}$, while 1. depends on the structure of the neural approximation errors. **In other words, compounding score and field approximation errors are, as expected, amplified with dimensions.**
> > >
> > > We will add this detailed analysis to the appendix.

---

### Official Review · Reviewer_yNCH · 2026-03-12

**Soundness:** 2
**Presentation:** 2
**Significance:** 2
**Originality:** 2
**Overall Recommendation:** 4
**Confidence:** 3

**Summary:**

This work introduces a novel approach for estimating the density ratio between conditional distributions under distinct conditions. The authors derive a velocity field for the log density ratio that can be computed using the learned conditional flow field from a pretrained conditional flow model. The proposed approach is compared to contemporary approaches for estimating the density ratio on benchmark tasks and then applied to single-cell genomics data.

**Compliance With Llm Reviewing Policy:**

Affirmed.

**Final Justification:**

The authors addressed my main concerns regarding the fairness of the empirical evaluation, and provided additional clarity on the practical use cases of their approach which was initially confusing to me. Therefore I have increased my score.

**Key Questions For Authors:**

1. I am confused by Figure 3. First, I believe that the log-density ratios are not plotted on the left as stated in the caption. Second, I am not sure what the takeaway from the UMAP plots are, and why they are included.
2. I agree with the authors’ statement in the limitation sections that scRatio may struggle for conditional distributions that have only a small mutual support, and further agree that this does not invalidate the approach, as I believe that for the majority of tasks this is not a concern. However, I am curious whether such cases can be detected without significant computational overhead? When I evaluate the likelihood ratio using the “naive approach”, I can see when the log-density of the denominator is very small, and know that it may be worth being skeptical of the ratio as a result. Is there such an equivalent heuristic for scRatio?

**Limitations:**

Yes.

**Strengths And Weaknesses:**

## Strengths:
1. The proposed method estimates likelihood ratio (any by extension the mutual information) with only one integration through the flow, and does not require training a new model to do so - it simply uses the existing conditional generative model.
2. ScRatio shows strong performance on benchmark tasks.

## Weaknesses:

1. I am not completely convinced by the last two empirical evaluations (batch correction evaluation and drug combination effect estimation) presented in this work. In the batch correction experiment, we see that when the likelihood ratio is supposed to decrease, it does. On the drug combination effect estimation, we see the same. These are reasonable sanity checks, but do not in my opinion constitute evaluations of the method presented by the authors. For example, these correlations would occur even if the estimated density ratio was extremely biased, as long as this bias was consistent. Therefore these experiments do not present strong evidence of good estimation of the likelihood ratio. On the other hand, I do not see what the utility of the method is in these applications. In the example of the drug combination effect, if the goal is to answer the question: “Does the combination of drugs have more of an effect than a single drug”, I would rather opt for the classifier based approach that the authors used to validate their method, then a likelihood-ratio based approach.
2. The empirical evaluation relative to TSM/CTSM appears biased. From my understanding of the Appendix, the authors perform a hyperparameter search for their approach (scRatio) and report the best values, whereas only the default TSM/CTSM hyperparameters are reported. While the hyperparameters are not exactly comparable between TSM and scRatio due to the inherently different probability paths defined by the two approaches, it is important to disentangle the effect of improved flow model (through hyperparameter tuning) and the different approaches for estimating likelihood ratios. This is especially important given the huge variation in performance for the different scRatio design choices in Table 1.
3. Proposition 4.1 formalizes the main insight into the method introduced by the authors, namely that density ratios can be computed using only one simulation of the probability path trajectories. However, its current statement is too informal to constitute a proof. Most notably, the density ratio $r_t(x)$ is only defined when the marginal probability paths $p_t$ and $p_t’$ share the same support. This clearly does not hold for any pair of flow fields $u_t$ and $u_t’$. The authors could refine this proposition to clearly state what assumptions are being made for it to be valid.

---

> ### Author Rebuttal · Authors · 2026-03-29
>
> We thank the reviewer for the thorough review and hope our responses will support a higher score. New experiments are available [here](https://figshare.com/s/46bb00e542dcce432831), referred to as R.x (x = fig. number).
>
> > A1. Unconvincing evidence
>
> As performance quality on tractable log-ratios was tested before in our synthetic and semi-synthetic settings, our last sections demonstrate the *practical utility* of our tool in real biological scenarios. Note that the tasks we choose in said settings are not arbitrary, but reflect existing analytical pipelines and community-established evaluation settings (more below).
>
> > A2. Batch correction
>
> Most correction evaluations in scRNA-seq [1] assess batch mixing in two steps:
>
> * Compute neighborhoods around cells with kNN before and after correction.
> * Measure how batch label purity in the neighborhoods changes in the process.
>
> Our approach follows the same logic by tracking a proxy for batch mixing before and after correction, but with a different scoring principle. We evaluate whether cell embeddings are more likely under a distribution conditioned on both batch and cell type labels than on cell type alone (see A7.). This can be used to compare different correction approaches, aligning our tool with common practices while leveraging exact-likelihood generative models.
>
> > A3. Drug combination
>
> The drug combination experiment is another differential abundance (DA) test (Sec. 5.3), with the denominator conditioned on a single drug rather than a control. In single-cell analysis, enrichment under perturbations is most of the time treated as a likelihood-based task (kNN [2], manifold [3], VAE [4]), and applications to treatment combinations are established [5]. Perturbations alter cellular state frequencies, and, typically, the goal of the community is to track how the probability mass changes across the manifold, rather than to classify.
>
> While a classifier is useful to rank perturbations, it does not directly quantify DA, and in **R.10** we show that its likelihood ratio prediction poorly scales with dimensionality, as the performance saturates and fails to quantify likelihood shifts. We used it to validate that our average ratios capture overall responses while preserving a likelihood formulation and highlight the distribution of ratios as a function of perturbation effect in a new figure (**R.1**).
>
> > A4. Tuning
>
> We kindly refer to A2. to h3MX.
>
>
> > A5. Performance differences
>
> Performance differences across scRatio models indicate that stochastic interpolants improve ratio estimates in high dimensions by reducing sensitivity to exact trajectories. This is valid for TSM/CTSM as well, with high SB variance improving the performance in high-dimensional cases (see **R.4**).
>
> > A6. Assumptions
>
> Thank you for the suggestion. Here are the new and already present assumptions in the paper.
>
> * Conservation of mass (Prop. 4.1): implied by the Continuity Equation.
> * Shared initial densities (Prop. 4.1): required for the backward log-density dynamics to define an IVP for both densities.
> * Strict positivity of $p_t$ and $p'_t$ (Background, will be added explicitly to Prop. 4.1): ensuring the ratio is well-defined.
>
> Missing:
>
> * Shared support (to be added to Prop. 4.1): $p_t(x)>0 \iff p'_t(x)>0$ for all $x\in\mathbb{R}^d$, ensuring $r_t$ is defined for $t\in[0,1]$.
>
>
> > A7. Fig 3
>
> The histogram shows absolute log-density ratios $|\log p(\mathbf{x}|y,b)/p(\mathbf{x}|y)|$ before and after correction.
>
> Pipeline: (1) train a batch correction model on data $\mathbf{x}$, (2) obtain corrected data $\hat{\mathbf{x}}$, (3) compute absolute log-ratio histograms on $\mathbf{x}$ and $\hat{\mathbf{x}}$.
>
> Ratios near 0 indicate the batch variable carries little information and does not affect the conditional density beyond the biological label after correction. Taking the absolute value focuses on discrepancy magnitude, aiming for concentration around 0.
>
> The UMAPs visually highlight batch mixing before and after correction for the embeddings used by scRatio for the left plots.
>
> > A8. Failure Detection
>
> We provide two suggestions.
>
> (i) Low denominator values yield high likelihood ratios. With low shared support, $\log p(\cdot \mid y)/p(\cdot \mid y')$ will explode in either direction. One can verify this by computing histograms of log-ratios evaluated on points from $y$ and $y'$, where overlap indicates similar weighing of some manifold regions.
>
> (ii) When we track ratios, we simulate from data to noise under a single model. One can use $u_t(\cdot \mid y)$ as the simulation field ($b_t$ in Prop.4.1). If $y'$ samples are plausible under $p(\cdot \mid y)$, they will distribute as the prior when transported by $u_t(\cdot \mid y)$, which can be evaluated with a divergence or likelihood measure.
>
> [1] doi.org/10.1038/s41592-021-01336-8
>
> [2] doi.org/10.1038/s41587-021-01033-z
>
> [3] doi.org/10.1038/s41587-020-00803-5
>
> [4] doi.org/10.1038/s41592-025-02808-x
>
> [5] doi.org/10.1101/2025.06.03.657769

---

> > ### Author Rebuttal · Reviewer_yNCH · 2026-04-03
> >
> > I thank the authors for addressing my concerns. Given the improved clarity of the technical aspects of this work, as well as the strong performance on the baseline experiments (Gaussians and MI estimation), I am willing to raise my score.

---

> > > ### Author Response · Authors · 2026-04-06
> > >
> > > We sincerely thank the reviewer for the positive feedback and for their engagement during the rebuttal process. We are very pleased that our clarifications and the additional results fully addressed their concerns, and we appreciate their willingness to raise the score to reflect this improved evaluation of our work.

---

### Decision · Program_Chairs · 2026-04-30

**Decision:**

Accept (regular)

**Comment:**

This paper represents a unique case as multiple reviewers have indicated that they would increase their scores, but at the time of writing they have not yet done so (despite multiple requests from myself).  As a result, I am basing this meta-review on the substance of the reviews and ensuing discussions, rather than directly on the current scores.  Moreover, reviewer U3Wf has advocated for publication, and feels that the paper "presents an interesting and novel generative approach to learning density ratios."  The reviewer further concludes that the proposed method "appears to scale more effectively in high-dimensional regimes, which is a meaningful advance."  Given this, and the material of the discussions, I am recommending acceptance.  I urge the authors to consider modifications resulting from the discussion with reviewers, particularly the detailed discussions with U3Wf.